# E($n$) Equivariant Topological Neural Networks

**Claudio Battiloro**[*,1,†], **Ege Karaismailoğlu**[*,1,2], **Mauricio Tec**[*,1], **George Dasoulas**[*,1],
**Michelle Audirac**[1], **Francesca Dominici**[1]
[1]Harvard University, [2]ETH Zurich

## Abstract

Graph neural networks excel at modeling pairwise interactions, but they cannot flexibly accommodate higher-order interactions and features. Topological deep learning (TDL) has emerged recently as a promising tool for addressing this issue. TDL enables the principled modeling of arbitrary multi-way, hierarchical higher-order interactions by operating on combinatorial topological spaces, such as simplicial or cell complexes, instead of graphs. However, little is known about how to leverage geometric features such as positions and velocities for TDL. This paper introduces E(n)-Equivariant Topological Neural Networks (ETNNs), which are E(n)-equivariant message-passing networks operating on combinatorial complexes, formal objects unifying graphs, hypergraphs, simplicial, path, and cell complexes. ETNNs incorporate geometric node features while respecting rotation, reflection, and translation equivariance. Moreover, being TDL models, ETNNs are natively ready for settings with heterogeneous interactions. We provide a theoretical analysis to show the improved expressiveness of ETNNs over architectures for geometric graphs. We also show how E(n)-equivariant variants of TDL models can be directly derived from our framework. The broad applicability of ETNNs is demonstrated through two tasks of vastly different scales: i) molecular property prediction on the QM9 benchmark and ii) land-use regression for hyper-local estimation of air pollution with multi-resolution irregular geospatial data. The results indicate that ETNNs are an effective tool for learning from diverse types of richly structured data, as they match or surpass SotA equivariant TDL models with a significantly smaller computational burden, thus highlighting the benefits of a principled geometric inductive bias. Our implementation of ETNNs can be found here.

## 1 Introduction

Graph Neural Networks (GNNs) are employed across various fields, including computational chemistry (Gilmer et al., 2017a), physics simulations (Shlomi et al., 2020), and social networks (Xia et al., 2021), to highlight a few. Their effectiveness stems from merging neural network adaptability with insights about data relationships through the graph topology. Research on GNNs covers a broad range of categories, mainly divided into spectral (Bruna et al., 2014) and non-spectral (Gori et al., 2005). In both cases, GNNs aim to learn representations for node (and/or) edge attributes through local aggregation driven by the graph topology, essentially defining message passing networks (MPN) (Gilmer et al., 2017c). Utilizing this capability, GNNs have delivered significant achievements in tasks like node and graph classification (Kipf & Welling, 2017), link prediction (Zhang & Chen, 2018), and specific challenges such as protein folding (Jumper et al., 2021).

However, GNNs usually struggle to model higher-order and multi-way interactions that the intrinsically pairwise structure of graphs cannot capture. Consequently, the field of *Topological Deep Learning* (TDL) (Barbarossa & Sardellitti, 2020; Hajij et al., 2022b; Papamarkou et al., 2024) started to gain significant interest as it seeks to address these limitations. Although the general scope of TDL is designing models for data defined over a variety of topological spaces, the main focus is usually on combinatorial topological spaces (CTS), i.e., topological spaces that can be described combinatorially, such as cell complexes (Grady & Polimeni, 2010; Bodnar, 2023; Battiloro, 2024). Leveraging this

---

[*]Equal contribution. Author ordering determined by a random number generator. [†]Corresponding author.
{cbattiloro,ekaraismailoglu,mauriciogtec,maudirac}@hsph.harvard.edu,
georgios_dasoulas@hms.harvard.edu

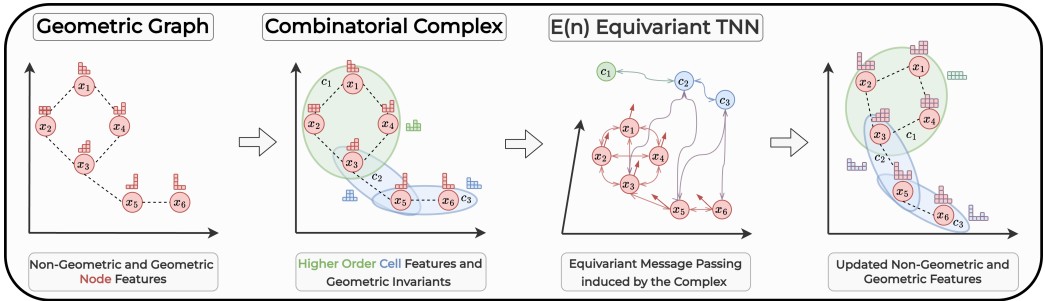

Figure 1: **Overview of the $E(n)$-Equivariant Topological Neural Networks framework.** An input geometric graph/point cloud with (possibly) non-geometric features is provided. A combinatorial complex, whose elements are called cells, is constructed from the input geometric graph/point cloud to encode higher-order hierarchical interactions, based on topological or domain-specific considerations. If available, higher-order cell features can be injected. Geometric invariants for the cells (e.g. pairwise distances, Hausdorff distances, volumes, ...) are computed. Finally, $E(n)$ Equivariant Topological Neural Networks update both geometric and non-geometric features to improve a downstream task while respecting rotation, reflection, and translation equivariance.

combinatorial structure, it is possible to design more sophisticated adjacency schemes among *cells*, i.e., single nodes or groups of nodes, than the usual adjacency in graphs. Message passing networks over CTS have been proven to be more expressive (Bodnar et al., 2021b; Battiloro et al., 2023c; Bodnar et al., 2021a; Truong & Chin, 2024; Jogl et al., 2023) (using the Weisfeiler-Lehman test criterion (Xu et al., 2019)), more suitable to handle long-range interactions (Giusti et al., 2023b;a), and more effective to handle heterophilic data settings (Bodnar et al., 2022; Hansen & Gebhart, 2020) than classical GNNs. Cell complexes are powerful objects to model certain classes of higher-order interactions along with a hierarchical structure among them, while general combinatorial objects like hypergraphs are useful to model arbitrary higher-order interactions, but without any hierarchy. Hajij et al. (2022b) introduced the notion of *Combinatorial Complex* (CC) to provide a formal, flexible, yet simple structure that pseudo-generalizes simplicial/path/cell complexes and hypergraphs, thus enabling the design of arbitrary higher-order interactions along with a hierarchical structure. They also introduced message passing networks over combinatorial complexes.

Little is known about how to integrate and leverage geometric information of the data in such TDL models. In this context, nodes are embedded in a manifold or, in general, in a metric space, and have geometric features, such as positions or velocities. In the case of geometric graphs, a common practice is designing message passing networks able to respect some suitable symmetry, i.e., being equivariant w.r.t. the action of some symmetry group. Here we focus on $E(n)$-equivariant neural networks, i.e., networks being equivariant w.r.t. rotations, reflections, and translations. The $E(n)$-Equivariant Graph Neural Network (EGNN) from Satorras et al. (2021) is one of the simplest yet powerful equivariant models for graph-structured data. Eijkelboom et al. (2023) generalized the approach from Satorras et al. (2021) to simplicial complexes (a particular instance of cell complexes), introducing $E(n)$-Equivariant Message Passing Simplicial Networks (EMPSNs), able to handle graph or simplicial-structured data. However, as we will show in the paper, both graphs and simplicial complexes are combinatorial objects often not flexible enough to model arbitrary higher-order interactions, and a general framework for designing equivariant neural networks over CTS is missing. Appendix B presents a review of related works.

**Contribution.** We generalize the approaches from Satorras et al. (2021) and Eijkelboom et al. (2023) to design $E(n)$-*Equivariant Topological Neural Networks* (ETNNs), i.e., $E(n)$-Equivariant Message Passing Networks over Combinatorial Complexes. CCs represent a pseudo-generalization of graphs, simplicial complexes, path complexes, cell complexes, and hypergraphs, therefore ETNNs represent a legitimate framework for designing arbitrary $E(n)$-Equivariant Message Passing Networks over combinatorial topological spaces or, more in general, over combinatorial objects used to model interactions among entities, such as hierarchical graphs. An ETNN layer is made of a message passing round over the CC comprising scalar invariants of the geometric features and a learnable update of the geometric features added up in the current directions. As such, ETNNs are a scalarization

method (Han et al., 2022; 2024). We study the improved expressiveness of ETNNs by evaluating their ability to distinguish $k$-distinct geometric graphs (Joshi et al., 2023), showing that they are at least as expressive as other well-known scalarization-based $E(n)$ equivariant graph neural networks. Furthermore, we empirically show that ETNNs are also able to distinguish most of the counterexample structures from (Pozdnyakov et al., 2020; Joshi et al., 2023) that are indistinguishable using $k$-body scalarization. Generalizing (Hajij et al., 2022b) to the geometric setting, we show how several $E(n)$-equivariant variants of TDL models can be directly derived from our framework. As an additional contribution, we test ETNNs in applications where domain knowledge induces combinatorial complex structures that are more natural and parsimonious than the graphs or higher-order structures that are usually employed to model the same phenomena. Specifically, we show the efficacy of ETNNs on i) molecular properties prediction on the QM9 benchmark, and on ii) land-use regression for hyper-local estimation of air pollution with multi-resolution irregular geospatial data, representing also a completely new challenging benchmark for TDL models. Crucially, thanks to the flexible and principled inductive bias induced by the underlying combinatorial complexes, ETNNs match or surpass the results of EMPSNs with less than half of the memory usage and almost half of the runtime per epoch. The obtained results jointly validate ETNNs, our CC-based approach, and, in general, the injection of geometric information in TDL models.

## 2 COMBINATORIAL COMPLEXES AND EQUIVARIANCE

We review the fundamentals of *combinatorial complexes* (CCs), fairly general objects providing an effective and flexible way to represent higher-order multi-way interactions. Such structures generalize graphs, simplicial complexes, path complexes, cell complexes, and hypergraphs. We first define our domain of interest -*CCs*-, the relations among its elements -*neighborhood functions*-, the data defined on it -*topological signals*-, and a class of deep networks operating on it -*CC message passing networks* (CCMPNs)-(Hajij et al., 2022b). We finally introduce the formal notion of equivariance, showing some properties of CCMPNs.

**Combinatorial Complex.** A *combinatorial complex* (CC) is a triple $(\mathcal{S}, \mathcal{X}, \mathrm{rk})$ consisting of a set $\mathcal{S}$, a subset $\mathcal{X}$ of $\mathcal{P}(S)\backslash\{\emptyset\}$, and a function $\mathrm{rk} : \mathcal{X} \to \mathbb{Z}_{\geq 0}$ with the following properties:

1. for all $s \in \mathcal{S}$, $\{s\} \in \mathcal{X}$;

2. the function $\mathrm{rk}$ is order-preserving, i.e., if $x, y \in \mathcal{X}$ satisfy $x \subseteq y$, then $\mathrm{rk}(x) \leq \mathrm{rk}(y)$.

The elements of $\mathcal{S}$ and $\mathcal{X}$ are nodes and cells, respectively, and $\mathrm{rk}(\cdot)$ is the rank function. $\mathcal{X}$ simplifies notation for $(\mathcal{S}, \mathcal{X}, \mathrm{rk})$. The rank function flexibly organizes the cells hierarchically. Therefore, CCs can model both hierarchical (as in simplicial and cell complexes) and set-type (as in hypergraphs) relations. For each singleton cell $\{s\}$ in a CC, we set $\mathrm{rk}(\{s\}) = 0$, aligning CCs with simplicial and cell complexes. The rank of a cell $x \in \mathcal{X}$ is $k := \mathrm{rk}(x)$, and we call it a $k$-cell. The dimension $\dim(\mathcal{X})$ of a CC is the maximal rank among its cells.

**Neighborhood functions.** Combinatorial Complexes can be equipped with a notion of neighborhood among cells. In particular, we can define a "neighborhood" function $\mathcal{N} : \mathcal{X} \to \mathcal{P}(\mathcal{X})$ on $\mathcal{X}$ as a function that assigns to each cell $x$ in $\mathcal{X}$ a collection of "neighbor cells" $\mathcal{N}(x) \subset \mathcal{X}$. Usually, two main types of neighborhood functions are considered (Hajij et al., 2022b): *adjacencies*, in which all of the elements of $\mathcal{N}(x)$ have the same rank as $x$, and *incidences*, in which its elements are from a higher or lower rank. A generally applicable choice for these functions is as follows. First, up/down incidences $\mathcal{N}_{I,\uparrow}$ and $\mathcal{N}_{I,\downarrow}$ are defined by the *containment* criterion

$$\mathcal{N}_{I,\uparrow}(x) = \{y \in \mathcal{X}|\mathrm{rk}(y) = \mathrm{rk}(x) + 1, x \subset y\}, \quad \mathcal{N}_{I,\downarrow}(x) = \{y \in \mathcal{X}|\mathrm{rk}(y) = \mathrm{rk}(x) - 1, y \subset x\}. \tag{1}$$

Therefore, a $k + 1$-cell $y$ is a neighbor of a $k$-cell $x$ w.r.t. to $\mathcal{N}_{I,\uparrow}$ if $x$ is contained in $y$; analogously, a $k - 1$-cell $y$ is a neighbor of a $k$-cell $x$ w.r.t. to $\mathcal{N}_{I,\downarrow}$ if $y$ is contained in $x$. The incidences induce up/down adjacencies $\mathcal{N}_{A,\uparrow}$ and $\mathcal{N}_{A,\downarrow}$ by the *common neighbor* criterion

$$\mathcal{N}_{A,\uparrow}(x) = \{y \in \mathcal{X}|\mathrm{rk}(y) = \mathrm{rk}(x), \exists z \in \mathcal{X} : \mathrm{rk}(z) = \mathrm{rk}(x) - 1, z \subset y, \text{ and } z \subset x\},$$
$$\mathcal{N}_{A,\downarrow}(x) = \{y \in \mathcal{X}|\mathrm{rk}(y) = \mathrm{rk}(x), \exists z \in \mathcal{X} : \mathrm{rk}(z) = \mathrm{rk}(x) + 1, y \subset z, \text{ and } x \subset z\}. \tag{2}$$

Therefore, a $k$-cell $y$ is a neighbor of a $k$-cell $x$ w.r.t. $\mathcal{N}_{A,\uparrow}$ if they are both contained in a $k + 1$-cell $z$; analogously, a $k$-cell $y$ is a neighbor of a $k$-cell $x$ w.r.t. $\mathcal{N}_{A,\downarrow}$ if they both contain a $k - 1$-cell $z$.

While this choice of neighborhood functions applies to any CC, other neighborhood functions are more natural in specific applications. This scenario will be illustrated in Section 4.

**Topological signals.** A topological signal (or feature) over $\mathcal{X}$ is a mapping $f : \mathcal{X} \to \mathbb{R}$ from the set of cells $\mathcal{X}$ to real numbers. Therefore, the feature vectors $\mathbf{h}_x \in \mathbb{R}^F$ and $\mathbf{h}_y \in \mathbb{R}^F$ of cells $x$ and $y$ are a collection of $F$ CC signals, i.e., $\mathbf{h}_x = [f_1(x), \dots, f_F(x)]$ and $\mathbf{h}_y = [f_1(y), \dots, f_F(y)]$.

**Message passing networks over CCs.** Let $\mathcal{X}$ and $\mathcal{CN}$ be a CC and collection , i.e., a set, of neighborhood functions on it. The $l$-th layer of a CC Message Passing Network (CCMPN) updates the embedding $\mathbf{h}_x^l$ of cell $x$ as

$$\mathbf{h}_x^{l+1} = \beta \left( \mathbf{h}_x^l, \bigotimes_{\mathcal{N} \in \mathcal{CN}} \bigoplus_{y \in \mathcal{N}(x)} \psi_{\mathcal{N}, \mathrm{rk}(x)} \left( \mathbf{h}_x^l, \mathbf{h}_y^l \right) \right), \tag{3}$$

where $\mathbf{h}_x^0 := \mathbf{h}_x$ are the initial features, $\bigoplus$ is an intra-neighborhood permutation invariant aggregator, $\bigotimes$ is an inter-neighborhood (possibly) permutation invariant aggregator, and the rank- and neighborhood-dependent message functions $\psi_{\mathcal{N}, \mathrm{rk}(x)}$ and the update function $\beta$ are learnable functions. In other words, the embedding of a cell is updated in a learnable fashion through aggregated messages with its neighboring cells over a set of neighborhoods, generalizing what has been done with Graph MPN (Gilmer et al., 2017c). Interestingly. the message functions can be customized to incorporate additional information, such as the orientation of the cells (Barbarossa & Sardellitti, 2020; Sardellitti et al., 2021) (if available), or the embeddings of "common" cells (if present) like $x \cup y$ or $x \cap y$. Finally, please notice that CCMPN as in (3) are inherently able to model *heterogeneous* settings, as two cells can be neighbors in multiple neighborhoods, but messages among them will be dependent on the neighborhood itself. Thus, this property allows us to model different relation types through different (overlapping) neighborhoods. In Appendix C, we show how to model graphs, simplicial complexes, cell complexes, and hypergraphs through the CC framework, deriving the corresponding message passing architectures from the CCMPN in (3).

**Equivariances of CCMPNs.** Symmetries are framed as groups (Bronstein et al., 2021). Let $G$ be a group with an action $a : G \times \mathcal{Y} \to \mathcal{Y}$ on a set $\mathcal{Y}$. Given a function $e : \mathcal{Y} \to \mathcal{Y}$, it is said to be equivariant w.r.t. the action of $G$ if, for all $x \in \mathcal{X}$ and $g \in G$, it holds

$$e(a(g, x)) = a(g, e(x)). \tag{4}$$

If $e(a(g, x)) = e(x)$, $e$ is said to be invariant w.r.t. the action of $G$. As Graph MPNs (Gilmer et al., 2017c), CCMPNs are permutation equivariant too (Hajij et al., 2022b), i.e., they are equivariant w.r.t. the action of the symmetric group $\mathrm{Sym}(\mathcal{X})$ on $\mathcal{X}$. In other words, CCMPNs are equivariant to the relabeling of cells in the complex.

## 3 E(n) EQUIVARIANT TOPOLOGICAL NEURAL NETWORKS

Let us now introduce the setting in which nodes (0-cells) are embedded in some Euclidean space, i.e., they come with both non-geometric and geometric features, such as positions or velocities. In the following, we assume to have only positions for the sake of exposition, but velocities can be directly injected into our model as explained in Appendix D. We denote the non-geometric features with $\mathbf{h}_x \in \mathbb{R}^F$ for cell $x \in \mathcal{X}$ and the position with $\mathbf{x}_z \in \mathbb{R}^n$ for cell $z \in \mathcal{S}$. Since many problems exhibit $n$-dimensional rotation, reflection, and translation symmetries, it is desirable to design models that are equivariant w.r.t. the action of $E(n)$ (Satorras et al., 2021; Eijkelboom et al., 2023; Han et al., 2022). An element of the $E(n)$ group is a tuple $(\mathbf{O}, \mathbf{b})$ consisting of an orthogonal matrix $\mathbf{O} \in \mathbb{R}^{n \times n}$ (the rotation/reflection) and a vector $\mathbf{b} \in \mathbb{R}^n$ (the translation). The action $a$ of $E(n)$ on a vector $\mathbf{x} \in \mathbb{R}^n$ is given by

$$a((\mathbf{O}, \mathbf{b}), \mathbf{x}) = \mathbf{O}\mathbf{x} + \mathbf{b}. \tag{5}$$

As has been done for graphs (Satorras et al., 2021) and simplicial complexes (Eijkelboom et al., 2023), we represent equivariance by scalarization (Han et al., 2024). Therefore, geometric features undergo an initial transformation into invariant scalars. Subsequently, they are processed in a learnable way (e.g., MLPs) before being combined along the original directions to achieve equivariance.

**E(n) Equivariant Topological Neural Networks.** Let $\mathcal{X}$ and $\mathcal{CN}$ be a CC and collection of neighborhood functions on it. The $l$-th layer of an $E(n)$ Equivariant Topological Neural Network

(ETNN) updates the embeddings $\mathbf{h}_x^l$ of every cell $x \in \mathcal{X}$ and the position $\mathbf{x}_z^l$ of every node $z \in \mathcal{S}$ as

$$\mathbf{h}_x^{l+1} = \beta \left( \mathbf{h}_x^l, \bigotimes_{\mathcal{N} \in \mathcal{CN}} \bigoplus_{y \in \mathcal{N}(x)} \underbrace{\psi_{\mathcal{N}, \text{rk}(x)} \left( \mathbf{h}_x^l, \mathbf{h}_y^l, \text{Inv} \left( \{\mathbf{x}_z^l\}_{z \in x}, \{\mathbf{x}_z^l\}_{z \in y} \right) \right)}_{\mathbf{m}_{x,y}^{\mathcal{N}}} \right), \text{ for all } x \in \mathcal{X} \quad (6)$$

$$\mathbf{x}_z^{l+1} = \mathbf{x}_z^l + C \sum_{\mathcal{N} \in \mathcal{CN}} \sum_{t \in \mathcal{S}:\{t\} \in \mathcal{N}(z)} \left( \mathbf{x}_z^l - \mathbf{x}_t^l \right) \xi \left( \mathbf{m}_{z,t}^{\mathcal{N}} \right), \text{ for all } z \in \mathcal{S}, \quad (7)$$

where $\mathbf{h}_x^0 = \mathbf{h}_x$, $\mathbf{x}_z^0 = \mathbf{x}_z$, and $\xi$ is a scalar learnable function. Inv takes as input the positions of the nodes forming the involved cells, and it is an invariant function w.r.t. the action of $E(n)$, i.e., $\text{Inv}(\{\mathbf{Ox}_z^l + \mathbf{b}\}_{z \in x}, \{\mathbf{Ox}_z^l + \mathbf{b}\}_{z \in y}) = \text{Inv}(\{\mathbf{x}_z^l\}_{z \in x}, \{\mathbf{x}_z^l\}_{z \in y})$. We refer to a specific Inv as a *geometric invariant*. The update in (7) again naturally allows for neighborhood heterogeneities: two nodes can be neighbors via more than a single neighborhood function, carrying different messages. This is common in many applications, as illustrated in Section 4.

The following theorem demonstrates the equivariance property of ETNNs.

**Theorem 1 .** An ETNN layer as in (6)-(7), synthetically denoted as $\{\mathbf{h}_x^{l+1}\}_{x \in \mathcal{X}}, \{\mathbf{x}_z^{l+1}\}_{z \in \mathcal{S}} = \text{ETNN}\left(\{\mathbf{h}_x^l\}_{x \in \mathcal{X}}, \{\mathbf{x}_z^l\}_{z \in x}\right)$, is $E(n)$ equivariant, that is

$$\{\mathbf{h}_x^{l+1}\}_{x \in \mathcal{X}}, \{\mathbf{Ox}_z^{l+1} + \mathbf{b}\}_{z \in x} = \text{ETNN}\left(\{\mathbf{h}_x^l\}_{x \in \mathcal{X}}, \{\mathbf{Ox}_z^l + \mathbf{b}\}_{z \in x}\right), \quad (8)$$

for all $(\mathbf{O}, \mathbf{b}) \in E(n)$.

*Proof.* See Appendix E. $\qquad\square$

When only (6) is applied but the update step in (7) is not performed, then ETNNs are $E(n)$-invariant. At this point, it is important to discuss geometric invariants. As has been shown in (Satorras et al., 2021; Eijkelboom et al., 2023), geometric invariants should make use of the underlying topological structure. For instance, the pairwise distance of connected nodes has been used in EGNN (Satorras et al., 2021), while the simplex volume has been used in EMPSN (Eijkelboom et al., 2023). However, both EGNN and EMPSN operate on rigid domains, such as simplicial complexes, while CCs can have a much more arbitrary set of relations. In the following, we present a series of geometric invariants that can be used in ETNNs, without assuming any specific structure for the CC. We consider two cells $x$ and $y$, along their corresponding node positions $\{\mathbf{x}_z\}_{z \in x}$ and $\{\mathbf{x}_z\}_{z \in y}$.

**Permutation invariant functions of pairwise distances.** Any function of the form

$$\bigoplus_{z \in x, t \in y} \tau \left( \|\mathbf{x}_z - \mathbf{x}_t\| \right) \quad (9)$$

is a geometric invariant, with $\bigoplus$ being a permutation invariant aggregator, and $\tau$ a (possibly) learnable function. A basic instance is $\bigoplus = \sum$ and $\tau = \text{Id}$, i.e., the sum of pairwise distances.

**Distances of permutation invariant functions.** Any function of the form

$$\tau \left( \left\| \bigoplus_{z \in x} \mathbf{x}_z - \bigoplus_{t \in y} \mathbf{x}_t \right\| \right) \quad (10)$$

is a geometric invariant, with $\bigoplus$ being a linear permutation invariant aggregator, and $\tau$ a (possibly) learnable function. A basic instance is $\bigoplus = \frac{1}{|\cdot|} \sum$ and $\tau = \text{Id}$, i.e., the distance between centroids.

**Hausdorff distance.** The Hausdorff distance, defined as

$$\max \left\{ \max_{z \in x} \min_{t \in y} \|\mathbf{x}_z - \mathbf{x}_t\|, \max_{t \in y} \min_{z \in x} \|\mathbf{x}_t - \mathbf{x}_z\| \right\} \quad (11)$$

is a geometric invariant. The Hausdorff distance measures the mutual proximity of two sets whose elements live in a metric space. It does so by computing the maximal distance between any point of one set to the other set. Moreover, the two $\max \min$ terms inside the outer $\max$, usually referred to as directed Hausdorff distances, can be used as geometric invariants themselves. This geometric

invariant has been used in various machine learning applications, especially in the area of computer vision (Aydin et al., 2021; van Kreveld et al., 2022).

**Volume of the convex hull.** Any function of the volumes of the convex hulls of $\{\mathbf{x}_z\}_{z \in x}$ and $\{\mathbf{x}_z\}_{z \in y}$ is a geometric invariant. Computing the volume of a convex hull is quite an expensive procedure, thus it is not always suitable to be performed at each training step. However, it can still be used if the cells have small cardinality, or if ETNNs are used in an invariant fashion, i.e., without the update in (7), because the volume needs to be computed only once before training. Finally, the volume of convex hulls reduces to the volume of simplices in case the considered CC is a (geometric) simplicial complex, generalizing what has been done in (Eijkelboom et al., 2023).

*Remark 1 .* It is noteworthy that if higher rank cells, not only nodes, come with attached geometric features, they can be seamlessly integrated into ETNNs. In that case, there would be geometric invariants of node geometric features, geometric invariants of higher-rank geometric features, and, possibly, geometric invariants of both nodes and higher rank cells. The equivariant update in (7) would be straightforwardly modified to include all the cells with geometric features in the sum.

**Expressiveness of ETNN.** The expressive power of neural networks over CTS is proportional to their ability to distinguish non-isomorphic CTSs. As the renowned Weisfeiler-Leman (WL) test for GNNs (Xu et al., 2019), several generalized WL tests have been developed for CTSs (e.g. simplicial (Bodnar et al., 2021b), cell (Bodnar et al., 2021a), or path (Truong & Chin, 2024) complexes) and have been shown to be upper bounds on the expressiveness of the corresponding TDL models. These tools also demonstrated that TDL models are usually more expressive than vanilla GNNs when it comes to distinguishing isomorphic graphs. For geometric graphs, the Geometric-WL (GWL) test (Joshi et al., 2023) has been recently introduced as an upper bound on the expressiveness of invariant/equivariant geometric GNNs. In Appendix F, we introduce the notion of *Geometric Augmented Hasse Graph* and we combine it with the GWL to show that ETNNs are, in general, more expressive than scalarization-based graph methods. We also perform experiments to validate our claim, including testing ETNN on the counterexample structures from (Pozdnyakov et al., 2020; Joshi et al., 2023). Next, we present the statements of our results. Proofs are in Appendix F.

*Proposition 2 . (Informal)* An ETNN is at least as powerful as an EGNN (Satorras et al., 2021) in distinguishing $k$-hop distinct graphs. In most of the cases, an ETNN is strictly more powerful than an EGNN. The quantitative gap between ETNNs and EGNNs depends on how the graphs are lifted into a CC, i.e., how the set of cells $\mathcal{X}$ and the collection of neighborhood functions $\mathcal{CN}$ are defined.

*Proposition 3 .* There exist a pair of CCs whose nodes come with geometric features, and a collection of neighborhoods such that the CCs are undistinguishable by ETNN.

*Proofs .* See Appendix F.

**Computational complexity of ETNN.** In Appendix G, we present an analysis of the computational complexity and runtime of ETNNs. In particular, we show that, if the connectivity of the CC is sufficiently sparse, which is usually the case, then the overall complexity is linear in the number of cells. In the worst case, all the cells all connected among them, resulting in a quadratic cost in the number of cells. In the same Appendix G, we also empirically validate these results by comparing the runtimes of ETNN with EGNN (Satorras et al., 2021) and EMPSN (Eijkelboom et al., 2023).

## 4 REAL-WORLD COMBINATORIAL COMPLEXES

Designing equivariant models for CCs tailored for specific applications is an exciting direction since CCs can accommodate hierarchical higher-order features that the literature's most used CTSs cannot readily incorporate, including graphs, simplicial complexes, path complexes, cell complexes, and hypergraphs[1]. This section presents two vastly different applications. The first application considers molecular data. While this data has been investigated under the lens of CTS, our approach allows us to naturally integrate their higher-order structures such as rings and functional groups. Next, we present a novel TDL framework for geospatial data. We show that the interactions between geographical objects in geospatial data such as polylines and polygons can be modeled via CCs. In the next section, we will use instances of these CCs in experiments evaluating the ETNN architecture.

**Molecular combinatorial complexes.** We introduce the notion of *Molecular Combinatorial Complexes*, showing that they can jointly retain all the properties of graphs, cell complexes, and hyper-

---

[1]Appendix C provides an overview of the aforementioned CTSs.

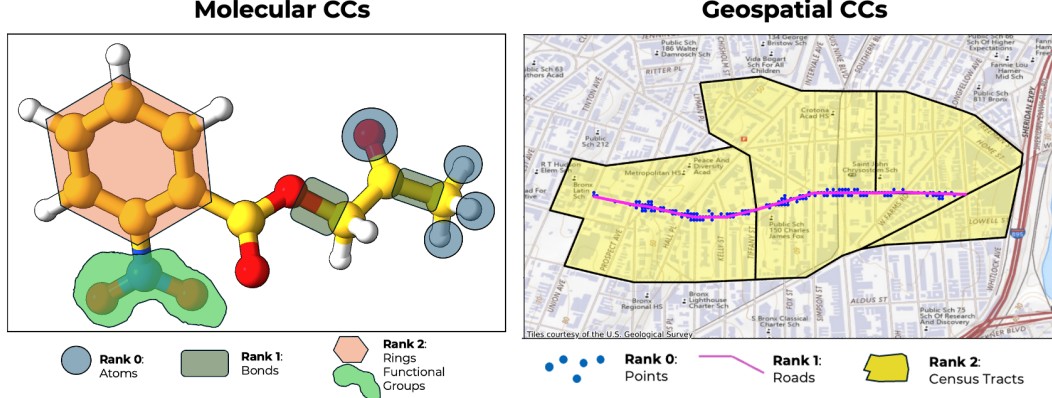

**Figure 2: Real-World Combinatorial Complexes.** *(Left)*: A molecular CC where 0-cells are atoms, 1-cells are bonds, and 2-cell represent rings and functional groups. *(Right)*: A geospatial CC where 0-cells are grid points, 1-cells are road polylines, and 2-cells are census tract polygons. For visibility, the geospatial CC only shows one road (rank 1) and its incident lower and higher rank cells.

graphs, summarized in Appendix H, thus overcoming their mutual limitations. In a molecular CC, the set of nodes $\mathcal{S}$ is the set of atoms, while the set of cells $\mathcal{X}$ and the rank function rk are such that: atoms are 0-cells, bonds and functional groups made of two atoms are 1-cells, and rings and functional groups made of more than two atoms are 2-cells (see Figure 2).

**Geospatial combinatorial complexes.** *Geospatial CCs* (GCCs) are CCs in which all cells are subsets of a geospatial domain. For example, cells of a GCC can consist of polygons, polylines (sequences of connected lines), and points[2], all which are elementary geometric objects used in geospatial information systems (Ayers, 1992). Fig. 2. illustrates this scenario, where 0-cells are points where air quality monitoring units are located (e.g., in a city), 1-cells are traffic roads as polylines, and 2-cells are census tracts represented as polygons, geographic areas with homogeneous demographic characteristics. Formally, GCCs are characterized by a spatial representation and a spatially-determined neighborhood system. Let $\mathcal{T} \subset \mathbb{R}^n$ represent a geospatial domain. A spatial representation is an embedding $s : \mathcal{X} \to \mathcal{P}(\mathcal{T})$ mapping each cell $x$ to a subset $s(x) \subset \mathcal{T}$. Although any neighborhoood system can be used with GCCs (such as up/down incidences), it is more natural to construct the neighborhood system of a GCC from the spatial representation $s(x)$, namely:

$$\mathcal{N}_A(x) = \{y \in \mathcal{X} \mid \mathrm{rk}(y) = \mathrm{rk}(x), s(x) \cap s(y) \neq \emptyset\}$$
$$\mathcal{N}_{I,\uparrow}(x) = \{y \in \mathcal{X} \mid \mathrm{rk}(y) = \mathrm{rk}(x) + 1, s(x) \cap s(y) \neq \emptyset\}$$
$$\mathcal{N}_{I,\downarrow}(x) = \{y \in \mathcal{X} \mid \mathrm{rk}(y) = \mathrm{rk}(x) - 1, s(x) \cap s(y) \neq \emptyset\}. \tag{12}$$

For instance, $\mathcal{N}_A(x)$ can represent adjacencies between geographic spaces that share a portion or a boundary, while the up and down incidences $\mathcal{N}_{I,\uparrow}(x)$ and $\mathcal{N}_{I,\downarrow}(x)$ can capture the notion of road crossing a county, or two geographic areas of different rank overlapping. Evidently, more than one neighborhood function can be used or combined with the standard choices in (1) -(2) according to the application. It is not always necessary to use points and polylines. One can also consider geographic divisions (polygons) with multiple resolution levels, each mapping to a different rank. For example, in the United States, census tracts are smaller than zip codes, which are smaller than counties, and so on. A census tract can intersect multiple zip codes, and zip codes can belong to multiple counties, partitioning space irregularly. As an example, Fig. 2 portrays a road polyline traversing multiple census tract polygons. Unlike classical multi-resolution spatial modeling (Kolaczyk & Huang, 2001; Hengl et al., 2021), we don't require regular structures such as grids or nested partitions. More importantly, we allow each resolution to have unique features. This point illustrates a crucial difference and advantage over graph modeling, which requires all data to be pre-aggregated at the same rank (typically the 0-cells), see Appendix H. This work is the first to explore combinatorial topological modeling of multi-resolution irregular geospatial data, which are the backbone of geographic information systems. The dataset and hyper-local air pollution

---

[2]Represented as singleton sets.

downscaling task that we will present in the next section will not only propose a novel technique for learning from irregular multi-resolution geospatial data, but can be used as a *benchmark* for TDL.

## 5 EXPERIMENTS

**Molecular property prediction.** We evaluate our model on molecular property prediction using the QM9 dataset (Ramakrishnan et al., 2014), a comprehensive collection of quantum chemical calculations for small organic molecules. It includes geometric, energetic, electronic, and thermodynamic properties for 134k stable molecules with up to nine heavy atoms (C, O, N, F). See Appendix I.1.1 for a summary.

We consider several different variants of ETNNs (see Appendix J), but the most general molecular combinatorial complexes (CCs) that we design is the one as in Fig. 2. It employs three ranks to capture different levels of molecular interactions: i) atoms as 0-cells, ii) bonds, i.e., pairs of atoms, as 1-cells, and iii) rings and functional groups of size greater than two, i.e., at least triplets of atoms, as 2-cells. Similar to methodologies for global connections (e.g. EGNN(Satorras et al., 2021) and VN-EGNN (Sestak et al., 2024)), we generalize the notion of virtual node by incorporating a single cell, that we refer to as *virtual cell*, containing the whole set of 0-cells (the atoms), and having maximum rank, e.g. in the most general case it would have rank 3. As neighborhood functions, we employed the up/down adjacencies and incidences from (1)-(2), and a "max" adjacency defined as

$$\mathcal{N}_{A,\max}(x) = \{y \in \mathcal{X} | \mathrm{rk}(y) = \mathrm{rk}(x), \exists z \in \mathcal{X} : \mathrm{rk}(z) = \max_{z \in \mathcal{X}} \mathrm{rk}(z), x \subset z, \text{ and } y \subset z\} \quad (13)$$

The virtual cell together with the max adjacency in (13) acts as a hub across all cells of the same rank, connecting each pair, following a common practice (Satorras et al., 2021; Eijkelboom et al., 2023). However, the max adjacency is an object of independent interest, even when the virtual cell is not in the complex. In that case, it would allow cells of lower ranks to communicate based on the highest order cells they are part of, e.g. atoms belonging to the same ring or functional group. In this setting, it is clear how ETNN naturally handles heterogeneity, e.g., the same pair of bonds could be connected because part of the same functional group (up adjacency) and the virtual cell (max adjacency), but the two messages will be different across the neighborhoods. Similarly, ETNN allows us to equip each rank with unique features without any ad-hoc aggregations at the atom or bond levels. We consider four distinct types of non-geometric feature vectors: atom features, bond features, ring features, and functional group features. We provide a full list of the non-geometric feature vectors we used in Appendix I.1.2. As geometric features, we use the atoms' positions. The complete list of the model parameters is in Appendix I.1.3. A description of all the configurations and a detailed reproducibility discussion is found in Appendix J. In all the experiments, following (Satorras et al., 2021; Eijkelboom et al., 2023), we use the invariant version of ETNN.

In Table 1, we present the results for the first 6 molecular properties, from which it is clear that ETNN significantly outperforms EGNN (Satorras et al., 2021). Details and full results for all the properties are described in Appendix J. ETNN generally improves the prediction performance of EGNN, with notable improvements on $\alpha$ and $\mu$, achieving state-of-the-art performance for $\mu$ despite being a general framework not tailored for chemical tasks. To ensure further fairness in the comparison, the table shows the results of a 1.5M EGNN obtained through our framework and codebase (EGNN-graph-W), as explained in Appendix J. Comparing this with the original EGNN, we observe performance improvements. Finally, the low prediction performance of ETNN-w/o VC (ETNN without the virtual cell) highlights the significant impact of the virtual cell. The reproduced EMPSN* (Eijkelboom et al., 2023) is obtained by running ETNN on a combinatorial complex being the same exact Vietoris-Rips simplicial complex of the original paper, similarly using the up/down adjacencies and incidences plus the max adjacency, and with architectural hyperparameters as similar as possible to the original architecture. See Appendix J for a detailed discussion. As the reader can see, ETNN generally matches or outperform the reproduced EMPSNs. However, ETNN employs *almost two orders of magnitude* less cells. In particular, the original VR complex of dimension 2 used in EMPSN has 555.25 cells per molecule on average (18 nodes, 157.9 edges, 379.36 triangles) and it is also expensive to compute in the preprocessing step, while our molecular CC (in the worst case) has 39.8 cells per molecule on average (18 nodes, 19 edges, 1.8 rings, and 1.3 functional groups) and it requires slightly more than a lookup table to be computed. As a consequence, ETNNs have half of the memory usage and almost half of the runtime per epoch w.r.t. EMPSNs, as we describe in Appendix G. This fact ultimately showcases the effectiveness of the principled inductive bias given by the molecular CC.

Table 1: Mean Absolute Error for the molecular property prediction benchmark in QM9 dataset. ETNN-* corresponds to the configurations, among the ones we tested, that produced the best results for each property. ETNN-single-* corresponds to the configuration, among the ones we tested, that produced the average best results across properties. ETNN-graph-W and EMPSN* are EGNN (Satorras et al., 2021) and EMPSN (Eijkelboom et al., 2023) derived as instances of ETNNs through our codebase. All the models except ETNN-w/o VC use the virtual cell. We report the improvement of ETNN-* over the original EGNN. TopNet$^\dagger$ is the best performing variant of TopNets (Verma et al., 2024). The other baselines are from (Satorras et al., 2021). Details and full results in Appendix J.

| Task | $\alpha$ | $\Delta\varepsilon$ | $\varepsilon_{\mathrm{HOMO}}$ | $\varepsilon_{\mathrm{LUMO}}$ | $\mu$ | $C_\nu$ |
|---|---|---|---|---|---|---|
| Units ($\downarrow$) | bohr$^3$ | meV | meV | meV | D | cal/mol K |
| NMP | .092 | 69 | 43 | 38 | .030 | .040 |
| Schnet | .235 | 63 | 41 | 34 | .033 | .033 |
| Cormorant | .085 | 61 | 34 | 38 | .038 | .026 |
| L1Net | .088 | 68 | 46 | 35 | .043 | .031 |
| LieConv | .084 | 49 | 30 | 25 | .032 | .038 |
| DimeNet++ | **.044** | 33 | 25 | 20 | .030 | .023 |
| TFN | .223 | 58 | 40 | 38 | .064 | .101 |
| SE(3)-Tr. | .142 | 53 | 35 | 33 | .051 | .054 |
| Equiformer (SotA) | .046 | **30** | **15** | **14** | **.011** | **.023** |
| TopNet$^\dagger$ | .083 | 47 | 37 | 24 | .035 | .032 |
| EGNN | .071 | 48 | 29 | 25 | .029 | .031 |
| **ETNN-*** | .062 | 45 | 26 | 22 | .022 | .030 |
| ETNN-graph-W | .067 | 46 | 27 | 25 | .030 | .036 |
| ETNN-w/o VC | .161 | 73 | 49 | 48 | .306 | .051 |
| EMPSN* | .061 | 42 | 29 | 25 | .030 | .028 |
| ETNN-single-* | .069 | 46 | 26 | 23 | .025 | .033 |
| Improvement over EGNN | **-13%** | **-6%** | **-10%** | **-12%** | **-26%** | **-3%** |

**Hyperlocal air pollution downscaling.** Next, we introduce a novel benchmark for TDL based on the geospatial framework introduced in Section 4. We first provide a data an overview of the data and tasks for completeness.

The task consists of predicting PM$_{2.5}$ air pollution at a high resolution of $0.0002° \approx 22m$. The prediction targets of PM$_{2.5}$ measurements are obtained from a publicly available dataset (Wang et al., 2023) consisting of measurements by mobile air sensors installed on cars, corresponding to the last quarter of 2021 in the Bronx, NY, USA. The targets are aggregated at the desired spatial resolution. The resulting Geospatial CC consists of 3,946 point measurement units (0-cells), 550 roads (1-cells), and 151 census tracts (2-cell). Fig. 3 provides an overview of the features for each rank, including demographics, land-use information, daily traffic rates and

Table 2: Results for the *air pollution downscaling* task. AEV is the average $R^2$ (explained variance) over 30 seeds, DEV is the difference in $R^2$, and MSE the Mean Squared Error.

| Baseline | AEV($R^2$) | Std. Err ($R^2$) | MSE |
|---|---|---|---|
| Linear | 0.51% | 0.95% | 1.106 |
| MLP | 2.35% | 1.61% | 1.022 |
| GNN | 2.44% | 0.99% | 0.987 |
| EGNN | 1.43% | 1.2% | 1.041 |
| ETNN | 9.34% | 2.05% | 0.935 |

(a) Baseline model comparison

| Baseline | DEV ($R^2$) | Std. Err ($R^2$) | MSE |
|---|---|---|---|
| no virtual node | -1.80% | 2.32% | 0.957 |
| no position update: invariant ETNN | -1.37% | 2.52% | 0.956 |
| no geometric features: CCMPN | -1.08% | 2.05% | 0.946 |

(b) Ablation study results

traffic composition per road, and distance to schools and other points of interest. The most important predictor for the downscaling task is the coarse-level annual PM 2.5 from the previous year (Di et al., 2019), available at a coarser 1 km resolution which is projected to the roughly equivalent census-tract area level. In total, we include 5 point-level features, 2 road-level features, and 17 tract-level features. The full list of features and their data sources is in Appendix I.2.2. All the pre-processing code is included in our code repository.

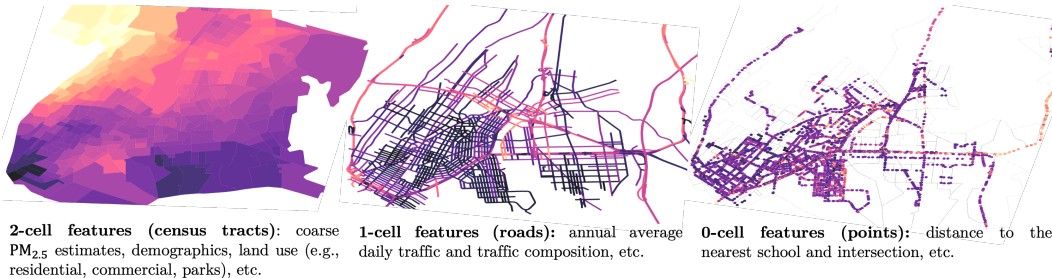

**2-cell features (census tracts)**: coarse $PM_{2.5}$ estimates, demographics, land use (e.g., residential, commercial, parks), etc.

**1-cell features (roads)**: annual average daily traffic and traffic composition, etc.

**0-cell features (points)**: distance to the nearest school and intersection, etc.

Figure 3: Overview of features used in the *air pollution downscaling* benchmark. The task is to predict the local $PM_{2.5}$ measured at the 0-cells, which have an approximate resolution of $22 \times 22$ meters. As such, this is a node regression task.

We create a geospatial CC using the neighborhood functions described by Eq. (12). For the geometric features, we use the coordinates of the 0-cells with Mercator projection. This projection has the key property that Euclidean distance in coordinate space approximates geodesic distance in meters. As such, $E(n)$-equivariance is implicitly essential to the purpose of the Mercator projection. The exact coordinate values are irrelevant: What is most important for our application is its usefulness in expressing geodesic distances. By using these geometric features in GCCs, we allow the model to use the coordinates but only modulo Euclidean transformations, which is the original intention of the Mercator projection. In other words, the model can still capture location-specific effects in node-level tasks without using the coordinates directly, which would lead to poor inductive biases.

Since only CCs can support multi-level features, we create point-level features for other baselines (MLP, GNN, and EGNN) by concatenating the features of each point with the features of the 1-cell and 2-cell that the point belongs to. For the base ETNN, we again employ a virtual cell as defined in the previous paragraph. Since the task is at the node level, the base implementation uses the equivariant update from 1. We will perform ablation experiments over these defaults. Appendix I.2.3 presents in detail the hyper-parameter and architecture configurations used for the experiments

The experiment results are shown in Table 2a. The results are averaged over 30 seeds. Standard errors are computed with the formula $\sigma/\sqrt{n}$, which is recommended when having at least sample size 30. The table indicates that ETNN outperforms EGNN, GNN, and MLP, emphasizing the benefits of multi-level modeling combined with $E(n)$ equivariance. We additionally conduct ablations studies on ETNN, presented in Table 2 (b). The results show that, as expected, not including geometric features decreases performance. Perhaps surprisingly, the ablation of ETNN using only the geospatial CC structure without geometric features (invariants) performed slightly better than using the features but not using the position update in (7), i.e., using the invariant version of ETNN. Nonetheless, the largest decrease came from not using the virtual cell.

## 6 CONCLUSION

We introduced E(n) Equivariant Topological Neural Networks (ETNNs), a framework for designing scalarization-based equivariant message passing networks on combinatorial complexes. Combinatorial complexes flexibly represent arbitrary hierarchical higher-order interactions, thus ETNNs are a legitimate proxy for designing equivariant models on a wide class of combinatorial topological spaces, such as simplicial or cell complexes, or, more in general, over combinatorial objects used to model interactions among entities, such as hierarchical graphs. We showed that ETNNs are at least as expressive as existing scalarization-based equivariant architectures for geometric graphs. Experiments on the QM9 molecular benchmark demonstrated ETNNs' effectiveness for richly structured molecular data. Moreover, we introduced a novel geographic regression task, highlighting ETNNs' ability and showing for the first time how combinatorial complexes can be used to model irregular multi-resolution geospatial data. We discuss the limitations and future directions in Appendix A. ETNNs represent, overall, a unifying framework showcasing the benefits of a principled synergy between combinatorial and geometric equivariances over beyond-graph domains. We believe that our methodological contribution, together with the easy-to-use and solid codebase we provide, can mark a step in boosting and standardizing research for scalarization-based equivariant TDL.

REPRODUCIBILITY STATEMENT

We include all the details about our experimental setting, e.g. the choice of hyperparameters and the specifications of our machine, in Appendix I. The code, data splits, and virtual environment needed to replicate the experiments and easily use the ETNN framework are provided at the following repository: `https://github.com/NSAPH-Projects/topological-equivariant-networks`. An in-depth discussion about the reproducibility of the molecular property prediction task is in Appendix J.

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

## Appendix Contents

## A  LIMITATIONS & FUTURE DIRECTIONS

To our knowledge, ETNN is the first unifying framework for scalarization-based $E(n)$ equivariant networks over combinatorial topological spaces or, more in general, over combinatorial objects used to model interactions among entities. In this work, we introduced ETNNs as a significantly more general, expressive, and flexible framework compared to prior geometric/topological deep learning models operating on geometric graphs (Satorras et al., 2021) or simplicial complexes (Eijkelboom et al., 2023). The promising results open several avenues.

**Methodological.** First, while we introduced several useful invariants like pairwise distances, centroids, Hausdorff distances, and convex hull volumes, there are likely many other meaningful geometric quantities that could be leveraged, especially for specific application domains. Therefore, developing a more comprehensive set of geometric invariants, as well as learning methods to discover them from data automatically, is an important future direction. In this sense, generalizing the approach from (Liu et al., 2024) of integrating Clifford group-equivariant layers with message passing layers. i.e., going beyond scalarization, is a possible solution, although requiring to solve possible scalability issues. Second, extending our framework to make it scale invariant could be useful in geospatial tasks. Third, at this stage ETNNs cannot directly handle time-varying scenarios, thus we aim to go beyond the static settings and develop dynamic or temporal variants of ETNNs. This could enable new applications in areas like spatio-temporal forecasting, and physical simulations. Fourth, and finally, ETNNs assume that (potentially) each rank should satisfy the same $E(n)$ equivariance, but another interesting direction is studying the setting in which each rank is required to satisfy different symmetries, pointing towards a principled use of products of groups.

**Computational.** While our experimental validation of ETNNs covered both supervised inductive and semisupervised transductive tasks, future works could tailor ETNNs to fit challenging specific applications at scale. In this sense, unsupervised or self-supervised training methods for ETNNs could enable the learning of meaningful representations in a data-driven manner without requiring labeled data for every task. This could allow ETNNs to better leverage large, unlabeled geometric datasets. Moreover, though computationally tractable, ETNNs may benefit from further improvements to effectively scale on very large datasets. The main bottlenecks are MP operations and geometric invariants computation. A possible solution could be neighbor samplers (Zhang et al., 2019), or

developing $E(n)$ equivariant message passing-free ETNNs, building on works like (Maggs et al., 2024).

## B  RELATED WORKS

**Topological Deep Learning.** TDL is informed by foundational work in Topological Signal Processing (TSP) (Barbarossa & Sardellitti, 2020; Schaub et al., 2021; Roddenberry et al., 2022; Sardellitti et al., 2021), highlighting the value of analyzing multi-way relationships in data. The works in (Bodnar et al., 2021b;a) extended the Weisfeiler-Lehman graph isomorphism test to simplicial and regular cell complexes (Bodnar et al., 2021a), respectively, showing that message passing over these spaces are more expressive than message passing over graphs. Convolutional (Ebli et al., 2020; Yang et al., 2022; Hajij et al., 2020; Yang & Isufi, 2023; Roddenberry et al., 2021; Hajij et al., 2022a) and attentional architectures (Battiloro et al., 2023c; Giusti et al., 2022; Goh et al., 2022; Giusti et al., 2023a) over simplicial and cell complexes have been introduced as well. Additionally, the application of message passing or diffusion on cellular sheaves (Hansen & Ghrist, 2019) over graphs (Hansen & Gebhart, 2020; Bodnar et al., 2022; Battiloro et al., 2023a; 2024b; Barbero et al., 2022) has proven effective in heterophilic scenarios. Models without message passing have been introduced for simplicial complexes (Madhu et al.; Gurugubelli & Chepuri, 2024; Maggs et al., 2024) and hypergraphs (Tang et al., 2024). An architecture for inferring a latent regular cell complex to improve a downstream task has been introduced in (Battiloro et al., 2024a). Gaussian Processes over cell complexes have been introduced in (Alain et al., 2023). Message passing networks over path complexes have been introduced in (Truong & Chin, 2024; Li et al., 2024a). An extensive review of TDL is available in (Papillon et al., 2023). A comprehensive framework for TDL was outlined in (Hajij et al., 2022b), proposing combinatorial complexes and message-passing networks (CCMPNs) over them. A general topological framework for expressivity analysis of CCMPNs has been proposed in (Eitan et al., 2024). Finally, software for neural networks on combinatorial topological spaces has been presented in (Hajij et al., 2024; Telyatnikov et al., 2024).

**Equivariant Neural Networks** The effectiveness of Convolutional Neural Networks (He et al., 2016), and their first group extension to the SO(2) group (Cohen & Welling, 2016) initially demonstrated the advantages of equivariant models, and paved the way for a series of works exploring how to design models respecting some symmetries (Weiler et al., 2023). Loosely speaking, some of these works employ natively equivariant function spaces (Thomas et al., 2018; Fuchs et al., 2020), or port the spatial space into high-dimensional spaces (Cohen et al., 2019; Finzi et al., 2020; Hutchinson et al., 2021; Batatia et al., 2022). An alternative strategy is designing neural networks (Köhler et al., 2019; Schütt et al., 2017; Brandstetter et al., 2021; Batzner et al., 2022) that perform equivariant operations directly within the original space, including the (VN-)EGNN from (Satorras et al., 2021; Sestak et al., 2024). A characterization theorem for equivariant networks has been presented in (Pacini et al., 2024). Recent works showed how to achieve equivariance by leveraging neural network architectures operating on the Clifford algebra (Ruhe et al., 2024; Brehmer et al., 2024). Each of these approaches has limitations and advantages, and the effectiveness of a specific design choice is related to the considered application and computational constraints (Han et al., 2022).

Our paper builds on both classes of works. To the best of our knowledge, the most relevant works merging TDL models and $E(n)$ equivariant neural networks are (Eijkelboom et al., 2023; Liu et al., 2024; Verma et al., 2024). In particular, (Eijkelboom et al., 2023) generalizes the approach from (Satorras et al., 2021) and introduces $E(n)$ Equivariant Simplicial Message passing Networks (EMPSNs). The framework in (Verma et al., 2024) injects information about persistent homology (PH) on top of (equivariant) simplicial message passing networks, thus interestingly combining TDL and Topological Data Analysis. Finally, (Liu et al., 2024) adopts the approach from (Ruhe et al., 2024) by combining Clifford group equivariant layers with simplicial message passing networks. However, no other combinatorial topological spaces are considered, and, overall, a general framework for designing equivariant neural networks over combinatorial topological spaces is missing. Here we address this important gap by introducing our ETNNs.

## C  COMBINATORIAL TOPOLOGICAL SPACES AS COMBINATORIAL COMPLEXES

In this appendix, we revisit a subclass of mathematical objects (not always formal topological spaces) that can be described combinatorially, message passing networks operating on them, and how they

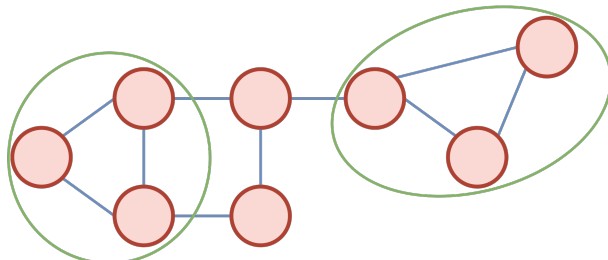

Figure 4: A simplicial complex of order 2, comprising of nodes (cells of rank 0), edges (cells of rank 1), and triangles (cells of rank 2).

can be cast in the ETNN frameworks. As we did for CCs, we will often overload the notation $\mathcal{X}$ in the following subsections, incurring some notation abuses for the sake of exposition and consistency.

## C.1 GRAPHS

The first combinatorial objects we describe are obviously graphs. They are remarkably useful in several fields, across several tasks, thanks to their ability to encode prior data knowledge, i.e., their networked structure.

**Graphs.** A graph $\mathcal{X}$ is a pair $(\mathcal{S}, \mathcal{E})$, where $\mathcal{S}$ is the set of nodes, and $\mathcal{E}$ is the set of edges (i.e., pairs of nodes).

**Neighbhorhood Functions in a Graph.** Given a graph $\mathcal{X}$, for all $x \in \mathcal{S}$, the up adjacency $\mathcal{N}_{A,\uparrow}$ (i.e., the usual node adjacency) is defined as

$$\mathcal{N}_{A,\uparrow}(x) = \{y \in \mathcal{S} | \exists z \in \mathcal{E} : x \subset z \text{ and } y \subset z\}. \tag{14}$$

**Graphs as Combinatorial Complexes.** Graphs can be easily cast as combinatorial complexes. Let us denote the set of singletons $\widetilde{\mathcal{S}} = \{\{s\}\}_{s \in \mathcal{S}}$, a graph $\mathcal{X}$ is a combinatorial complex $(\mathcal{S}, \mathcal{X}, \mathrm{rk})$, where $\mathcal{S}$ is the set of nodes, the set of cells $\mathcal{X} = \widetilde{\mathcal{S}}$ is the set of singletons, and the rank function $\mathrm{rk}$ assigns rank 0 to each singleton. Please notice that there is no need to include edges as cells in the complex, differently from, e.g., simplicial complexes, as detailed in the next subsection, because a CCMPN as in (3) with $\mathcal{CN} = \{\mathcal{N}_{A,\uparrow}\}$ recovers the standard Graph Message Passing Network from (Gilmer et al., 2017b). An ETNN as in (6) with the same collection of neighborhoods $\mathcal{CN}$ recovers the $E(n)$ Equivariant Graph Neural Network (EGNN) from (Satorras et al., 2021).

## C.2 SIMPLICIAL COMPLEXES

Graphs are capable of explicitly managing only pairwise interactions among nodes. However, in numerous applications (Giusti et al., 2016; Kanari et al., 2018; Patania et al., 2017; Sardellitti et al., 2021), including the ones presented in this work, higher-order multiway interactions need to be taken into account. Simplicial complexes are one of the most intuitive objects to start incorporating them.

**Simplicial Complexes.** Given a finite set of nodes $\mathcal{S}$, a *k-simplex* $x$ is a subset of $\mathcal{S}$ with cardinality $k + 1$. A *face* of $x$ is a $k - 1$-simplex being a subset of it, thus a $k$-simplex has $k + 1$ faces. A *coface* of $x$ is a $(k + 1)$-simplex being a superset of it (Barbarossa & Sardellitti, 2020; Lim, 2020). A simplicial complex (SC) $\mathcal{X}$ of order $K$ is a collection of $k$-simplices $x$, $k = 0, \ldots, K$ such that, if a simplex $x$ belongs to $\mathcal{X}$, then all its subsets $y \subset x$ also belong to the complex (inclusivity property). In most of the cases, the focus is on complexes of order up to two, thus having a set of nodes, a set of edges, and a set of triangles. To give a simple example, in Fig. 4 we sketch a simplicial complex of order 2.

**Neighbhorhood Functions in a Simplicial Complex.** Given a SC $\mathcal{X}$, the *coboundary* and *boundary* neighborhood functions $\mathcal{N}_C$ and $\mathcal{N}_B$, respectively, are defined, as

$$\mathcal{N}_C(x) = \{y \in \mathcal{X} | y \text{ is a coface of } x\}, \quad \mathcal{N}_B(x) = \{y \in \mathcal{X} | y \text{ is a face of } x\}. \tag{15}$$

The up and down adjacencies $\mathcal{N}_{A,\uparrow}$ and $\mathcal{N}_{A,\downarrow}$, respectively, are defined, as

$$\mathcal{N}_{A,\uparrow}(x) = \{y \in \mathcal{X} | y \text{ shares a coface with } x\}, \mathcal{N}_{A,\downarrow}(x) = \{y \in \mathcal{X} | y \text{ shares a face with } x\}. \tag{16}$$

**Simplicial Complexes as Combinatorial Complexes.** It is easy to frame SCs in the CC framework. In particular, a simplicial complex $\mathcal{X}$ is a combinatorial complex $(\mathcal{S}, \mathcal{X}, \mathrm{rk})$, where $\mathcal{S}$ is the set of nodes, the set of cells $\mathcal{X}$ is the set of simplices, and the rank function $\mathrm{rk}$ assigns to each cell its cardinality minus one. A CCMPN as in (3) with $\mathcal{CN} = \{\mathcal{N}_C, \mathcal{N}_B, \mathcal{N}_{A,\uparrow}, \mathcal{N}_{A,\downarrow}\}$ recovers the Message Passing Simplicial Network from (Bodnar et al., 2021b). An ETNN as in (6) with the same collection of neighborhoods $\mathcal{CN}$ recovers the $E(n)$ Equivariant Message Passing Simplicial Network (EMPSN) from (Eijkelboom et al., 2023).

### C.3  CELL COMPLEXES

Simplicial complexes are capable of managing relationships of varying degrees, yet the inclusion property often binds them. This property mandates that if a set is part of the space, then all its respective subsets must also be included in that space. In numerous applications, this restriction is not only unnecessary but also lacks a justified basis. Consequently, cell complexes are frequently employed instead (Mulder & Bianconi, 2018; Hirani et al., 2010). These structures are versatile and support a hierarchical organization.

**Regular Cell Complexes.** A *regular cell complex* (or regular CW complex) is a topological space $\mathcal{X}$ together with a partition $\{\mathcal{X}_x\}_{x \in \mathcal{H}_{\mathcal{X}}}$ of subspaces $\mathcal{X}_x$ of $\mathcal{X}$ called cells, where $\mathcal{H}_{\mathcal{X}}$ is the indexing set of $\mathcal{X}$, such that

1. For each $x \in \mathcal{X}$, every sufficient small neighborhood of $x$ intersects finitely many $\mathcal{X}_x$;
2. For all $x,y$ we have that $\mathcal{X}_x \cap \overline{\mathcal{X}}_y \neq \varnothing$ iff $\mathcal{X}_x \subseteq \overline{\mathcal{X}}_y$, where $\overline{\mathcal{X}}_y$ is the closure of the cell;
3. Every $\mathcal{X}_x$ is homeomorphic to $\mathbb{R}^k$ for some $k$;
4. For every $x \in \mathcal{H}_{\mathcal{X}}$ there is a homeomorphism $g$ of a closed ball in $\mathbb{R}^k$ to $\overline{\mathcal{X}}_x$ such that the restriction of $g$ to the interior of the ball is a homeomorphism onto $\mathcal{X}_x$.

From 2, $\mathcal{H}_{\mathcal{X}}$ has a poset structure, given by $y \leq x$ iff $\mathcal{X}_y \subseteq \overline{\mathcal{X}_x}$, and we say that $y$ *bounds* $x$. This is known as the *face poset* of $\mathcal{X}$. From 4, all of the topological information about $\mathcal{X}$ is encoded in the poset structure of $\mathcal{H}_{\mathcal{X}}$. Then, a regular cell complex can be identified with its face poset. From now on we will indicate the cell $\mathcal{X}_x$ with its corresponding face poset element $x$. The dimension or order $\dim(\sigma)$ of a cell $\sigma$ is $k$, and we call it a $k-$cell. The dimension or order of a regular cell complex is the largest dimension of any of its cells. In the following, we will refer to regular cell complexes simply as cell complexes or CW complexes. Cell complexes can be described via an incidence relation (boundary relation) with a reflexive and transitive closure that is consistent with the partial order introduced in the above definition. In particular, we have the boundary relation $y \prec x$ iff $\dim(y) \leq \dim(x)$ and there is no cell $z$ such that $y \leq z \leq x$. In other words, the boundary of a cell $x$ of dimension $k$ is the set of all cells of dimension less than $k$ bounding $x$. The *k-skeleton* of a regular cell complex $\mathcal{X}$ is the subcomplex of $\mathcal{X}$ consisting of cells of dimension at most $k$. Usually, CW complexes $\mathcal{X}$ are constructed through a hierarchical gluing procedure starting from a set of nodes $\mathcal{S}$ (Hansen & Ghrist, 2019; Bodnar et al., 2021a). Therefore, we can interpret the 0-skeleton of a cell complex $\mathcal{X}$ as the set $\mathcal{S}$ of 0-cells (nodes) and the 1-skeleton as the set of nodes together with the set of 1-cells (edges), thus a graph. For this reason, given a graph $\mathcal{G}$, i.e., a realization of an order-1 cell complex, it is possible to lift it to an order-2 cell complex $\mathcal{X}$ whose 1-skeleton is isomorphic to $\mathcal{G}$. A way to do it is by attaching order 2 cells to the/a set/subset of simple/induced cycles of $\mathcal{G}$. Cells of order greater than 2 can be attached as well using the same gluing procedure mentioned above (Sardellitti et al., 2021; Grady & Polimeni, 2010), but there is usually little interest in them. In Fig. 5, we sketch an order 2 CW complex.

***Remark 2 .*** It is easy to notice that a CW complex considering cycles of length up to 3 is equivalent to a simplicial complex of order 2 having the same set $\mathcal{S}$ of 0-simplices (nodes), the same set of 1-simplices (edges), and the same set of 2-simplices (triangles). In general, simplicial complexes are special cases of CW complexes for which any $k$-cell is composed of exactly $k + 1$ nodes.

***Remark 3 .*** The reader may have noticed that we gave no explicit characterization of a simplicial complex as a topological space, although it is pretty easy to particularize cell to simplicial complexes. This is because our definition of SC in Sec. C.2 is, to be completely rigorous, the definition of an *abstract* simplicial complex, that itself is not a topological space. However, when we refer to a simplicial complex as a topological space, we are implicitly talking about the *geometric realization* of an abstract simplicial complex. Therefore, we are implicitly stating that the SC has a topology

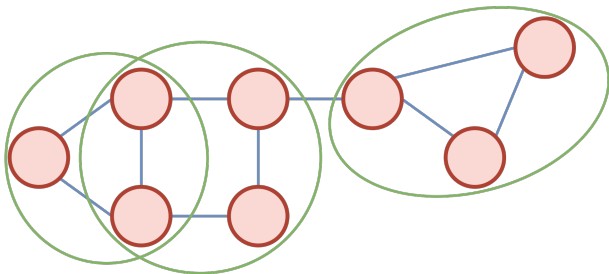

Figure 5: A CW complex of order 2, comprising of nodes (cells of rank 0), edges (cells of rank 1), and cycles (cells of rank 2).

coherent with its simplices, which are "topologized" as homeomorphisms of the standard simplices living in Euclidean space. Please refer to (Barbarossa & Sardellitti, 2020) for a detailed but accessible description. We will not incur any ambiguity by referring to abstract simplicial complexes as just simplicial complexes.

**Neighbhorhood Functions in a Cell Complex.** Given a CW complex $\mathcal{X}$, the *coboundary* and *boundary* neighborhood functions $\mathcal{N}_C$ and $\mathcal{N}_B$, respectively, are defined, as

$$\mathcal{N}_C(x) = \{y \in \mathcal{X} | y \prec x\}, \quad \mathcal{N}_B(x) = \{y \in \mathcal{X} | x \prec y\}. \tag{17}$$

The up and down adjacencies $\mathcal{N}_{A,\uparrow}$ and $\mathcal{N}_{A,\downarrow}$, respectively, are defined, as

$$\mathcal{N}_{A,\uparrow}(x) = \{y \in \mathcal{X} | \exists z \text{ such that } x \prec z, y \prec z\}, \; \mathcal{N}_{A,\downarrow}(x) = \{y \in \mathcal{X} | \exists z \text{ such that } z \prec x, z \prec y\}. \tag{18}$$

**Cell Complexes as Combinatorial Complexes.** It is again straightforward to frame CW complexes in the CC framework. In particular, a cell complex is a combinatorial complex $(\mathcal{S}, \mathcal{X}, \mathrm{rk})$, where $\mathcal{S}$ is the set of nodes, $\mathcal{X}$ is the set of cells, and the rank function $\mathrm{rk}$ assigns to each cell its dimension dim. A CCMPN as in (3) with $\mathcal{CN} = \{\mathcal{N}_B, \mathcal{N}_{A,\uparrow}\}$ on a CW complex recovers the Molecular Message Passing Network from (Bodnar et al., 2021a). ETNNs as in (6) with the same collection of neighborhoods $\mathcal{CN}$ give rise to $E(n)$ Equivariant Molecular Message Passing Networks, not present in literature[3].

*Remark 4*. Simplicial Complexes and Cell Complexes are rich topological objects. Neighborhoods in a simplicial or a cell complex are strongly rooted in arguments from algebraic topology. In particular, Hodge theory (Griffiths & Schmid, 1975) is used to derive formal notions of boundary, coboundary, and Hodge Laplacians (Barbarossa & Sardellitti, 2020). More detailed descriptions can be found in (Grady & Polimeni, 2010; Barbarossa & Sardellitti, 2020; Battiloro et al., 2023b; Ribando-Gros et al., 2022). In this work, we are interested in the definition of cells (in the CC sense) and neighborhood functions that these objects (SCs and CWCs) induce. For this reason, a reader familiar with algebraic topology could find our definitions and treatment not totally comprehensive. However, we aim to provide a unifying framework in the context of (equivariant) TDL, thus our presentation is sufficiently consistent and strikes a good tradeoff between generality, rigor, and applicability.

## C.4  HYPERGRAPHS

Cell complexes have their formal hierarchical and topology-grounded characterization as their main strength. However, their structure is still constrained by the homeomorphism requirements, preventing them from modeling arbitrary higher-order hierarchical interactions. On the other hand, hypergraphs are combinatorial objects that have total flexibility, but without any hierarchical and (in the general case) topological structure.

**Hypergraphs.** A hypergraph (HG) on a nonempty set of nodes $\mathcal{S}$ is a pair $(\mathcal{S}, \mathcal{E})$, where $\mathcal{X}$ is a subset of the power set of $\mathcal{P}(\mathcal{S}) \setminus \{\emptyset\}$ of $\mathcal{S}$. Elements of $\mathcal{S}$ are called *hyperedges*. In Fig. 6, we sketch a hypergraph with 3 nodes and 3 hyperedges.

---

[3]This approach was recently explored in (Kovač et al., 2024), appeared online only *after* we released the first preprint of ETNN.

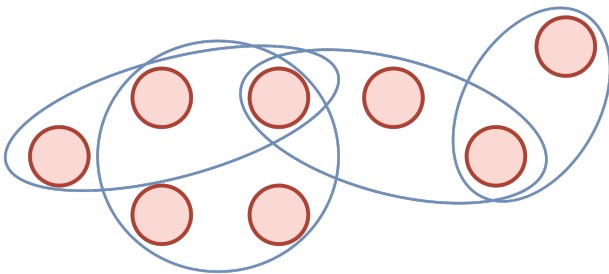

Figure 6: A hypergraph, comprising of nodes (cells of rank 0), hyperedges (cells of rank 1).

**Neighbhorhood Functions in a Hypergraph.** Let us again denote the set of singletons $\widetilde{\mathcal{S}} = \{\{s\}\}_{s \in \mathcal{S}}$. Given a HG $\mathcal{X}$, for all $x \in \widetilde{\mathcal{S}} \cup \mathcal{E}$, the up and down incidences $\mathcal{N}_{I,\uparrow}$ and $\mathcal{N}_{I,\downarrow}$, respectively, are defined as

$$\mathcal{N}_{I,\uparrow}(x) = \{y \in \widetilde{\mathcal{S}} \cup \mathcal{E} | x \subset y\}, \quad \mathcal{N}_{I,\downarrow}(x) = \{y \in \widetilde{\mathcal{S}} \cup \mathcal{E} | y \subset x\}. \tag{19}$$

Therefore, if $x$ is a node, the down and up incidences evaluated at $x$ are the empty set and the hyperedges containing $x$, respectively. If $x$ is a hyperedge, then the down and up neighborhood functions evaluated at $x$ are the nodes contained in $x$ and the empty set, respectively.

**Hypergraphs as Combinatorial Complexes.** Even in this case, framing HGs in the CC framework is straightforward. In particular, a hypergraph $\mathcal{X}$ is a combinatorial complex $(\mathcal{S}, \mathcal{X}, \text{rk})$, where $\mathcal{S}$ is the set of nodes, the set of cells $\mathcal{X} = \widetilde{\mathcal{S}} \cup \mathcal{E}$, and the rank function $\text{rk}$ assigns rank 0 to the nodes and rank 1 to the hyperedges. A CCMPN as in (3) with $\mathcal{CN} = \{\mathcal{N}_{I,\uparrow}, \mathcal{N}_{I,\downarrow}\}$ recovers the Hypergraph Message Passing Networks from (Feng et al., 2019) (in the convolutional realization) and (Heydari & Livi, 2022). ETNNs as in (6) with the same collection of neighborhoods $\mathcal{CN}$ give rise to $E(n)$ Equivariant Hypergraph Message Passing Networks, not present in literature.

## D   ETNNs with Velocity Type Inputs

As EGNNs, ETNNs can handle settings in which nodes come with velocity type geometric features. In this case, we have non-geometric feature $\mathbf{h}_x \in \mathbb{R}^F$, and positions $\mathbf{x}_z \in \mathbb{R}^n$ and velocities $\mathbf{v}_z$ for cell $z \in \mathcal{S}$. We simply need to modify the position update in (7) to include the velocity update. Let us denote the velocity of the $z$-th node at layer $l$ with $\mathbf{v}_z^l$. At the $l$-th layer, an ETNN updates the embeddings $\mathbf{h}_x^l$ of every cell $x \in \mathcal{X}$ as in (6), and the position $\mathbf{x}_z^l$ and velocity $\mathbf{v}_z^l$ of every node $z \in \mathcal{S}$ as

$$\mathbf{v}_z^{l+1} = \zeta\left(\mathbf{h}_z^l\right) \mathbf{v}_z + C \sum_{\mathcal{N} \in \mathcal{CN}} \sum_{t \in \mathcal{S}:\{t\} \in \mathcal{N}(z)} \left(\mathbf{x}_z^l - \mathbf{x}_t^l\right) \xi\left(\mathbf{m}_{z,t}^{\mathcal{N}}\right), \quad \mathbf{x}_z^{l+1} = \mathbf{x}_z^l + \mathbf{v}_z^{l+1}. \tag{20}$$

## E   Proof of Theorem 1

The proof of Theorem 1 is a straightforward generalization of the proof provided for Equivariant Graph Neural Networks (EGNN) in (Satorras et al., 2021). In particular, we can directly state that the update of the features in (6) is $E(n)$ invariant because the positions $\{\mathbf{x}_z\}_{z \in x}$ are used only as input to the geometric invariant(s) Inv and the features $\mathbf{h}_x$ contain no geometric information, i.e.

$$\mathbf{h}_x^{l+1} = \beta\left(\mathbf{h}_x^l, \bigotimes_{\mathcal{N} \in \mathcal{CN}} \bigoplus_{y \in \mathcal{N}(x)} \underbrace{\psi_{\mathcal{N},\text{rk}(x)}\left(\mathbf{h}_x^l, \mathbf{h}_y^l, \text{Inv}\left(\{\mathbf{O}\mathbf{x}_z^l + \mathbf{b}\}_{z \in x}, \{\mathbf{O}\mathbf{x}_z^l + \mathbf{b}\}_{z \in y}\right)\right)}_{\mathbf{m}_{x,y}^{\mathcal{N}}}\right), \tag{21}$$

for all $x \in \mathcal{X}$ and $(\mathbf{O}, \mathbf{b}) \in E(n)$. Therefore, we just need to show that the update of the positions in (7) is $E(n)$ equivariant, i.e., we want to prove that

$$\mathbf{O}\mathbf{x}_z^{l+1} + \mathbf{b} = \mathbf{O}\mathbf{x}_z^l + \mathbf{b} + C \sum_{\mathcal{N} \in \mathcal{CN}} \sum_{t \in \mathcal{S}:\{t\} \in \mathcal{N}(z)} \left((\mathbf{O}\mathbf{x}_z^l + \mathbf{b}) - (\mathbf{O}\mathbf{x}_t^l + \mathbf{b})\right) \xi\left(\mathbf{m}_{z,t}^{\mathcal{N}}\right), \tag{22}$$

for all $z \in \mathcal{S}$ and $(\mathbf{O}, \mathbf{b}) \in E(n)$. By direct computation, as showed in (Satorras et al., 2021), we can write

$$\mathbf{O}\mathbf{x}_z^l + \mathbf{b} + C \sum_{\mathcal{N} \in \mathcal{CN}} \sum_{t \in S:\{t\} \in \mathcal{N}(z)} \left( (\mathbf{O}\mathbf{x}_z^l + \mathbf{b}) - (\mathbf{O}\mathbf{x}_t^l + \mathbf{b}) \right) \xi \left( \mathbf{m}_{z,t}^{\mathcal{N}} \right)$$

$$= \mathbf{O}\mathbf{x}_z^l + \mathbf{b} + \mathbf{O}C \sum_{\mathcal{N} \in \mathcal{CN}} \sum_{t \in S:\{t\} \in \mathcal{N}(z)} \left( \mathbf{x}_z^l - \mathbf{x}_t^l \right) \xi \left( \mathbf{m}_{z,t}^{\mathcal{N}} \right)$$

$$= \mathbf{O} \left( \mathbf{x}_z^l + C \sum_{\mathcal{N} \in \mathcal{CN}} \sum_{t \in S:\{t\} \in \mathcal{N}(z)} \left( \mathbf{x}_z^l - \mathbf{x}_t^l \right) \xi \left( \mathbf{m}_{z,t}^{\mathcal{N}} \right) \right) + \mathbf{b} = \mathbf{O}\mathbf{x}_z^{l+1} + \mathbf{b}. \quad (23)$$

Therefore, the proof is concluded as the updates in (6)-(7) are $E(n)$ invariant and equivariant, respectively.

The proof is slightly different if velocities $\mathbf{v}_z$ are available. In this case, we want to prove that the updates in (20) are $E(n)$ equivariant. Therefore, for the velocity update, we want to show that

$$\mathbf{O}\mathbf{v}_z^{l+1} = \zeta \left( \mathbf{h}_z^l \right) \mathbf{O}\mathbf{v}_z^0 + C \sum_{\mathcal{N} \in \mathcal{CN}} \sum_{t \in S:\{t\} \in \mathcal{N}(z)} \left( (\mathbf{O}\mathbf{x}_z^l + \mathbf{b}) - (\mathbf{O}\mathbf{x}_t^l + \mathbf{b}) \right) \xi \left( \mathbf{m}_{z,t}^{\mathcal{N}} \right), \quad (24)$$

for all $z \in \mathcal{S}$ and $(\mathbf{O}, \mathbf{b}) \in E(n)$. Again by direct computation, we can write

$$\zeta \left( \mathbf{h}_i^l \right) \mathbf{O}\mathbf{v}_z^0 + C \sum_{\mathcal{N} \in \mathcal{CN}} \sum_{t \in S:\{t\} \in \mathcal{N}(z)} \left( (\mathbf{O}\mathbf{x}_z^l + \mathbf{b}) - (\mathbf{O}\mathbf{x}_t^l + \mathbf{b}) \right) \xi \left( \mathbf{m}_{z,t}^{\mathcal{N}} \right)$$

$$= \mathbf{O}\zeta \left( \mathbf{h}_i^l \right) \mathbf{v}_z^0 + \mathbf{O}C \sum_{\mathcal{N} \in \mathcal{CN}} \sum_{t \in S:\{t\} \in \mathcal{N}(z)} \left( \mathbf{x}_z^l - \mathbf{x}_t^l \right) \xi \left( \mathbf{m}_{z,t}^{\mathcal{N}} \right)$$

$$= \mathbf{O} \left( \zeta \left( \mathbf{h}_i^l \right) \mathbf{v}_z^0 + C \sum_{\mathcal{N} \in \mathcal{CN}} \sum_{t \in S:\{t\} \in \mathcal{N}(z)} \left( \mathbf{x}_z^l - \mathbf{x}_t^l \right) \xi \left( \mathbf{m}_{z,t}^{\mathcal{N}} \right) \right) = \mathbf{O}\mathbf{v}_z^{l+1}. \quad (25)$$

It is trivial to see that the position update in (20) is $E(n)$ equivariant. Therefore, the proof is concluded as the updates in (6)-(20) are $E(n)$ invariant and equivariant, respectively.

## F   EXPRESSIVENESS OF ETNNs

The ability of Graph Neural Networks (GNNs) to differentiate between non-isomorphic graphs is typically employed as an expressivity metric. In particular, it is well known that vanilla GNNs are at maximum as powerful as the Weisfeiler-Leman (WL) test (Weisfeiler & Leman, 1968) in discriminating isomorphic graphs. As such, the WL test sets a limit on the expressiveness of GNNs. To broaden the use of this method to include geometric graphs, the Geometric Weisfeiler-Leman test (GWL) has been introduced (Joshi et al., 2023). This test evaluates whether two (undirected) graphs are geometrically isomorphic, as defined next.

**Geometrically Isomorphic Graphs.** (Joshi et al., 2023; Sestak et al., 2024) Two geometric graphs $\mathcal{G}_1 = (\mathcal{S}_1, \mathcal{E}_1)$ and $\mathcal{G}_2 = (\mathcal{S}_2, \mathcal{E}_2)$ with $|\mathcal{S}_1| = |\mathcal{S}_2|$, non-geometric and geometric $i$-th node features $\mathbf{h}_{x_i}^{\mathcal{G}_j}$ and $\mathbf{x}_{x_i}^{\mathcal{G}_j}$ ($j \in \{1, 2\}$), are called geometrically isomorphic if there exists an edge-preserving bijection $b : \mathcal{I}_1 \to \mathcal{I}_2$ between their corresponding node indices set $\mathcal{I}_1$ and $\mathcal{I}_2$, such that their positions are equivalent up to the $E(n)$ group action, i.e.:

$$\mathbf{h}_{x_{b(i)}}^{\mathcal{G}_2}, \mathbf{x}_{x_{b(i)}}^{\mathcal{G}_2} = \mathbf{h}_{x_i}^{\mathcal{G}_1}, \mathbf{O}\mathbf{x}_{x_i}^{\mathcal{G}_1} + \mathbf{b} \quad (26)$$

for all $i \in \mathcal{I}_1, (\mathbf{O}, \mathbf{b}) \in E(n)$.

**k-hop Distinct Graphs.** (Joshi et al., 2023; Sestak et al., 2024) Two geometric graphs $\mathcal{G}_1$ and $\mathcal{G}_2$ are said to be $k$-hop distinct if for all graph isomorphisms $b$, there is some node $i \in \mathcal{I}_1, b(i) \in \mathcal{I}_2$ such that the corresponding $k$-hop neighborhood subgraphs are distinct (not geometrically isomorphic). Otherwise, if they are identical up to $E(n)$ actions for all $i \in \mathcal{I}_1$, we say $\mathcal{G}_1$ and $\mathcal{G}_2$ are said to be $k$-hop identical.

To study Combinatorial Complexes (CCs) and the expressiveness of Equivariant CCMPNs (ETNNs), we introduce the notion of Geometric Augmented Hasse Graph representation of CCs, generalizing

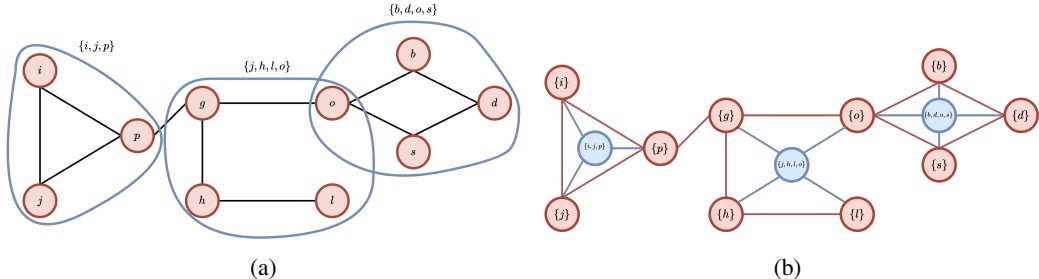

Figure 7: (a) A combinatorial complex with nodes of the underlying graph as cells of rank 0, and with three arbitrary cells of rank 1. (b) The corresponding geometric augmented Hasse graph if $\mathcal{CN} = \{\mathcal{N}_{A,\uparrow}, \mathcal{N}_{I,\uparrow}, \mathcal{N}_{I,\downarrow}\}$ with $\mathcal{N}_{A,\uparrow}$ as in (14), and $\bigoplus$ is the mean. This is also an example of a skeleton-preserving lift.

(Hajij et al., 2022b) and the notion of Hasse diagram for finite partially ordered sets (Demey & Smessaert, 2014) to the geometric setting.

**Geometric Augmented Hasse Graph of a Combinatorial Complex.** Let $(\mathcal{S}, \mathcal{X}, \mathrm{rk})$ be a CC where each node $z \in \mathcal{S}$ comes with positions $\mathbf{x}_z$, and $\mathcal{CN}$ a collection of neighborhood functions. The Geometric Augmented Hasse graph $\mathcal{G}_\mathcal{X}$ of $(\mathcal{S}, \mathcal{X}, \mathrm{rk})$ is a (possibly) directed geometric graph $\mathcal{G}_\mathcal{X} = (\mathcal{X}, \mathcal{E})$ with cells as nodes, edges given by

$$\mathcal{E} = \{(y, x) | x \in \mathcal{X}, y \in \mathcal{X}, \exists \mathcal{N} \in \mathcal{CN} : y \in \mathcal{N}(x)\}, \tag{27}$$

and positions of cell $x$ being a linear permutation invariant function $\bigoplus_{z \in x} \mathbf{x}_z$ of its corresponding node positions$\{\mathbf{x}_z\}_{z \in x}$, $z \in \mathcal{S}$. Therefore, the geometric augmented Hasse graph of a CC is a graph representation of it obtained by considering the cells as nodes, by inserting edges among them if the cells are neighbors in the CC, and by assigning positions to higher-order cells as functions of the nodes (of the CC) they contain. Fig. 7 shows an example of a CC and the corresponding geometric augmented Hasse graph. Given a CC $\mathcal{C}$, a standard graph message-passing network that runs over the augmented Hasse graph $\mathcal{G}_\mathcal{X}$ induced by a collection of neighborhoods $\mathcal{CN}$ is equivalent to a specific CCMPN derived as a particular case of (3) and running over $\mathcal{X}$ using $\mathcal{CN}$ (Hajij et al., 2022b). The following also holds.

*Proposition 1* . Let $(\mathcal{S}, \mathcal{X}, \mathrm{rk})$ be a CC where each node $z \in \mathcal{S}$ comes with positions $\mathbf{x}_z$, $\mathcal{CN}$ a collection of neighborhood functions, and $\mathcal{G}_\mathcal{X}$ the corresponding geometric augmented Hasse graph. Assuming that:

1. a single message function for all neighborhoods and ranks, i.e., $\psi_{\mathcal{N}, \mathrm{rk}(x)} = \psi$ in (6);

2. the inter- and intra-neighborhood aggregation functions are the same, i.e., $\bigotimes = \bigoplus$ in (6);

3. the only employed geometric invariants are sum of distances of linear permutation invariant functions as in (10), e.g. sum or mean.

Then ETNNs over $(\mathcal{S}, \mathcal{X}, \mathrm{rk})$ are equivalent to EGNNs (Satorras et al., 2021) over $\mathcal{G}_\mathcal{X}$ with the following modified update for positions of node $x$ (of the Hasse graph):

$$\mathbf{x}_x^{l+1} = \begin{cases} \mathbf{x}_x^l + C \sum_{\mathcal{N} \in \mathcal{CN}} \sum_{t \in S:\{t\} \in \mathcal{N}(z)} \left(\mathbf{x}_x^l - \mathbf{x}_t^l\right) \xi\left(\mathbf{m}_{x,t}\right), & \text{if } x \in \mathcal{S}, \\ \mathbf{x}_x^{l+1} = \bigoplus_{z \in x} \mathbf{x}_z^{l+1}, x \in \mathcal{X} \setminus \mathcal{S} \end{cases} \tag{28}$$

*Proof.* The proof is trivial and is obtained by simple direct substitution in (6)-(7). $\square$

Intuitively, the modified update is required because higher-order cells do not come with positions, thus they cannot be updated as the positions of the nodes in $\mathcal{S}$.

**Lifting of a Graph into a CC.** Given a graph, many ways exist to lift it into a combinatorial complex. As shown in Appendix C.1, denoting the set of singletons $\widetilde{\mathcal{S}} = \{\{s\}\}_{s \in \mathcal{S}}$, a graph $\mathcal{G}(\mathcal{S}, \mathcal{E})$ is a

combinatorial complex $(\mathcal{S}, \mathcal{X}, \mathrm{rk})$, where $\mathcal{S}$ is the set of nodes, the set of cells $\mathcal{X} = \widetilde{\mathcal{S}}$ is the set of singletons, the rank function $\mathrm{rk}$ assigns rank 0 to each singleton, and the up adjacency from (14) is employed. However, we can enrich the representation by including the edges $\mathcal{E}$ as cells, i.e., $\mathcal{X} = \widetilde{\mathcal{S}} \bigcup \mathcal{E}$, and assign rank 1 to them. In this setting, it is possible to make nodes communicate *with* edges and vice-versa, rather than just making nodes communicate *through* edges. In general, a natural question is how to define higher-order cells, i.e., cells of rank greater than 0. We have already shown in Section 4 that domain knowledge can be injected by properly modeling higher-order cells. However, we are now interested in understanding which conditions should higher-order cells respect to result in an increased expressive power of ETNN. To perform a fair analysis, we consider only skeleton-preserving lifts (Bodnar et al., 2021a), i.e., lifts in which the original connectivity among the nodes in $\mathcal{S}$ is preserved. In other words, we don't directly rewire the underlying graph $\mathcal{G}$. As shown in Fig. 7, a straightforward example of skeleton-preserving lifts is choosing (1)-(14) as neighborhood functions without necessarily adding the edges as cells. Another example is choosing (1)-(2) as neighborhood functions including the edges as cells.

**Expressivity of ETNN.** Due to Proposition 1, given two geometric graphs $\mathcal{G}_1$ and $\mathcal{G}_2$ and two combinatorial complexes $\mathcal{X}_{\mathcal{G}_1}$ and $\mathcal{X}_{\mathcal{G}_2}$ obtained by respectively lifting them, we can study how expressive ETNNs are by analyzing how the GWL performs on the geometric augmented Hasse graphs $\mathcal{G}_{\mathcal{X}_{\mathcal{G}_1}}$ and $\mathcal{G}_{\mathcal{X}_{\mathcal{G}_1}}$ of $\mathcal{X}_{\mathcal{G}_1}$ and $\mathcal{X}_{\mathcal{G}_2}$, respectively, w.r.t. how it would perform directly on the underlying graphs $\mathcal{G}_1$ and $\mathcal{G}_2$. Since the GWL operates on undirected graphs, we need to assume that the produced Hasse graph is undirected. The two examples of skeleton-preserving lifts clearly result in undirected Hasse graphs. We would obtain a directed Hasse graph if, e.g., only one of the up/down incidences in (1) is used as a neighborhood function. Similar to the standard graph WL test, the GWL method updates node colors iteratively based on the features of nodes in the local neighborhood. Additionally, GWL maintains $E(n)$-equivariant hash values that capture each node's local geometry. Therefore, any $k$-hop distinct, $(k-1)$-hop identical geometric graphs can be distinguished by $k$ iterations of GWL, i.e., when the updated hash values that capture each node's local geometry differ across the two graphs for the first time, or by $k$ layers of an EGNN, i.e., when the geometric invariants differ across the two graphs for the first time (Joshi et al., 2023).

***Proposition 2 .*** Assume to have two $k$-hop distinct geometric graphs $\mathcal{G}_1 = (\mathcal{S}_1, \mathcal{E}_1)$ and $\mathcal{G}_2 = (\mathcal{S}_2, \mathcal{E}_2)$ where the underlying graphs are isomorphic in the standard sense, two combinatorial complexes $\mathcal{X}_{\mathcal{G}_1}$ and $\mathcal{X}_{\mathcal{G}_2}$ and a collection $\mathcal{CN}$ of neighborhood functions obtained via a skeleton-preserving lift and leading to undirected geometric Hasse graphs $\mathcal{G}_{\mathcal{X}_{\mathcal{G}_1}}$ and $\mathcal{G}_{\mathcal{X}_{\mathcal{G}_2}}$, respectively. An ETNN operating over $\mathcal{X}_{\mathcal{G}_1}$ and $\mathcal{X}_{\mathcal{G}_2}$ respecting Assumptions 1-3 can distinguish $\mathcal{G}_1$ and $\mathcal{G}_2$ in $M$ layers, where $M$ is the number of layers required to have at least one cell in $\mathcal{X}_{\mathcal{G}_1}/\mathcal{X}_{\mathcal{G}_2}$ whose receptive field is the whole set of nodes in $\mathcal{G}_1/\mathcal{G}_2$.

*Proof.* To prove the result, we study the behavior of the GWL on the geometric augmented Hasse graphs $\mathcal{G}_{\mathcal{X}_{\mathcal{G}_1}}$ and $\mathcal{G}_{\mathcal{X}_{\mathcal{G}_2}}$ of $\mathcal{X}_{\mathcal{G}_1}$ and $\mathcal{X}_{\mathcal{G}_2}$, respectively. For 1-hop distinct graphs, one iteration of GWL suffices to distinguish them even without any lifting into a CC and, thus, the proposition holds.

Now, let us assume that $\mathcal{G}_1$ and $\mathcal{G}_2$ are $k$-hop distinct and $(k-1)$-hop identical for any $k > 1, k \in \mathbb{N}$. As a direct consequence of Proposition 1, $M$ GWL iterations are required for (at least one) node of the Hasse graphs $\mathcal{G}_{\mathcal{X}_{\mathcal{G}_1}}/\mathcal{G}_{\mathcal{X}_{\mathcal{G}_2}}$ to have the entire set of nodes $\mathcal{S}_1/\mathcal{S}_2$ in its receptive field. Denote that node with $x_i \in \mathcal{G}_1$ and $x_{b(i)} \in \mathcal{G}_2$, where $b$ is any isomorphism between $\mathcal{G}_1$ and $\mathcal{G}_2$. Due to the k-hop distinctness of the graphs, the hash values corresponding to node $x_i \in \mathcal{G}_1$ and $x_{b(i)} \in \mathcal{G}_2$ will differ in the $M$-th iteration, thus $M$ iterations of GWL can distinguish the graphs. Therefore. Combining the fact that EGNNs are as powerful as the GWL in distinguishing $k$-hop distinct graphs (Sestak et al., 2024; Joshi et al., 2023) with the result from Proposition 1, the proof is completed. □

***Corollary 1 .*** An ETNN as in Proposition 2 is strictly more powerful than an EGNN (Satorras et al., 2021) in distinguishing $k$-hop distinct graphs $\mathcal{G}_1$ and $\mathcal{G}_2$ if the employed skeleton-preserving lifting is such that the number of ETNN layers required to have at least one cell in $\mathcal{X}_{\mathcal{G}_1}/\mathcal{X}_{\mathcal{G}_2}$ whose receptive field is the whole set of nodes in $\mathcal{G}_1/\mathcal{G}_2$ is smaller than the number of EGNN layers required to have at least one node in $\mathcal{G}_1/\mathcal{G}_2$ whose receptive field is the whole set of nodes in $\mathcal{G}_1/\mathcal{G}_2$.

*Proof.* Direct consequence of Proposition 2. □

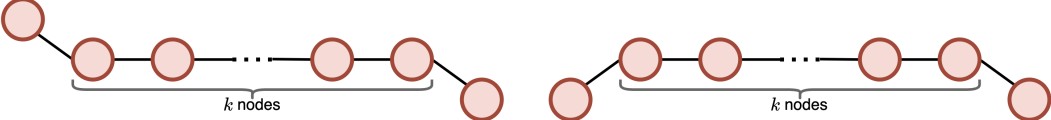

Figure 8: $k$-chain geometric graphs, i.e., $k$ aligned nodes and two end nodes with opposite orientations.

***Remark 5 .*** The reader may have noticed that, in proving Proposition 2, we used the result from Proposition 1 ignoring the modified position update rule in (28), i.e., implicitly assuming that also the positions of higher-order cells, when seen as nodes of the corresponding augmented Hasse graph, are updated as in (7), following the standard EGNN. This is not relevant, as the modified update rule in (28) is actually more powerful than the standard rule in (7), because the positions of the higher-order cell $x$ at layer $l$ is updated using the already updated node positions $\bigoplus_{z \in x} \mathbf{x}_z^{l+1}$ and not the $\mathbf{x}_z^l$s. For this reason, Proposition 2 is somehow a pessimistic scenario for the practical use of ETNNs, and there could be corner cases in which ETNNs can distinguish $k$-hop distinct geometric graphs with a lower number of layers than prescribed. Finally, in practice, we observed it is not important to stick to the homogeneous setting (Assumption 1), to a unique aggregation function (Assumption 2), or to a certain geometric invariant (Assumption 3). These assumptions are useful to simplify the tractation and have a perfect match with the geometric augmented Hasse graph formalism.

**Discussion.** Proposition 2 provides formal hints about how different lifts of graphs into combinatorial complexes can impact the expressive power of ETNNs. Our result can be seen as a proper generalization of Proposition 2 in (Sestak et al., 2024) about the expressivity of VN-EGNNs. Indeed, adding a virtual node connected to all the other nodes can be interpreted as a combinatorial complex in which the cells are the set of singletons (the nodes) plus a cell containing all the nodes (the virtual node). A (simplified) variant of a VG-EGNN is then obtained by using the adjacency induced by the edges of the original graph and the up/down incidences as neighborhood functions. In this case, as prescribed from Proposition 2, the receptive field of the virtual node is the whole graph by definition, making VN-EGNNs able to distinguish $k$-hop distinct graphs in one layer. Our results also go in the direction of (Jogl et al., 2023), in which the expressivity of standard graph message passing networks on transformed/lifted graphs is systematically studied. This work provides a first, preliminary intuition on how the framework from (Jogl et al., 2023) could be extended to the geometric graph setting. Moreover, it is worth noting that our expressivity analysis could be enriched through the same GWL framework from (Joshi et al., 2023), eventually leading to a topological/higher-order generalization of it. An interesting direction is studying the expressivity of ETNNs when the lifts result in directed geometric augmented Hasse graphs. Finally, our result could shed light on which lifts should be preferred. In the current literature (Bodnar et al., 2021a; Hajij et al., 2022b; Battiloro et al., 2023c; Eijkelboom et al., 2023) except for (Battiloro et al., 2024a), the lifts are always decided based on simple criteria (e.g., if the domain is a simplicial complex, all the triangles of the underlying graphs are considered as 2-simplices). However, lifts could strike desirable tradeoffs between expressivity and learning performance (e.g. if the domain is again a simplicial complex, only some triangles could be chosen without expressivity drop).

**Experimental Validation.** To validate our theoretical claims, we follow the approach from (Joshi et al., 2023; Sestak et al., 2024). In particular, we employ pairs of $k$-chain geometric graphs, where each graph pair consists of $k$ nodes linearly aligned and two endpoints with different orientations, as shown in Fig. 8. These graphs are clearly $\left(\left\lfloor \frac{k}{2} \right\rfloor + 1\right)$-hop distinct, and thus should be distinguishable by $\left(\left\lfloor \frac{k}{2} \right\rfloor + 1\right)$ EGNN layers or the same number of GWL iterations (Joshi et al., 2023; Sestak et al., 2024). To show the correctness of our result, we consider $k = 4$, and we lift each pair of graphs into a CC in different ways, listed below. In all the cases, we use the mean to assign positions to cells in the Hasse graph (Assumption 3 of Proposition 1). Without loss of generality, we use $\mathcal{N}_{A,\uparrow}$ as in (14) (the usual node adjacency) and the up/down incidences $\mathcal{N}_{I,\uparrow}/\mathcal{N}_{I,\downarrow}$ in (1) as neighborhood functions, i.e., we keep the connectivity among nodes as in the original pair of graphs without including edges as cells and we allow higher-order cells to communicate only with nodes but not among them (thus resulting in a hypergraph-like structure). The readout classifier takes as input the geometric invariants computed on the final updated positions.

- Lift 1a: we consider a set of cells composed of the nodes of the original graphs plus one 1-cell containing all the nodes, as shown in Fig. 9a (a). In Figure 9b, we show the corresponding

Table 3: Classification Accuracy on 4-chain geometric graphs.

| $k = 4$ | Hidden Dim. | Number of Layers | | | |
| | | 1 | 2 | 3 | 4 |
| --- | --- | --- | --- | --- | --- |
| GWL | | 50% | 50% | 100% | 100% |
| EGNN from (Sestak et al., 2024) | 32 | 50% | 50% | 56.5% | 50% |
| | 64 | 50% | 50% | 100% | 99% |
| | 128 | 50% | 50% | 96.5% | 98.5% |
| ETNN Lift 1a | 32 | 100% | 100% | 100% | 100% |
| | 64 | 100% | 100% | 100% | 100% |
| | 128 | 100% | 100% | 100% | 100% |
| ETNN Lift 2a | 32 | 100% | 100% | 100% | 100% |
| | 64 | 90% | 100% | 100% | 100% |
| | 128 | 90% | 100% | 100% | 100% |
| ETNN Lift 3a | 32 | 50% | 85% | 100% | 100% |
| | 64 | 50% | 80% | 95% | 85% |
| | 128 | 50% | 80% | 95% | 95% |

geometric Hasse graphs. All the nodes will have the whole graph as their receptive field before the first iteration, thus we expect the GWL on the Hasse graph, hence ETNN, to distinguish the graphs using one layer. This case would be solved even by just feeding the initial geometric invariants to the readout classifier (using "0" layers).

- Lift 2a: we consider a set of cells composed of the nodes of the original graphs plus two 1-cells of the same cardinality being overlapping paths of length 4, as shown in Fig. 10a. In Figure 10b, we show the corresponding geometric Hasse graphs. It is clear that nodes $\{p_1\}, \{p_2\}$ will have the whole graph as their receptive field in one iteration, thus we expect the GWL on the Hasse graph, hence ETNN, to distinguish the graph using one layer.

- Lift 3a: we consider a set of cells composed of the nodes of the original graphs plus two 1-cells of different cardinality and not overlapping, as shown in Fig. 11a. In Figure 11b, we show the corresponding geometric Hasse graphs. It is clear that nodes $\{p_1\}, \{p_2\}, \{o_1\}, \{p_2\}$ will have the whole graph as their receptive field in two iterations, thus we expect the GWL on the Hasse graph, hence ETNN, to distinguish the graph using two layers.

Per each lift, we trained ETNNs with an increasing number of layers to classify the pairs of graphs. Due to possible oversquashing effects (Joshi et al., 2023), we also trained ETNNs with 5 different hidden dimensions and averaged over 10 different seeds. In Table 3, we report the results. We also report the results for EGNN (taken from (Sestak et al., 2024)) and the GWL on the original pair of graphs without any lifting into a CC. As the reader can see, empirical results confirm the validity of our analysis, and the clear advantage, in terms of expressivity, of using CCs and ETNNs over standard EGNNs. Further empirical advantage is reported, pointing at higher-order interactions as a practical solution to oversquashing (Giusti et al., 2023b; Di Giovanni et al., 2023).

***Remark 6 .*** The lifts we used in the experiments above are handcrafted lifts, employed to validate Proposition 2 in a easy and consistent way. However, our result obviously holds in all the cases, even with more ranks and more sophisticated collections of neighborhood functions (e.g. induced by skeleton-preserving lift to simplicial, cell, or path complexes(Bodnar et al., 2021a; Giusti et al., 2023a; Battiloro et al., 2023c; Truong & Chin, 2024; Bodnar et al., 2021b)). Moreover, it is worth it to notice that, with the experiments above, we investigated the role of the choice of the cells given the collection of neighborhood functions. However, it is interesting to study even the opposite case, in which the cells are fixed and the neighborhood functions are chosen. For example, in Lift 2a, adding the lower adjacency from (2) would have led to solving the task with just one layer because the cells

Table 4: Classification Accuracy on the counterxamples graphs.

| | | Body Order | | |
|---|---|---|---|---|
| | | 2-body | 3-body | 4-body |
| EGNN from (Joshi et al., 2023) | | 50% | 50% | 50% |
| ETNN Lift 1b | Inv. | 100% | 100% | 50% |
| | Equiv. | 100% | 100% | 50% |
| ETNN Lift 2b | Inv. | 100% | 50% | 50% |
| | Equiv. | 100% | 100% | 50% |
| ETNN Lift 3b | Inv. | 100% | 100% | 50% |
| | Equiv. | 100% | 100% | 50% |
| ETNN Lift 4b | Inv. | 100% | 100% | 50% |
| | Equiv. | 100% | 100% | 50% |

$c_1/c_2$ and $d_1/d_2$ would have had the whole graph as their receptive field before the first iteration. An opposite example serves as proof of Proposition 3 of the main body.

***Proposition 3 .*** There exists a pair of CCs whose nodes come with geometric features and a collection of neighborhoods such that the CCs are undistinguishable by ETNN.

*Proof.* We take again a pair of $k$-chain graphs $\mathcal{G}_1$ and $\mathcal{G}_1$, and we lift them in two CCs using Lift 3a but employing only the up adjacency from (2) as neighborhood function. In this case, ETNN is not able to distinguish the CCs because the resulting geometric augmented Hasse graph is a disconnected graph, since $p_1/p_2$ and $o_1/o_2$ are been connected. □

**Counterexamples Structures.** To provide an even more in-depth analysis of the expressivity of ETNNs beyond the distinguishability of $k$-hop distinct graphs, we also tested them on the counterexamples graphs from (Pozdnyakov et al., 2020; Joshi et al., 2023), which test a layer's ability to create distinguishing fingerprints for local neighborhoods and stress exactly the importance of higher body order of scalarization. In particular, this task evaluates single-layer geometric architectures in distinguishing counterexample graphs that cannot be differentiated using $k$-body scalarization. we refer the reader to (Pozdnyakov et al., 2020; Joshi et al., 2023) for further details. We tested 4 different lifts:

- Lift 1b: we consider nodes and edges as 0-cells and 1-cells.

- Lift 2b: we consider a set of cells composed of the nodes of the original graphs plus one 1-cell containing all the nodes (equivalent to Lift 1a).

- Lift 3b: we consider nodes, edges, and a cell containing all the nodes as 0-cells, 1-cells, and 2-cell, respectively.

- Lift 4b: we consider nodes, edges, and paths of length 3 as 0-cells, 1-cells, and 2-cells (this is equivalent to a path complex).

For all the lifts, we use the up/down incidences $\mathcal{N}_{I,\uparrow}/\mathcal{N}_{I,\downarrow}$ and adjacencies $\mathcal{N}_{A,\uparrow}/\mathcal{N}_{A,\downarrow}$ from (1)-(2), respectively, as neighborhood functions, and we test both the invariant and equivariant versions of ETNN. The results are presented in Table 4. As the reader can see, ETNN is able to distinguish all the counterexamples but the 4-body (chiral).

***Remark 7 .*** Finally, please notice that we also indirectly showed that lifting (in a skeleton-preserving way) a graph to a CC and using ETNN can be more effective than lifting it to a simplicial complex (SC) and using EMPSN (Eijkelboom et al., 2023). In particular, on both the numerical expressivity tasks, EMPSN with any skeleton preserving lifting would be as expressive as an EGNN (and, thus, less expressive than an ETNN), as there are no triangles nor cliques.

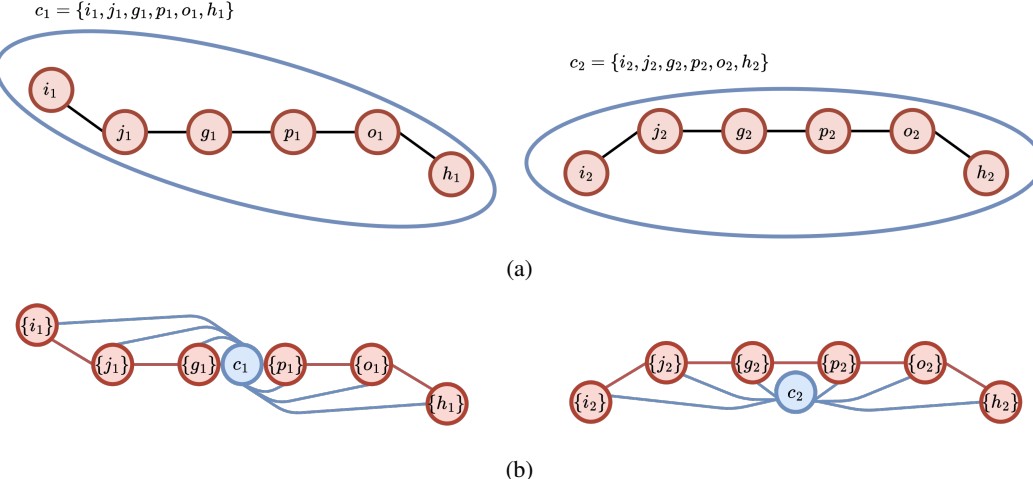

Figure 9: (a) Lift 1a. (b) The corresponding geometric augmented Hasse graphs.

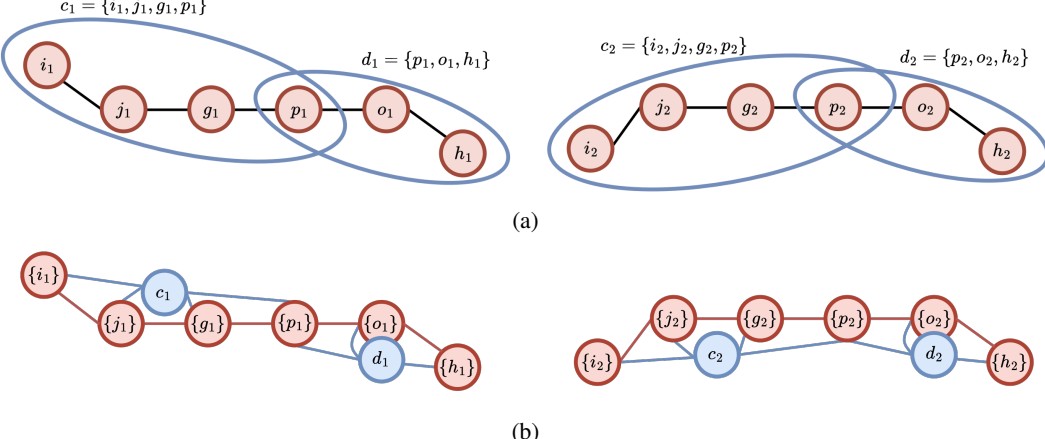

Figure 10: (a) Lift 2a. (b) The corresponding geometric augmented Hasse graphs.

## G  COMPLEXITY ANALYSIS

Given the flexibility in the definition of cells and neighborhood functions, the complexity of an ETNN layer can vary. To provide a reference complexity, let us consider a CC with $|\mathcal{X}|$ cells, and the up/down incidences and adjacencies from (1)-(2) as neighborhood functions. A cell $x \in \mathcal{X}$ has then $|\mathcal{N}_{I,\uparrow}(x)| + |\mathcal{N}_{I,\downarrow}(x)|$ incident cells and there are $|\mathcal{N}_{A,\uparrow}(x)| + |\mathcal{N}_{A,\downarrow}(x)| = \binom{|\mathcal{N}_{I,\uparrow}(x)|+|\mathcal{N}_{I,\downarrow}(x)|}{2}$ up/down adjacencies between them. Therefore, each node $z \in \mathcal{S}$ has $|\mathcal{N}_{A,\downarrow}(z)| = \binom{|\mathcal{N}_{I,\uparrow}(z)|}{2}$ adjacent nodes. Then, an ETNN layer as in (6)-(7) has a computational complexity given by

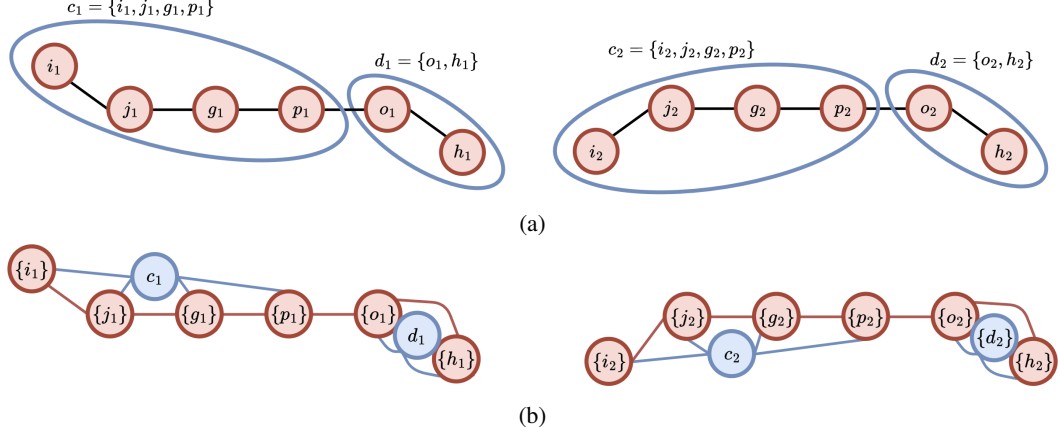

Figure 11: (a) Lift 3a. (b) The corresponding geometric augmented Hasse graphs.

$$
\Theta\left(\underbrace{\sum_{x\in\mathcal{X}}\left((|\mathcal{N}_{I,\uparrow}(x)| + \mathcal{N}_{I,\downarrow}(x)|) + \binom{|\mathcal{N}_{I,\uparrow}(x)| + |\mathcal{N}_{I,\downarrow}(x)|}{2}\right)}_{\mathbf{h}_x \text{ update from (6)}} + \underbrace{\sum_{z\in\mathcal{S}}\binom{|\mathcal{N}_{I,\uparrow}(z)|}{2}}_{\mathbf{x}_z \text{ update from (7)}}\right)
$$

$$
= \Theta\left(\sum_{x\in\mathcal{X}}\left((|\mathcal{N}_{I,\uparrow}(x)| + \mathcal{N}_{I,\downarrow}(x)|) + (|\mathcal{N}_{A,\uparrow}(x)| + |\mathcal{N}_{A,\downarrow}(x)|)\right) + \sum_{z\in\mathcal{S}}|\mathcal{N}_{A,\uparrow}(z)|\right)
$$

$$
= \Theta\left(\sum_{x\in\mathcal{X}}\left(|\mathcal{N}_{A,\uparrow}(x)| + |\mathcal{N}_{A,\downarrow}(x)|\right)\right). \tag{29}
$$

If we consider the connectivity of the CC to be sufficiently sparse, which is usually the case, then

| | Avg. runtime (seconds/epoch) |
|---|---|
| EGNN-graph-W | 60.34 |
| ETNN | 193.16 |
| EMPSN* | 251.3 |

Table 5: Average runtime per epoch for different ETNN configurations

| | Avg. memory consumption (GB) |
|---|---|
| EGNN-graph-W | 13.63 |
| ETNN | 12.05 |
| EMPSN* | 27.7 |

Table 6: Average memory usage for different ETNN configurations

the neighborhoods' cardinalities can be absorbed in the bound. Therefore, the overall complexity will be linear in the number of the cells $\Theta\left(|\mathcal{X}|\right)$. Clearly, in the worst case, each cell could be connected to all the other cells, resulting in a quadratic cost in the number of cells. In practice, with our implementation, both the model's forward and backward pass scale linearly with the number of neighborhood functions, as they are processed in a loop. In Table 5, we compare the runtimes of EGNN-graph-W, EMPSN* (i.e., EMPSN reproduced with our codebase as an instance of ETNN), and an instance of ETNN using the complete Molecular CC (configs. 35 of Tables 11-12). We decided to use this configuration of ETNN because is one of the most computationally intensive. However, better learning performance can be obtained with even (much) lighter configurations, as shown in

Tables 11-12. In this case, ETNN uses exactly 3 times as many neighborhoods as in EGNN-graph-W, as the runtimes confirm. A different implementation could parallelize the processing, making the runtime constant w.r.t. the number of neighborhoods. Most importantly, Table 5 provides the reader with a quantitative metric of the significantly reduced computational burden of ETNN w.r.t. EMPSN (Eijkelboom et al., 2023), as described in the main body. To further highlight this gain, in Table 6 we show the memory usage of the same architectures, proving that ETNN needs less than half memory w.r.t. EMPSN.

# H    ADDITIONAL REFERENCES ON REAL-WORLD COMBINATORIAL TOPOLOGICAL SPACES

## H.1    MOLECULAR DATA

**Molecular graphs.** One of the most renowned representations of molecules is graphs. In this case, atoms are the nodes, and bonds among them are the edges(Gilmer et al., 2017b; Gasteiger et al., 2020; Jiang et al., 2021). Message passing networks on molecular graphs have been proven to be effective on several tasks. However, graphs cannot explicitly model higher-order, multiway interactions among the atoms. For instance, graphs cannot explicitly consider the information coming from the rings, i.e., the induced cycles of the molecular graphs. To represent different atomic relations, efforts focused on learning representations for small molecules and larger biomolecules (i.e., proteins) via multi-view or relational graphs(Ma et al., 2020; Zhang et al., 2023; Sestak et al., 2024; Li et al., 2024b).

**Molecular cell complexes.** To overcome the above limitation, cell complexes (CW complexes), being a generalization of simplicial complexes (Eijkelboom et al., 2023; Bodnar et al., 2021b), have been employed (Bodnar et al., 2021a). Molecular CW complexes can be seen as "hierarchical augmented graphs" in which atoms are 0-cells (nodes), bonds are 1-cells (edges), and rings are 2-cells (induced cycles). Message passing networks on molecular CW complexes have shown superior performance w.r.t. models on molecular graphs (Giusti et al., 2023b; Bodnar et al., 2021a). However, CW complexes cannot model higher-order interactions among the atoms and bonds that are not described by a cycle in the underlying molecular graph. For instance, molecular CW complexes cannot explicitly consider the information coming from the functional groups of the molecule, i.e., specific motifs (not necessarily cycles) of the underlying molecular graphs.

**Molecular hypergraphs.** Hypergraphs (HGs) can model arbitrary higher-order interactions due to the flexibility of hyperedges. HGs and message passing networks on HGs have been employed for modeling and learning from molecules (Chen & Schwaller, 2023; Park et al., 2021). In molecular HGs, atoms are nodes, and bonds, rings, functional groups, etc, are hyperedges. However, HGs cannot explicitly consider any hierarchy in the hyperedges, thus putting pairwise and higher-order interactions of any kind on the same level.

## H.2    GEOSPATIAL DATA

**Spatial graphs and CTS** The single resolution version of the spatial setting in Section 4 has been traditionally handled using proximity graphs, i.e., geometric graphs whose connectivity is dictated by some notion of spatial closeness. In the multiresolution case, the most common approach is to aggregate all the data at a common denominator of spatial resolution. However, substantial information is clearly lost in this process. Some GNN methods for expanding the receptive field of GNNs in spatial data have been proposed. For instance, through edge removal during training to address over-smoothing and over-fitting (Rong et al., 2019), and the introduction of skip connections (He et al., 2016; Li et al., 2018). Yu et al. (Yu et al., 2019) introduced ST-UNet, a heuristic graph pooling method for coarsened graph construction. Note that these methods do not admit multi-level data, i.e., features available in each level of a hierarchy. Hierarchical GNNs (HGNNs) (Diehl et al., 2019; Lee et al., 2019; Ying et al., 2018) create multi-level graphs balancing the expansion of receptive fields and preservation of local features. HGNNs can admit multi-level features. A highly relevant application is due to Li et al. (Li et al., 2019), who utilized a geography-based HGNN for modeling geographic information systems. Importantly, this work does not consider the use of geometric (coordinate information) directly, and therefore, it does not explore equivariant architectures. Other related work includes Wu et al. (Wu et al., 2020), who proposed HRNR for learning road segment representations. Zhang et al. (Zhang et al., 2020) combined HGNNs with semi-supervised learning

for citywide parking availability prediction. It is worth noticing that the constructions of such HGNNs are highly reliant on the specifics of the application, while Geospatial CCs offer a more generally applicable formalism. Motivated by topological approaches, some works used hypergraphs (Jia et al., 2023), whose limitation have been explained in the previous subsection. Some Topological Data Analysis (TDA) work leveraged simplicial complexes (Bajardi et al., 2015; Lo & Park, 2018; Stolz et al., 2016; Feng & Porter, 2021; Hickok et al., 2022). However, TDL should not be confused with TDA, which is a data-driven approach to describing the mathematical properties and "shape" of a topological space. Instead, TDL focuses on modeling relations and data distributions defined over such topological spaces (Barbarossa & Sardellitti, 2020).

# I EXPERIMENTAL SETUP

## I.1 MOLECULAR PROPERTY PREDICTION

In this section, we present all dataset information, data featurization, and model parameter configuration for the molecular property prediction task.

### I.1.1 DATASET STATISTICS

QM9 dataset consists of over $130,000$ molecules with 12 property prediction targets. See Table 7 for a detailed list of the 12 molecular properties, as well as statistics over the number of molecules, atoms, bonds, rings, and functional groups.

Table 7: Statistics of the QM9 Dataset

| Property | Description | Units |
|---|---|---|
| $\alpha$ | Electronic spatial extent | bohr$^3$ |
| $\Delta\epsilon$ | Gap between $\epsilon_{HOMO}$ and $\epsilon_{LUMO}$ | meV |
| $\epsilon_{HOMO}$ | Energy of the highest occupied molecular orbital | meV |
| $\epsilon_{LUMO}$ | Energy of the lowest unoccupied molecular orbital | meV |
| $\mu$ | Dipole moment | D |
| $C_v$ | Heat capacity at 298.15 K | cal/mol K |
| $G$ | Free energy at 298.15 K | meV |
| $H$ | Enthalpy at 298.15 K | meV |
| $U$ | Internal energy at 298.15 K | meV |
| $U_0$ | Internal energy at 0 K | meV |
| $ZPVE$ | Zero-point vibrational energy | meV |
| Number of molecules | | 133,885 |
| Average number of atoms | | 18 nodes |
| Average number of bonds | | 19 |
| Average number of rings | | 1.6 |
| Average number of functional groups | | 1.2 |

### I.1.2 CELL FEATURES

For the molecular prediction task, we considered different peculiar feature vectors for each rank, that can be easily integrated thanks to the ETNN structure.

1. **0-cells** features (atom features):

   - *Atom Types*: QM9 molecular samples consist of 5 atoms: hydrogen (H), carbon (C), nitrogen (N), oxygen (O), and fluorine (F). We used one-hot encoding for the atom types.
   - *Functions over atom types*: Following Cormorant's methodology(Anderson et al., 2019), we computed the atomic fraction (current atomic number divided by the maximum atomic number), multiplied the atomic number with the atomic fraction, and computed the squared quantities.

2. **1-cells** features (bond features):

- *Bond Types*: We can meet $4$ different bond types in QM9 molecules: single, double, triple, aromatic. We used one-hot encoding for the bond types.
- *Conjugation*: A binary feature telling whether the bond is part of a conjugated system, affecting electronic and optical properties.
- *Ring Membership*: binary feature telling whether the bond is part of a ring, affecting electronic and structural characteristics of the molecule.

3. **2-cells** features (ring and functional group features):

*Ring features*

- *Ring Size*: Number of atoms that form the ring.
- *Aromaticity*: A binary feature telling whether the ring is aromatic.
- *Has Heteroatom*: A binary feature telling whether the ring contains a heteroatom, which is an atom other than carbon, such as nitrogen, oxygen, or sulfur. Heteroatoms can introduce different electronic effects and reactivity patterns within the ring structure.
- *Saturation*: A binary feature telling whether the ring is saturated, meaning all the bonds between carbon atoms are single bonds.

*Functional Group features*

- *Conjugation*: A binary feature telling whether the functional group is part of a conjugated system, which involves alternating double and single bonds.
- *Acidity*: The ability of the functional group to donate protons ($H+$ ions) in a solution. Possible values: ["neutral", "high", "basic", "weakly acidic"].
- *Hydrophobicity*: Tendency of the functional group to repel water. 3 Possible values: [hydrophilic, "moderate", "hydrophobic"]
- *Electrophilicity*: Tendency of the functional group to accept electrons during a chemical reaction. High electrophilicity means the functional group readily accepts electrons, influencing its reactivity with nucleophiles. 3 Possible values: ["low", "moderate", "high"].
- *Nucleophilicity*: Tendency of the functional group to donate electrons during a chemical reaction. High nucleophilicity means the functional group readily donates electrons, affecting its reactivity with electrophiles. 3 possible values: ["low", "moderate", "high"].
- *Polarity*: Distribution of electron density within the functional group. High polarity can significantly affect solubility, reactivity, and interactions with other molecules. 3 possible values: ["low", "moderate", "high"].

### I.1.3 PARAMETER CONFIGURATION

Next, we present the ETNN's hyperparameters we used for the molecular property prediction tasks. Table 8 shows the list of the hyperparameters of the optimizer, as well of our model. Specifically:

- Following EGNN's experimentation setup, we use $1e$-3 as the initial learning rate for the properties $\Delta\epsilon, \epsilon_{\text{HOMO}}, \epsilon_{\text{LUMO}}$, and $5e$-4 for the rest of the properties.

- We calibrate the number of hidden units (i.e., layer width) among $\{70, 104, 128, 182\}$ for each configuration, so that we maintain **roughly the same** size of parameter space ($\approx 1.5$M). The motivation for this choice is to provide the fairest comparison possible. see Appendix J for further motivation and details.

- All the reported configurations were run for $1000$ epochs, in agreement with the original EGNN(Satorras et al., 2021).

- We used the same train/validation/test splits as introduced in EGNN(Satorras et al., 2021). See Appendix J for further motivation and details.

- For all the configurations we normalize the geometric invariants via batch normalization. That includes the normalization of pairwise distances, Hausdorff distances, and volumes. Moreover, we perform gradient clipping to a maximum norm of gradients equal to $1.0$.

Table 8: Hyperparameters for ETNN Model in the Molecular Task

| Hyperparameter | Value |
|---|---|
| Optimizer | Adam |
| Initial Learning Rate | $[5 * 10^{-4}, 10^{-3}]$ |
| Learning Rate Scheduler | Cosine Annealing |
| Weight Decay | 1e-5 |
| Batch Size | 96 |
| Epochs | $[100, 200, 350, 1000]$ |
| **Model** | |
| Number of Message Passing Layers | $[4, 7, 10]$ |
| Hidden Units per Layer | $[70, 104, 128, 182]$ |
| Activation Function | SiLU |
| Invariant Normalization | [True, False] |
| Gradient Clipping | [True, False] |

## I.2 HYPERLOCAL AIR POLLUTION DOWNSCALING

### I.2.1 DATASET STATISTICS

We create a geospatial CC using the neighborhood functions described by Eq. (12). The hyperlocal air measurements points (target) are taken from (Wang et al., 2023). The targets are aggregated at the desired resolution of $0.0002° \sim 22m$. After removing roads and tracts that do not contain any measurements, the resulting dataset contains 3,946 point measurements, 550 roads, and 151 census tracts, corresponding to 0-, 1- and 2-cells, respectively.

### I.2.2 CELL FEATURES

The learning targets are the fine-resolution PM 2.5 from (Wei et al., 2023) as described in the main text. As such, this is a semisupervised node regression task. We now describe the features of each rank. For the 0-cells, we generate point-level features using the `osmnx` Python package (Boeing, 2017), calculating metrics such as density and distance to points of interest within a buffer radius. We remove roads and tracts that do not contain any measurements. We consider only points that correspond to weekdays between 9 and 5 pm. For 1-cells, we use the 2021 annual average daily traffic information (New York City Department of Transportation, 2024). For 2-cells we generate land-use features from the Primary Land Use Tax Lot Output (PLUTO)(New York City Department of City Planning, 2024) and previous coarse-resolution estimates of PM 2.5 at the census-tract level are taken from Wei et al. (Wei et al., 2023). Values are monthly averages over the last quarter of 2020, projected to the census tract level using bilinear interpolation from their native 1 km resolution. Below is a list of the employed features. Table 9 summarizes the data sources.

1. **0-cells** features (points; OpenStreetMap (OSMnx) and Wang et al.):
   - *Intersection density*: Using the `osmnx` Python package, we calculate the density of intersections within a 100m buffer radius (Boeing, 2017).
   - *Nearest school distance*: Calculated using `osmnx` to determine the distance to the nearest school (Boeing, 2017).
   - *Restaurant density*: Using `osmnx`, we calculate the density of restaurants within a 300m radius (Boeing, 2017).
   - *Road density*: Derived from `osmnx` by calculating the density of roads within a buffer radius (Boeing, 2017).
   - *Time of day*: We include the median time of measurements by Wang et al. (Wang et al., 2023) for each of the points in a $0.0002° \times 0.0002°$ area.

2. **1-cells** features (road/polylines; AADT, NYC, Dept. of Transportation):
   - *AADT 2021*: Annual Average Daily Traffic data for 2021 (New York City Department of Transportation, 2024).
   - *PCT 2021*: Percentage of trucks traffic (New York City Department of Transportation, 2024).

Table 9: Data sources of the Geospatial Air Pollution Downscaling Benchmark

| Source | Citation | LICENSE |
|---|---|---|
| Wang et al. | (Wang et al., 2023) | CC 4.0 |
| PLUTO, NYC Dept. of Planning | (New York City Department of City Planning, 2024) | Open Data |
| AADT, NYC Dept. of Transportation | (New York City Department of Transportation, 2024) | Open Data |
| OpenStreetMap (OSMnx) | (Boeing, 2017) | Open Data |
| Wei et al. | (Wei et al., 2023) | CC 4.0 |

3. **2-cells** features (census tracts):

   *PLUTO, NYC Dept. of Planning (New York City Department of City Planning, 2024)*

   - *Assessment total*
   - *Building area*
   - *Commercial area (% area)*
   - *Factory area (% area)*
   - *Mixed residential/commercial (% tax lots)*
   - *Multi-family elevator (% tax lots)*
   - *One/two family (% tax lots)*
   - *Open space/outdoor recreation (% tax lots)*
   - *Parking/vacant (% tax lots)*
   - *Public facilities/institutions (% tax lots)*
   - *Transportation/utility (% tax lots)*
   - *Lot area (sq. km)*
   - *Lot depth (m)*
   - *Office area (% area)*
   - *Residential area (% area)*
   - *Retail area (% area)*

   *Other sources*

   - *Coarse PM$_{2.5}$ predictor* (Di et al., 2019): Coarse-resolution monthly estimates of mean PM$_{2.5}$ by census tract.

### I.2.3 PARAMETER CONFIGURATION

The training and evaluation scheme is similar to the molecular case. For training, we use the Huber loss which provides additional robustness to outliers (Huber, 1964). The evaluation metric is obtained similarly to the molecular task. Since the data consists of a single graph, we use a standard procedure for semisupervised tasks on spatial data and generate training, validation, and test masks. To do so while being mindful of spatial leaks and correlations, we randomly select 70% of census tracts for training and split the remaining 30% equally in test and validation tracts. Since points in the dataset only belong to a single census tract, we can use these splits to partition points in their corresponding sets, ensuring that neighboring points belong, on average, to the same split. While all the nodes (training/validation/test) are given as inputs to the model, the loss function for backpropagation is only computed from the error in the training nodes, as in any standard semisupervised setting (Kipf & Welling, 2017).

The evaluation metric is the root-mean-squared error (RMSE) in the test split from the best model of each baseline. The base model is determined by the lowest RMSE in the validation RMSE at any point during training from a sweep over the hyperparameters described in Table 10. The test split only reports the final metric to ensure validity. The EGNN baseline used the same architecture but no lower/upper adjacencies. It also has higher-order features concatenated with point-level features. The multi-layer perception and linear baselines are evaluated similarly. The MLP baseline is implemented with 5 layers to match the number of hidden layers in the ETNN and graph-based baselines.

Table 10: Hyperparameters for ETNN Model in the Air Pollution Downscaling Task

| Hyperparameter | Value |
|---|---|
| Optimizer | Adam |
| Initial Learning Rate | $[10^{-3}, 10^{-2}]$ |
| Learning Rate Scheduler | Cosine Annealing |
| Weight Decay | 1e-4 |
| Batch Size | 1 |
| Epochs | 500 |
| **Model** | |
| Number of Message Passing Layers | 4 |
| Hidden Units per Layer | $[4, 32]$ |
| Activation Function | SiLU |
| Invariant Normalization | False |
| Gradient Clipping | True |
| Dropout | $[0.025, 0.25]$ |

### I.3 GEOMETRIC FEATURES

We note that for both the molecular property prediction, and the hyperlocal air pollution downscaling task we are using coordinates as *geometric features*. Specifically, we employ 0-cell geometric features, which are inherited in the higher ranks. Regarding the molecular property prediction, we are using the 3D atom coordinates, while for the geospatial task, we are using the latitude and longitude variables.

**Computational Resources**  All experiments that produced the reported configurations in both tasks were conducted using Nvidia A100-SXM4-40GB GPUs. Each system has a capacity of 80GB RAM, 40GB GPU memory, and 6 CPUs per unit.

### J ABLATION STUDY AND REPRODUCIBILITY FOR MOLECULAR PROPERTY PREDICTION

In this section, we present the whole set of experiments we carried out to obtain our results in Table 1, i.e., all the different configurations of ETNN we tested. Moreover, we want to further specify some details that are useful to fairly evaluate our results and ensure reproducibility.

**Reproducibility.** When we started to set up the experiments, we found a poor reproducibility level for the molecular prediction task on QM9 in the literature. In particular, practically every model of the ones listed in Table 1 uses different data splits. The two closest models to ETNN are EGNN (Satorras et al., 2021) and EMPSN (Eijkelboom et al., 2023), which are also particular cases of our framework. We were not able to reproduce the results of EMPSN (and this is also the reason we did not include it in Table 1), while we were able to reproduce the EGNN results using the original codebase. For all of the reasons above, we decided to employ the same data splits as EGNN(Satorras et al., 2021). To guarantee a fair comparison with EMPSN, however, in this appendix we also report the results of EMPSN from the original paper, EMPSN reproduced in TopNets (Verma et al., 2024), and EMPSN reproduced with our codebase (as described in the main body). Again, and as explained in Appendix C.1, EGNN can be directly derived from ETNN. Before exploring different ETNN configurations, we aligned our training and architecture parameters with EGNN. We ensured that our ETNN code, configured as an EGNN, could reproduce all the considered properties of the QM9 dataset reported in the original EGNN paper. Moreover, we trained our models for 1000 epochs as in EGNN and EMPSN. Finally, to ensure further consistency, we also trained a configuration of ETNN, dubbed ETNN-graph-W, that is again EGNN derived from our codebase but with a higher number of hidden units, chosen such that the number of parameters of ETNN-graph-W matches the average number of parameters of the other ETNN configurations (1.5M, while the original EGNN has 748k).

**Results.** In Tables 11- 12, we present the full list of the ETNN configurations we tested, representing also an exhaustive ablation study on all the main components of our architectures. We also add the

results of TopNet, presented in the contemporary work (Verma et al., 2024). We grouped the relevant model hyperparameters of ETNN into the following five groups:

- *Definition of the Molecular CC.* We study the performance of ETNN when bonds, rings, and functional groups, are included or not as cells in the complex. In Tables 11- 12, this is described in the first argument, Lifters. The values of the argument indicate what are the considered cells, e.g. A+B+F+R means that atoms, bonds, functional groups, and rings are used as cells. The rank is assigned in increasing order as described in Section 5.

- *Adjacencies.* We study the performance of ETNN when different adjacencies are employed in the collection of neighborhood functions $\mathcal{CN}$. In Tables 11-12, this is described in the second argument, Adjacencies. In particular, we test the usage of $\mathcal{N}_{A,\max}$ as in (13) in conjunction with the up/down adjacencies $\mathcal{N}_{A,\uparrow}$ and $\mathcal{N}_{A,\downarrow}$ from (2). In Tables 11- 12, Adjacencies = max means that only the max adjacency from (13) is used, while Adjacencies = max,$\uparrow$,$\downarrow$ means that both the max adjacency from (13) and the up/down adjacencies from (2) are used (effectively resulting in a heterogeneous setting as described in the body of the paper).

- *Incidences.* We study the performance of ETNN when different incidences are employed in the collection of neighborhood functions $\mathcal{CN}$.In Tables 11- 12, this is described in the third argument, Incidences. Testing only the he up/down incidences $\mathcal{N}_{I,\uparrow}$ and $\mathcal{N}_{I,\downarrow}$ from (2) would always result in configurations in which cells of rank $k$ communicate only with cells of ranks $k-1$ and $k+1$. For this reason, we also test the $k$-up/down incidences, defined as:

$$\mathcal{N}_{I,\uparrow_k}(x) = \{y \in \mathcal{X} | \mathrm{rk}(y) = \mathrm{rk}(x) + k, x \subset y\},$$
$$\mathcal{N}_{I,\downarrow_k}(x) = \{y \in \mathcal{X} | \mathrm{rk}(y) = \mathrm{rk}(x) - k, y \subset x\}. \tag{30}$$

  In this way, cells can communicate in a more flexible way across ranks. In Tables 11- 12, Incidences = all means that the $k$-up/down incidences from (30) are used for all the possible $k$, i.e., each rank communicates with all the other ranks; Incidences = 1 means that the up/down incidences from (1) (equivalently, $k=1$ in (30)) are used; Incidences = 0 means that no incidences are used, i.e., there are no inter-rank message exchanges and the embeddings are aggregated only in the readout layer.

- *Higher-order Features.* We study the performance of ETNN when higher-order features (not only atom features) are employed. In Tables 11- 12, this is described in the fourth binary argument, HO Features. If HO Features = 1, higher-order features (bonds, ring and/or functional groups based on the complex) are fed to the model.

- *Virtual Cell.* We study the performance of ETNN when the virtual cell is employed. In Tables 11- 12, this is described in the fifth binary argument, Virtual Cell. If Virtual Cell = 1, the virtual cell is included in the complex.

***Remark 8 .*** In all the configurations, we don't allow the virtual cell to exchange messages or being included in the readout layer. The reason is that we mostly use it in the experiments to obtain a fully connected computational graph on each rank when it is used together with the max adjacency from (13). However, it remains an interesting object, as it is i) a straightforward way to include whole molecule-level features in ETNN by assigning them as features to the virtual cell, and ii) useful to improve expressiveness and generality of ETNN, as shown in Appendix F. Therefore, when Adjacencies = max and Virtual Cell=1, the resulting configuration is equivalent to not including the virtual cell in the complex and using the neighborhood function $\mathcal{N}_{A,\mathrm{all}}(x) = \{y \in \mathcal{X} | \mathrm{rk}(y) = \mathrm{rk}(x)\}$.

To provide the reader with some usage examples of ETNN employing the above hyperparameter grouping (that we follow in our codebase), EGNN as in the original paper (Satorras et al., 2021) can be obtained setting Lifters = A, Adjacencies = max, Incidences = 0, HO Features = 0, Virtual Cell = 1; an instance of EMPSN (Eijkelboom et al., 2023) can be obtained setting Lifters = A+B+R, Adjacencies = $\uparrow$, $\downarrow$, Incidences = 1, HO Features = 1, Virtual Cell = 0, and considering only rings of length up to three; an equivariant version of (Bodnar et al., 2021a) can be obtained setting Lifters = A+B+R, Adjacencies = $\uparrow$, $\downarrow$, Incidences = 1, HO Features = 1, Virtual Cell = 0, and considering rings of arbitrary length.

As the reader can notice from Tables  11- 12, several higher-order ETNN configurations reach near-SotA results and significant improvements over the reference EGNNs baselines, i.e., the original EGNN (Satorras et al., 2021) in the upper part of the tables, and its implementations ETNN-graph and ETNN-graph-W obtained through our codebase. ETNN-graph is the same exact architecture as the original EGNN but trained with gradient clipping and geometric invariants normalization.

Table 11: Mean Absolute Error for the first six molecular property predictions in QM9.

| Task
Units | $\alpha$
bohr$^3$ | $\Delta\varepsilon$
meV | $\varepsilon_{\text{HOMO}}$
meV | $\varepsilon_{\text{LUMO}}$
meV | $\mu$
D | $C_\nu$
cal/mol K |
|---|---|---|---|---|---|---|
| NMP | .092 | 69 | 43 | 38 | .030 | .040 |
| Schnet | .235 | 63 | 41 | 34 | .033 | .033 |
| Cormorant | .085 | 61 | 34 | 38 | .038 | .026 |
| L1Net | .088 | 68 | 46 | 35 | .043 | .031 |
| LieConv | .084 | 49 | 30 | 25 | .032 | .038 |
| DimeNet++* | .044 | 33 | 25 | 20 | .030 | .023 |
| TFN | .223 | 58 | 40 | 38 | .064 | .101 |
| SE(3)-Tr. | .142 | 53 | 35 | 33 | .051 | .054 |
| EGNN | .071 | 48 | 29 | 25 | .029 | .031 |
| EMPSN(Eijkelboom et al., 2023) | .066 | 37 | 25 | 20 | .023 | .024 |
| EMPSN* | .061 | 42 | 29 | 25 | .030 | .028 |
| EMPSN(Verma et al., 2024) | .066 | 51 | 32 | 25 | .031 | .027 |
| TopNet (VC, discrete)(Verma et al., 2024) | .083 | 47 | 37 | 24 | .035 | .032 |
| TopNet (VC, continuous)(Verma et al., 2024) | .075 | 49 | 36 | 27 | .030 | .035 |
| TopNet (RePHINE, discrete)(Verma et al., 2024) | .072 | 57 | 33 | 28 | .029 | .028 |
| TopNet (RePHINE, continuous)(Verma et al., 2024) | .070 | 50 | 35 | 25 | .032 | .030 |
| **ETNN (Configurations)**
[ Lifters \| Adjacencies \| Incidences \| HO Features \| Virtual Cell ] | | | | | | |
| 1: [A+B \| max \| 0 \| 1 \| 1 ] | .074 | 46 | 30 | 24 | .03 | .032 |
| 2: [A+B+F \| max \| 0 \| 1 \| 1 ] | .078 | 50 | 32 | 25 | .029 | .032 |
| 3: [A+B+R \| max \| 0 \| 1 \| 1 ] | .079 | 50 | 34 | 25 | .029 | .033 |
| 4: [A+B+F +R \| max \| 0 \| 1 \| 1 ] | .077 | 54 | 31 | 27 | .032 | .033 |
| 5: [A+B \| max \| 0 \| 0 \| 1 ] | **.062** | 48 | 27 | 25 | .03 | .033 |
| 6: [A+B \| max \| all \| 1 \| 1 ] | .07 | 47 | 26 | 23 | .023 | **.03** |
| 7: [A+B+F \| max \| all \| 1 \| 1 ] | .072 | 47 | 27 | 23 | .024 | .032 |
| 8: [A+B+R \| max \| all \| 1 \| 1 ] | .071 | 49 | 26 | 23 | .023 | .032 |
| 9: [A+B+F+R \| max \| all \| 1 \| 1 ] | .069 | 46 | **26** | 23 | .025 | .033 |
| 10: [A+B+F+R \| max, ↑,↓ \| all \| 1 \| 0 ] | .157 | 72 | 45 | 46 | .31 | .053 |
| 11: [A+B \| max \| 0 \| 1 \| 0 ] | .149 | 69 | 45 | 45 | .298 | .048 |
| 12: [A \| max \| 0 \| 0 \| 1 ] (ETNN-graph) | .07 | **45** | 28 | 23 | .029 | .035 |
| 13: [A \| max \| 0 \| 0 \| 1 ] (ETNN-graph-W) | .07 | 49 | 28 | 25 | .03 | .036 |
| 14: [A+B+F+R \| max \| all \| 1 \| 0 ] | .13 | 66 | 44 | 42 | .285 | .046 |
| 15: [A+B+R \| max \| all \| 1 \| 1 ] (triangles only, EMPSN-like) | .071 | 46 | 27 | 24 | **.022** | .032 |
| 16: [A+B+F+R \| max \| all \| 1 \| 1 ] (no geometric features, CCMPN-like) | .191 | 85 | 62 | 59 | .339 | .078 |
| 17: [A+B+F+R \| max \| all \| 1 \| 0 ] (no geometric features, CCMPN-like) | .184 | 86 | 61 | 59 | .351 | .075 |
| 18: [A+B \| max \| 1 \| 1 \| 1 ] | .069 | 47 | 26 | **22** | .024 | .03 |
| 19: [A+B+F \| max \| 1 \| 1 \| 1 ] | .067 | 44 | 27 | 23 | .023 | .032 |
| 20: [A+B+R \| max \| 1 \| 1 \| 1 ] | .069 | 46 | 26 | 24 | .023 | .033 |
| 21: [A+B+F+R \| max \| 1 \| 1 \| 1 ] | .075 | 45 | 27 | 23 | .024 | .034 |
| 23: [A+B+F+R \| max \| 1 \| 1 \| 0 ] | .139 | 68 | 41 | 43 | .281 | .046 |
| 25: [A+B+F+R \| max \| 1 \| 1 \| 1 ] (no geometric features, CCMPN-like) | .193 | 90 | 60 | 63 | .359 | .075 |
| 26: [A+B \| max, ↑,↓ \| 0 \| 1 \| 1 ] | .187 | 85 | 61 | 58 | .342 | .074 |
| 27: [A+B+F \| max, ↑,↓ \| 0 \| 1 \| 1 ] | 1.179 | 438 | 282 | 333 | .772 | .678 |
| 28: [A+B+R \| max, ↑,↓ \| 0 \| 1 \| 1 ] (equivariant CWN-like) | 1.143 | 352 | 262 | 262 | .713 | .628 |
| 29: [A+B \| max, ↑,↓ \| 0 \| 0 \| 1 ] | 1.038 | 332 | 237 | 284 | .721 | .548 |
| 30: [A+B \| max, ↑,↓ \| 0 \| 0 \| 1 ] | .984 | 267 | 209 | 229 | .657 | .496 |
| 31: [A+B \| max, ↑,↓ \| 1 \| 0 \| 1 ] | 1.224 | 410 | 251 | 307 | .712 | .848 |
| 32: [A+B+F \| max, ↑,↓ \| 1 \| 0 \| 1 ] | .141 | 74 | 50 | 48 | .312 | .049 |
| 33: [A+B+R \| max, ↑,↓ \| 1 \| 0 \| 1 ] | .143 | 75 | 48 | 47 | .337 | .048 |
| 34: [A+B+F+R \| max, ↑,↓ \| 1 \| 0 \| 1 ] | .129 | 69 | 44 | 45 | .297 | .046 |
| 35: [A+B+F+R \| max, ↑,↓ \| 1 \| 0 \| 0 ] | .134 | 71 | 46 | 45 | .302 | .046 |
| 36: [A+B+R \| max, ↑,↓ \| 1 \| 0 \| 0 ] (triangles only, heterogeneous EMPSN-like) | .132 | 71 | 45 | 45 | .309 | .045 |
| 37: [A+B+R \| max, ↑,↓ \| 1 \| 1 \| 0 ] (no geometric features, CWN-like) | .145 | 75 | 49 | 49 | .324 | .049 |
| 38: [A+B+F+R \| max, ↑,↓ \| 1 \| 0 \| 0 ] (no geometric features, CCMPN-like) | .184 | 90 | 62 | 62 | .371 | .077 |
| 39: [A+B+F+R \| max, ↑,↓ \| 1 \| 0 \| 0 ] (no geometric features, CCMPN-like) | .19 | 92 | 62 | 60 | .36 | .076 |
| **Best ETNN Configuration** | | | | | | |
| ETNN-* | .062 | 45 | 26 | 22 | .022 | .030 |
| Improvement over EGNN | **-13%** | **-6%** | **-10%** | **-12%** | **-26%** | **-3%** |

As mentioned at the beginning of this section, we made sure that ETNN-graph without gradient clipping and normalization almost exactly reproduces the results of the original EGNN. We report EGNN-graph because it is sometimes able to reach better results than the original EGNN on some targets. Moreover, some configurations that we test are also instances of other architectures like EMPSN (Eijkelboom et al., 2023) or CWN (Bodnar et al., 2021a), further highlighting the versatility of our framework. Notably, ETNN consistent outperforms TopNet(Verma et al., 2024) too. Overall, these results validate our TDL approach and show that the flexibility of CCs is necessary.

## K ARCHITECTURE DESCRIPTION

In this section, we describe the implementation of E(n) Equivariant Topological Neural Networks (ETNNs). The model architecture is kept as similar to EGNN as possible so as to not introduce

Table 12: Mean Absolute Error for the last five molecular property predictions in QM9.

| Task Units | $G$ meV | $H$ meV | $U$ meV | $U_0$ meV | ZPVE meV |
|---|---|---|---|---|---|
| NMP | 19 | 17 | 20 | 20 | 1.50 |
| Schnet | 14 | 14 | 19 | 14 | 1.70 |
| Cormorant | 20 | 21 | 21 | 22 | 2.03 |
| L1Net | 14 | 14 | 14 | 13 | 1.56 |
| LieConv | 22 | 24 | 19 | 19 | 2.28 |
| DimeNet++* | 8 | 7 | 6 | 6 | 1.21 |
| TFN | - | - | - | - | - |
| SE(3)-Tr. | - | - | - | - | - |
| EGNN | 12 | 12 | 12 | 11 | 1.55 |
| EMPSN(Eijkelboom et al., 2023) | 6 | 9 | 7 | 10 | 1.37 |
| EMPSN* | 11 | 11 | 10 | 10 | 1.46 |
| EMPSN(Verma et al., 2024) | - | - | - | - | 1.44 |
| TopNet (VC, discrete)(Verma et al., 2024) | - | - | - | - | 1.45 |
| TopNet (VC, continuous)(Verma et al., 2024) | - | - | - | - | 1.43 |
| TopNet (RePHINE, discrete)(Verma et al., 2024) | - | - | - | - | 1.38 |
| TopNet (RePHINE, continuous)(Verma et al., 2024) | - | - | - | - | 1.37 |
| **ETNN (Configurations)** | | | | | |
| **[ Lifters | Adjacencies | Incidences | HO Features | Virtual Cell ]** | | | | | |
| 1: [A+B | max | 0 | 1 | 1] | 15 | 13 | 14 | 13 | 1.75 |
| 2: [A+B+F | max | 0 | 1 | 1] | 14 | 12 | 12 | 12 | 1.733 |
| 3: [A+B+R | max | 0 | 1 | 1] | 15 | 15 | 14 | 14 | 1.763 |
| 4: [A+B+F +R | max | 0 | 1 | 1] | 15 | 15 | 15 | 15 | 1.749 |
| 5: [A+B | max | 0 | 0 | 1] | 13 | 12 | 12 | 12 | **1.589** |
| 6: [A+B | max | all | 1 | 1] | **11** | 12 | 12 | **11** | 1.648 |
| 7: [A+B+F | max | all | 1 | 1] | 12 | 12 | 12 | 12 | 1.66 |
| 8: [A+B+R | max | all | 1 | 1] | 11 | 13 | **11** | 13 | 1.668 |
| 9: [A+B+F+R | max | all | 1 | 1] | 12 | 12 | 13 | 11 | 1.742 |
| 11: [A+B | max | 0 | 1 | 0] | 26 | 27 | 27 | 27 | 2.275 |
| 12: [A | max | 0 | 0 | 1] (ETNN-graph) | 15 | 15 | 14 | 16 | 1.658 |
| 13: [A | max | 0 | 0 | 1] (ETNN-graph-W) | 14 | 14 | 14 | 13 | 1.624 |
| 14: [A+B+F+R | max | all | 1 | 0] | 23 | 24 | 24 | 24 | 2.278 |
| 15: [A+B+R | max | all | 1 | 1] (triangles only, EMPSN-like) | 12 | 12 | 11 | 12 | 1.703 |
| 16: [A+B+F+R | max | all | 1 | 1] (no geometric features, CCMPN-like) | 32 | 34 | 35 | 32 | 3.268 |
| 17: [A+B+F+R | max | all | 1 | 0] (no geometric features, CCMPN-like) | 32 | 34 | 34 | 34 | 3.202 |
| 18: [A+B | max | 1 | 1 | 1] | 12 | 12 | 11 | 13 | 1.65 |
| 19: [A+B+F | max | 1 | 1 | 1] | 13 | 14 | 12 | 13 | 1.644 |
| 20: [A+B+R | max | 1 | 1 | 1] | 12 | 13 | 12 | 11 | 1.714 |
| 21: [A+B+F+R | max | 1 | 1 | 1] | 12 | **12** | 13 | 11 | 1.694 |
| 23: [A+B+F+R | max | 1 | 1 | 0] | 24 | 24 | 24 | 24 | 2.183 |
| 25: [A+B+F+R | max | 1 | 1 | 1] (no geometric features, CCMPN-like) | 32 | 35 | 35 | 34 | 3.361 |
| 26: [A+B+F+R | max | 1 | 1 | 0] | 32 | 35 | 34 | 34 | 3.183 |
| 27: [A+B | max, ↑,↓ | 0 | 1 | 1] | 427 | 432 | 434 | 431 | 17.279 |
| 28: [A+B+R | max, ↑,↓ | 0 | 1 | 1] (equivariant CWN-like) | 367 | 371 | 376 | 373 | 16.298 |
| 29: [A+B+R | max, ↑,↓ | 0 | 1 | 1] | 302 | 302 | 302 | 302 | 13.019 |
| 30: [A+B | max, ↑,↓ | 0 | 0 | 1] | 236 | 238 | 237 | 236 | 12.175 |
| 31: [A+B | max, ↑,↓ | 0 | 0 | 1] | 477 | 491 | 485 | 484 | 25.588 |
| 32: [A+B | max, ↑,↓ | 1 | 0 | 1] | 29 | 30 | 29 | 30 | 2.369 |
| 33: [A+B+F | max, ↑,↓ | 1 | 0 | 1] | 28 | 29 | 28 | 30 | 2.227 |
| 34: [A+B+R | max, ↑,↓ | 1 | 0 | 1] | 23 | 24 | 25 | 23 | 2.165 |
| 35: [A+B+F+R | max, ↑,↓ | 1 | 0 | 1] | 25 | 25 | 25 | 24 | 2.178 |
| 36: [A+B+R | max, ↑,↓ | 1 | 0 | 0] (triangles only, heterogeneous EMPSN-like) | 25 | 26 | 25 | 25 | 2.215 |
| 37: [A+B+R | max, ↑,↓ | 1 | 0 | 0] (no geometric features, CWN-like) | 27 | 28 | 27 | 27 | 2.222 |
| 38: [A+B+R | max, ↑,↓ | 1 | 1 | 0] (no geometric features, CCMPN-like) | 34 | 34 | 34 | 34 | 3.154 |
| 39: [A+B+F+R | max, ↑,↓ | 1 | 0 | 0] (no geometric features, CCMPN-like) | 32 | 33 | 33 | 32 | 3.148 |
| **Best ETNN Configuration** | | | | | |
| ETNN-* | 11 | 12 | 11 | 11 | 1.589 |
| Improvement over EGNN | **-8%** | **0%** | **-8%** | **0%** | **2.5%** |

alternative explanations for differences in model performance. The main components of ETNNs are as follows:

**Model Initialization**:

- *Invariant Normalizer*: Normalizes invariant features dynamically using batch normalization with no learnable parameters. Each rank is normalized separately.

- *Feature Embedding*: Embeds input features into a hidden representation. Input features of each rank are embedded with a separate featurizer.

- *Equivariant Message Passing (EMP) Layers*: Repeatedly updates the hidden representations of each cell using messages from neighboring cells. For details, see below.

- *Pre-Pool Layer*: Updates the hidden representation of each cell using distinct MLPs for each rank.

- *Post-Pool Layer*: MLP applied after the global pooling operation to produce the final output.

**Forward Pass**:

1. *Input Handling*:

   - Retrieves pre-computed non-geometric cell features, adjacency relationships, and positional information.

2. *Initial Feature Computation*:

   - Computes node and membership features for each rank. Cells with higher rank may either use their own non-geometric features, the mean feature vector of the nodes the cell is composed of, a membership vector denoting the lifter that generated the cell, or any combination of these.

   - Embeds the chosen *non-geometric* features into a common hidden representation.

3. *Geometric Invariant Computation*:

   - Computes E(n) invariant geometric features using the absolute positions of nodes and the specified `compute_invariants` function.

   - Optionally normalizes the computed invariant features.

4. *Message Passing*:

   - Applies *EMP* layers to update cells' hidden represntations via message passing. For details on message passing, see below.

5. *Readout and Pooling*:

   - Applies pre-pooling MLPs to the hidden representations of cells. Each rank is processed by a different MLP.

   - If the task is graph-level prediction, hidden representations of cells of same rank are added together to arrive at *rank representations*. These representations are concatenated and a post-pooling MLP is applied to generate the final output.

**Equivariant Message Passing (EMP) Layers**: The computation of the EMP layer takes place in the following steps:

1. *Computation of individual messages*:

   - For each cell, a message is computed for each of its parents. This message is computed by an *adjacency-specific MLP* that takes as input the hidden representations of the sending and receiving cells, and the E(n) invariant geometric features derived from the absolute positions of the nodes they are composed of. A separate MLP is defined for each rank pair.

2. *Adjacency-wise aggregation of computed messages*:

   - For any given cell, the set of messages it received from its neighbors are grouped according to the rank of the sender. The messages within each group are aggregated using a weighted sum of the messages, where the weights are the nonnegative, scalar outputs of an adjacency-specific MLP that takes as input a message.

3. *Cell representation update*:

   - Finally, an update is computed by feeding the concatenation of the adjacency-wise aggregated messages to a *rank-specific update MLP*. This update is added to the cell's hidden representation.

4. *(Optional) position update*:

  - Optionally, node positions are updated as a weighted sum of differences between the position of the node and its neighboring nodes. The weight is a learnable function of the message that was passed to the node from its neighbor.

