# OpenReview forum: "E(n) Equivariant Topological Neural Networks"
_ICLR.cc/2025/Conference — ICLR 2025 Poster_

### Official Review · Reviewer_J6CW · 2024-10-30

**Soundness:** 3
**Presentation:** 2
**Contribution:** 3
**Rating:** 6
**Confidence:** 4

**Summary:**

**Method:** The paper introduces a framework, termed ETNN, for processing combinatorial complexes with geometric cell features in an $\mathrm{E}(n)$-equivariant way (explicitly introduces a method for $0$-cell geometric features and explains how to generalize to geometric features of cells of arbitrary rank). ETNNs are equivariant to the action of both $\mathrm{Sym}(\mathcal{X})$ (cell renaming) and $\mathrm{E}(n)$ isometries acting on the geometric features. The architecture builds on higher-order message-passing over CCs and has both invariant and equivariant versions. ETNNs generalizes previous work on $\mathrm{E}(n)$-equivariant graph neural networks, and $\mathrm{E}(n)$-equivariant cellular complex networks.

**Theory:** The expressivity of the proposed methods is evaluated based on its ability to distinguish $k$-hop distinct geometric graphs. The paper proves that in *most cases* ETNN is strictly more expressive than baseline geometric graph methods (i.e. message-passing architectures on geometric graphs that do not operate on higher-order cells).

**Experiments:**
- **QM9:** The first application considered in the paper is QM9 molecular property prediction, where molecules are represented as combinatorial complexes with geometric features. ETNN variants demonstrate clear performance increase over standard geometric graph methods.
- **Air Pollution Benchmark:** The second application introduces a new benchmark for air pollution prediction. CCs are constructed from point measurements (0-cells), road measurements (1-cells), and census data (2-cells).

**Notes.**
- Typo in equation (13); should be "$x \subset z$, and $y \subset z$".

**Strengths:**

- **Framework and Theoretical Contributions:**
  - The ETNN architecture introduced in the paper is a novel framework for $\mathrm{E}(n)$-equivariant processing of a wide class of topological data objects.
  - The framework elegantly unifies and generalizes existing architectures -- e.g. EGNN [1] and EMPSN [2].
  - The paper theoretically proves expressivity improvements over $\mathrm{E}(n)$-equivariant graph methods.

- **Empirical Results:**
  - ETNN variants achieve clear performance gains over standard $\mathrm{E}(n)$-equivariant graph methods for QM9 molecular property prediction tasks; ETNNs also improves upon EMPSNs while using less memory and having a faster runtime.

- **Applications and Dataset Contributions:**
  - The paper introduces a principled approach to modeling irregular multi-resolution geospatial data using combinatorial complexes.
  - The *air pollution downscaling* benchmark introduced in the paper is a new dataset for benchmarking TDL architectures. Additionally, the construction and analysis of geospatial combinatorial complexes is in itself an interesting contribution of the paper.

[1] Satorras et al. "E(n) equivariant graph neural networks", ICML 2021.

[2] Eijkelboom et al. "E(n) equivariant message passing simplicial networks", ICML 2023.

**Weaknesses:**

- **Novelty:** While the proposed method generalizes previous work to general combinatorial complexes, the architectural changes are incremental.

- **Theoretical analysis:**
  - The expressivity analysis focuses solely on distinguishing geometric *graphs*, despite defining expressive power in terms of separating more general non-isomorphic *CTSs*.
  - Proposition 2's statement is imprecise: the claim of improved expressivity "in most cases" lacks formal qualification, and the relationship between the expressivity gap and choice of lifting method needs clearer formulation and concrete characterization. A clearer restatement would be helpful.
  - No formal comparison to expressivity of other TDL methods (e.g. equivariant simplicial networks).

- **Empirical evaluation:**
   - Results lack statistical significance analysis (no standard deviations reported).
  - On the air pollution benchmark, improvements over the MLP baseline are modest (~1.5% RMSE reduction) and their statistical significance cannot be assessed due to unreported standard deviations.

**Questions:**

- Can the authors clarify the motivation for using "k-hop distinct" graphs as the primary measure of expressivity? Is it possible extend the expressivity results to CCs? E.g. is it possible to define a notion of "k-hop distinctness" for geometric combinatorial complexes and analyze ETNN's ability to distinguish between such complexes?
- How does ETNN's expressivity compare to that of simplicial complex networks in distinguishing non-isomorphic geometric graphs/complexes?
- In proposition 2, can you specify the conditions under which ETNNs are provably more expressive than EGNNs?
- Could you include standard deviations for the air pollution benchmark and QM9 results to assess statistical significance?

---

> ### Author Response · Authors · 2024-11-16
> **Addressing Reviewer's comments and questions**
>
> We’d like to thank the Reviewer for their feedback, as well as the raised concerns and questions. Also, we’d like to thank them for identifying as strengths of the paper both the framework and the theoretical contributions, the empirical results, as well as the applications and the novel dataset. Next, we address each raised weakness in detail, and also point them out as answers to their questions.
>
> ## Weaknesses.
> **[W1] Novelty**
>
> On the one hand, it is true that ETNNs are scalarization-based architectures, as EGNNs [1], on higher-order combinatorial domains, as EMPSNs [2]. As such, they inevitably resound with each other.
> However, we kindly disagree with the reviewer’s comment since ETNNs are a formal and more expressive generalization of both EMPSNs and EGNNs. For this reason, ETNNs unlock several features (e.g. arbitrary modeling of heterogeneous hierarchical higher-order interactions, tunable expressive power, and general learnable geometric invariants) that the constrained graph or simplicial structures of EGNNs and EMPSNs cannot accommodate (see Appendix C and F). As such, our framework can be used to design arbitrary E(n) equivariant TDL models. No other framework has the same power.
>
> Moreover, although we believe that the generality of our framework is fundamental, we also recognized the necessity for applications whose exhaustive modeling (in a combinatorial/topological sense) is possible only via combinatorial complexes, and we introduced MolecularCCs and GeospatialCCs to tackle this problem (no other complex nor graph can model molecular and multi-resolution irregular geospatial data as straightforwardly as CCs). As a consequence, we achieved or matched SotA results among the Equivariant TDL models with a huge reduction in computational complexity.  As a byproduct, as reviewer NQcw noticed too, our air pollution downscaling task represents a novel benchmark for the TDL community, addressing a need highlighted in the recent position paper [3].
>
> Finally, we believe that the expressivity analysis is novel. In particular, our approach is, to the best of our knowledge, currently the most exhaustive one for scalarization-based equivariant TDL models, as it could be applied to analyze the (geometric WL) expressivity of any scalarization-based equivariant TDL model without the need for domain-specific coloring procedures.
>
> **[W2] Expressivity on CTSs**
>
> Our choice of studying expressivity in terms of distinguishing geometric graphs (empirically, we have the counterexamples experiment too, not only the $k$-hop distinct graphs experiment) is due to 3 main reasons:
>
> (i) Equivariant TDL is an extremely recent (sub-)field, thus it made sense to us to first show that ETNNs are more expressive than scalarization-based graph models (i.e., EGNNs). This is something already new, as neither [2] nor [4] show any expressivity result.
>
> (ii) To achieve (i), we wanted to rely on strong and recognized theoretical tools, and the Geometric WL (and related tasks, e.g., $k$-hop distinct graphs and counterexamples) looked like a natural choice.
>
> (iii) The approach we adopt to port the Geometric WL machinery in our setting is itself completely novel and represents a versatile and general approach for studying the (geometric WL) expressivity of any scalarization-based equivariant TDL model that can be derived from the ETNN framework.
>
> This said, we undoubtedly agree with the reviewer's comment. Having a higher-order geometric WL  (and related tasks, e.g., $k$-hop distinct complexes) to distinguish among geometric realizations of CTSs is one of the most interesting directions. However, adapting the entire topological-/k-WL machinery to the geometric setting is a highly non-trivial task, and situating it within the context of existing literature adds further complexity. We believe that addressing these challenges fully would necessitate a separate dedicated paper.
>
>
> **[W3] Statement of Proposition 2**
>
> The reviewer is correct that we need to refine the statement to make it clearer. In the revised manuscript, we clearly state the condition of improved expressivity induced by **Proposition 2** in the updated **Corollary 1**. In particular, an ETNN is strictly more powerful than an EGNN in distinguishing $k$-hop distinct graphs if the skeleton-preserving lifting is such that the number of ETNN layers required to have at least one cell in  $X_{G_1}$/$X_{G_2}$ whose receptive field is the whole set of nodes in $G_1$/$G_2$ is smaller than the number of EGNN layers required to have at least one node in  $G_1$/$G_2$ whose receptive field is the whole set of nodes in $G_1$/$G_2$.
>
> *Continuing response in the thread (References are included in the last part of the thread)*

---

> ### Author Response · Authors · 2024-11-16
>
> **[W4]  ETNN vs EMPSN expressivity**
>
> As a consequence of what we wrote in our reply to **[W1](iii)**, a comparison is not required, because CCs generalize Cellular/Simplicial Complexes (SCs) and our approach based on the novel notion of *Geometric Augmented Hasse Graph* can thus be directly applied to SCs. As we carefully explained in the paper, an expressivity comparison (in the geometric WL sense) is fair only when the lift is skeleton-preserving, i.e. when the underlying graph connectivity is preserved; in [2], they do not lift the graphs (and they do not provide any expressivity analysis), but rather they replace the connectivity with a Vietori-Rips SC. Although one can indeed lift a graph into a simplicial complex preserving the graph connectivity by taking triangles or cliques as higher-order simplices, the expressivity would be characterized exactly as for **Proposition 2** (and as in [W3] to compare with GNNs).
>
> **[W5] Statistical Significance on QM9**
>
> At the beginning of **Appendix J**, we exhaustively explain why we don’t run the model on multiple splits (each one of the baseline models uses different splits) and, in general, our meticulous care for reproducibility and fairness. Practical examples of this are also the fact that, despite its irreproducibility, we cared a lot about a fair comparison with EMPSN, or with an EGNN having our same parameter budget. Overall, for the QM9 benchmark, we followed the established experimentation setup of this specific dataset, following prior works, in which authors reported their best performance results and we were completely transparent about it (see **Appendix J** and **I**). This can be resonated by the large size of the molecular dataset, which makes multiple iterations of experiments challenging. In this sense,  ETNN sets an unprecedented standard in the TDL community regarding exhaustiveness and transparency in the presentation of the results.
>
> **[W6] Statistical Significance on Air Pollution Downscaling**
>
> We originally did not report variances since our experiments are averaged over 3 seeds, which makes variance calculations unreliable. We are now working on running with additional seeds and will report the results when the experiments are complete.
>
> ## Questions.
> **[Q1] K-hop distinct graphs**
>
> Please see **[W2]**.
>
> **[Q2] ETNN vs EMPSN expressivity**
>
> Please see **[W4]**.
>
>
> **[Q3] Statement of Proposition 2**
>
> Please see **[W3]**
>
> **[Q4] Statistical Significance**
>
> Please see **[W5]-[W6]**.
>
> ## References
>
> [1] Víctor Garcia Satorras, Emiel Hoogeboom, and Max Welling. "E(n) Equivariant Graph Neural Networks." In *International Conference on Machine Learning*, pages 9323–9332. PMLR, 2021.
>
> [2] Floor Eijkelboom, Rob Hesselink, and Erik J Bekkers. "E(n) Equivariant Message Passing Simplicial Networks." In *International Conference on Machine Learning*, pages 9071–9081. PMLR, 2023.
>
> [3] Papamarkou, Theodore, et al. "Position: Topological Deep Learning is the New Frontier for Relational Learning." *Forty-first International Conference on Machine Learning*, 2024.
>
> [4] Liu, Cong, et al. "Clifford group equivariant simplicial message passing networks." ICLR 2024.

---

> > ### Comment · Reviewer_J6CW · 2024-11-18
> >
> > I Thank the authors for their thoughtful and detailed response. A few followup questions/comments:
> >
> > - **Re: architectural novelty:** ETNNs are a direct generalization of EMPSNs. Apart from the fact that the input to an ETNN model is a signal defined over a combinatorial complex (CC) and the input to an EMPSN model is a signal defined over a simplicial complex (SC), what is the difference between their forward passes? That is, could I have used an EMPSN model to process signals on a CC?
> >
> >   While I understand that applying this architecture to a more general data object is interesting and allows for experimental flexibility (and that is a contribution of the paper), I’m still trying to understand the architectural modifications needed in order to do that.
> >
> > - **Re: Expressivity on CTSs:** I recognize the difficulty of analyzing expressivity in a new (sub)field, but I believe this point is important and worth further investigation. Since the premise of the paper is that operating on CCs with geometric features is useful, it’s important to understand the expressive power *in this domain specifically*. Even without a higher order geometric WL hierarchy, a concrete example of CCs with geometric features that ETNNs can/cannot distinguish (provably) is valuable.
> >
> > - **Re: expressivity comparison to equivariant TDL methods:** While ETNNs and e.g. EMPSNs operate on different objects, both objects (CCs and SCs) can be lifted from (geometric) graphs. In the graph expressivity sense, is lifting to a CC and using ETNN better than lifting to a SN and using EMPSN?
> >
> > - **Re: statistical significance of experimental results:** I see that other baselines don’t report std results on QM9 either. While I still believe this is problematic I accept the authors’ answer. As to the air pollution benchmark, standard deviation results (even on 3 seeds, but ideally more) would be appreciated.

---

> > > ### Author Response · Authors · 2024-11-18
> > >
> > > We would like to thank the reviewer for the swift response, allowing us to have an engaging discussion. Next, we address the four points they raised:
> > >
> > > **[W1]** Architectural novelty: ETNNs are a direct generalization of EMPSNs.
> > >
> > > In mere terms of forward pass (which, although an important component, represents just one aspect of our more comprehensive work),  EMPSN forward cannot be used to process signals on a CC. The main reasons concern the two most critical components of a scalarization-based equivariant architecture, i.e., (1) Feature updates and (2) Geometric invariants.
> > >
> > > Regarding (1), SCs induce specific neighborhood functions (Boundary, Coboundary, Up/Down Adjacencies), thus if a CC comes with a non-standard collection of neighborhoods, EMPSN forward cannot handle it, both for non-geometric and geometric features updates.
> > >
> > > Regarding (2), EMPSN forward uses tailored geometric invariants for SCs. Computing the volume of a non-simplex makes no sense. Similarly, pruning the pairwise distances based on the simplicial structure or using the angles of the planes induced by the dimension of the simplices makes no sense in the CC case.
> > >
> > > Please notice that an ETNN forward can be used to process signals on an SC, as all the geometric invariants tailored for SCs and the neighborhoods induced by an SC can be directly used in our framework.
> > >
> > >
> > > **[W2]** Expressivity on CTSs.
> > >
> > > Thanks for clarifying the question. In fact, we have already discussed an example where the lifting of $k$-chain graphs would result in two undistinguishable (geometric) CCs for ETNN. In particular, in **Remark 6**, we explain that in Lift 3a, using the up adjacency from (2) without including the edges as cells, would prevent solving the task. This is because it would result in a disconnected geometric augmented Hasse graph, since $p_1$/$p_2$ and $o_1$/$o_2$ would not have been linked. We are more than happy to enhance this part more in the revised manuscript, highlighting that failing to solve the task is equivalent to proving that there exists a collection of neighborhoods and a pair of geometric CCs that are undistinguishable by ETNN.
> > >
> > > **[W3]** Expressivity comparison to equivariant TDL methods.
> > >
> > > Yes, lifting to a CC and using ETNN is indeed better than lifting to a SC and using EMPSN. We already showed this indirectly in the paper. In particular, on both the numerical expressivity tasks that we show, the $k$-chain graphs (**Table 3**) and the counterexamples (**Table 4**), EMPSN with any skeleton preserving lifting would be as expressive as an EGNN (and, thus, way less expressive than an ETNN), as there are no triangles nor cliques. We will add a remark in the paper to make this clearer.
> > >
> > > **[W4]** Statistical significance of experimental results.
> > >
> > > We thank the reviewer for their understanding regarding QM9, and we hope that our particular focus on transparency and honesty in the description of our experimental setup (**Appendices I** and **J**) can compensate for this partially problematic habit of the community.
> > >
> > > We are currently running the geospatial experiment on a significantly higher number of seeds and we will post the results as soon as the runs are done.

---

> > > ### Author Response · Authors · 2024-11-21
> > >
> > > ​​**Standard errors for the geospatial task and additional seeds**
> > >
> > > We have conducted multiple additional runs for the geospatial task based on your suggestion. We ran 10x more seeds (30 in total). Remarkedly, the results are very similar to those reported in the paper despite the high variance reflected by the standard error,
> > >
> > > *Updated tables*
> > >
> > > | Baseline | Av. Explained Variance ($R^2$) | Std. Err ($R^2$) | **MSE** |
> > > | -------- | ------------------------------ | ---------------- | ------- |
> > > | ETNN     | 9.34%                           | 2.05%             | 0.935    |
> > > | GNN      | 2.44%                           | 0.99%             | 0.987    |
> > > | MLP      | 2.35%                           | 1.61%             | 1.022    |
> > > | EGNN     | 1.43%                           | 1.2%             | 1.041    |
> > > | Linear   | 0.51%                           | 0.95%             | 1.106   |
> > > **Table 2a**. Baseline model comparison
> > >
> > > | Baseline                      | Diff. Explained Variance ($R^2$) | Std. Err  ($R^2$) | **MSE** |
> > > | ----------------------------- | -------------------------------- | ---------------- | ------- |
> > > | no geometric features (CCMPN) | -1.08%                            | 2.05%             | 0.946    |
> > > | no position update (invariant ETNN)     | -1.37%                            | 2.52%             | 0.956    |
> > > | no virtual node               | -1.80%                            | 2.32%             | 0.957    |
> > > |                               |                                  |                  |         |
> > > **Table 2b**. Ablation study

---

> > > > ### Author Response · Authors · 2024-11-22
> > > > **Follow-up: Did our responses address the Reviewer's questions?**
> > > >
> > > > We’d like to follow up to ask if our responses sufficiently addressed the Reviewer’s comments.
> > > >
> > > > Also, we'd like to ask them if the extra experiments we conducted to provide the statistical measures (as shown in the tables above) meet expectations or if further experiments are needed.
> > > > Otherwise, in case the Reviewer is satisfied with our response and the clarifications, we'd greatly appreciate it if they could reconsider the rating of our submission.

---

> ### Comment · Reviewer_J6CW · 2024-11-24
> **Addressing authors' clarifications**
>
> I thank the reviewers for thorough responses, which have strengthened my understanding of this work. The paper presents good contributions that merit publication, and I'd like to address the clarifications while explaining why I maintain my current score.
>
> **[W1]** The authors' response clarifies the key differences between ETNNs and EMPSNs, particularly regarding neighborhood functions and geometric invariants. While I now have a clearer technical understanding of these differences, I view them as adaptations for handling CCs rather than fundamental architectural innovations. As I noted in my review, this contribution is still valuable – I simply maintain this as a non-detrimental weakness.
>
> **[W2]** The authors' clarification about Remark 6 and the example of undistinguishable geometric CCs is helpful, and I think the paper would benefit from highlighting it in the main text. However, two points remain: (a) while instructive, this specific example doesn't easily generalize, and it’s unclear how we can establish a more comprehensive expressivity hierarchy, and (b) it demonstrates one direction (indistinguishable geometric CCs) without fully characterizing which geometric CC classes can be distinguished. I again want to stress that this is a non-detrimental weakness and that, as I pointed out in my review, the paper's theoretical contributions are valuable.
>
> **[W3]** I understand the authors' argument re: the expressivity comparison (for $k$-chains and counterexamples) through Tables 3 and 4 + the fact that EMPSN with skeleton-preserving lifting would be limited to EGNN expressivity in those cases. This is a subtle argument that mixes empirical and theoretical observations. A formal theoretical characterization would be valuable in future work.
>
> **[W4]** The additional seeds for the geospatial experiment strengthen the empirical validation and are a good precedent for the benchmark.
>
> In conclusion, I maintain my positive assessment of the paper. The paper makes valuable contributions to equivariant topological deep learning, particularly through its practical innovations and empirical results. While the noted limitations prevent me from increasing my score, they don't detract from the paper's value and publishability.

---

> > ### Author Response · Authors · 2024-11-24
> >
> > We thank the reviewer for their engagement, further feedback, and for agreeing that our paper deserves to be published. We deeply appreciate the time the reviewer dedicated to our work. Here are some further comments.
> >
> >
> > **[W1]**  We thank the reviewer for clarifying their point. Overall, it is undoubtedly true that ETNN is a simple equivariant generalization of CCMPNs, similar to  EMPSN  being a simple generalization of MPSNs, and EGNN  being a simple generalization of MPNs. Moreover, it is also true that EMPSN resonates with ETNN  being both scalarization-based equivariant TDL models. However, we showed that ETNNs are **more general and flexible**, **more expressive**, and **similar or better performing while being hugely more scalable** than SotA methods in the same class. These facts, together with the **benchmark**, **software**, and **outreaching** contribution of our work, should motivate its publication, as the reviewer agrees. This said, we already wrote in the future directions section that beyond scalarization-based architectures are of great interest. However, given that **a gap in the literature was still present for scalarization-based models** too, it made sense to us to **first comprehensively fill it** by working in the direction of ETNN.
> >
> >
> > **[W2]** We will highlight it in the next revision of the paper we will upload in the next couple of days. Moreover, we agree with the reviewer, that carrying out an in-depth study of which are the distinguishable classes of CCs beyond showing the existence of undistinguishable pairs of CCs is definitely an interesting direction. However, given its complexity and relevance, it would require a standalone work (that builds on top of our expressivity analysis).
> >
> >
> > **[W3]** We agree with the reviewer (and this is somehow also linked to [W2]), although we believe that the generality of Proposition 2 is already a valuable theoretical tool to compare the two models.
> >
> > Thanks again!

---

> > > ### Author Response · Authors · 2024-11-28
> > > **Follow-up after deadline extension**
> > >
> > > In light of the extension of the discussion deadline, we would like to thank the reviewer again for the already positive assessment of our work and for their engagement, and ask if there is anything else we can do to further improve the manuscript and its score.
> > >
> > > We also kindly invite the reviewer to parse the discussions we had with the other reviewers, as we believe that the overall engagement and the individual feedback have clarified or improved several aspects of the paper.
> > >
> > > Thanks a lot!

---

### Official Review · Reviewer_wwvW · 2024-10-31

**Soundness:** 3
**Presentation:** 3
**Contribution:** 2
**Rating:** 6
**Confidence:** 4

**Summary:**

This paper extends message passing supported on geometric graphs [3] and geometric simplicial complexes [2] to the more general setting of geometric combinatorial complexes. Theoretically, similar to how [1] establishes that higher-order message passing is more expressive than standard graph message passing, this paper demonstrates that the same holds true in the geometric setting. The paper also highlights the effectiveness of the proposed architecture through two novel real-world applications: (1) property prediction over geometric graphs representing molecules, where higher-order cells consist of rings and active groups, and (2)  a regression task over geospatial combinatorial complexes. The proposed architecture outperforms previous geometric graph architectures on these tasks.


[1] Cristian Bodnar, Fabrizio Frasca, Nina Otter, Yuguang Wang, Pietro Lio, Guido F Montufar, and Michael Bronstein. Weisfeiler and lehman go cellular: Cw networks. Advances in neural information processing systems, 34:2625–2640, 2021.

[2]  Floor Eijkelboom, Rob Hesselink, and Erik J Bekkers. E (n) equivariant message passing simplicial networks. In International Conference on Machine Learning, pages 9071–9081. PMLR, 2023.

[3] Vıctor Garcia Satorras, Emiel Hoogeboom, and Max Welling. E (n) equivariant graph neural networks. In International conference on machine learning, pages 9323–9332. PMLR, 2021.

**Strengths:**

1. The paper offers a simple and straightforward way to adapt higher order message passing to respect $O(d)$ symmetries.

2. The experimental section effectively demonstrates the architecture's performance and introduces two new, interesting real-world TDL benchmarks, addressing a need highlighted in a recent position paper [1].



[1] Theodore Papamarkou, Tolga Birdal, Michael M Bronstein, Gunnar E Carlsson, Justin Curry, Yue Gao, Mustafa Hajij, Roland Kwitt, Pietro Lio, Paolo Di Lorenzo, et al. Position: Topological deep learning is the new frontier for relational learning. In Forty-first International Conference on Machine Learning, 2024.

**Weaknesses:**

1. The novelty of the proposed architecture and the theoretical section is somewhat limited. The architecture closely resembles those presented in [4] and [2]. Additionally, the theoretical contributions feel somewhat straightforward, offering limited new insights.

2. Building on the previous comment about the theoretical section, the paper lacks an analysis of how the choice of invariant function (see Equation (6)) affects the architecture's expressivity. It would be valuable to examine whether simpler, computationally efficient invariant functions could result in architectures that are  as expressive as architectures which use more complex alternatives, thereby guiding the choice of which invariant function to use in practice. Additionally, it would have been insightful to analyze whether there exist any natural geometric invariant functions that  geometric graph models are unable to compute while the proposed model succeeds in doing so.

3. The end of Proposition 2 states "In most of the cases, an ETNN is strictly more powerful than an EGNN". I tried finding the proof to this in appendix F and had a hard time. I think the authors refer to proposition 2 in appendix F but I'm not sure. A clearer framing of this result in the appendix, and perhaps an illustrative example or plot in the main body, would improve readability and support this claim.

4. The paper refers to the architecture proposed in [2] for geometric simplicial complexes. A comparison of the expressive power of ETNN using different lifts compared to this architecture would be interesting.

5. The paper [3] benchmarks higher-order message passing on several geometric benchmarks, using data augmentation to address $O(d)$ symmetries. An empirical comparison to this approach would provide valuable insight.

[1] Cristian Bodnar, Fabrizio Frasca, Nina Otter, Yuguang Wang, Pietro Lio, Guido F Montufar, and Michael Bronstein. Weisfeiler and lehman go cellular: Cw networks. Advances in neural information processing systems, 34:2625–2640, 2021.

[2]  Floor Eijkelboom, Rob Hesselink, and Erik J Bekkers. E (n) equivariant message passing simplicial networks. In International Conference on Machine Learning, pages 9071–9081. PMLR, 2023.

[3] Mustafa Hajij, Ghada Zamzmi, Theodore Papamarkou, Nina Miolane, Aldo Guzm´an-S´aenz, Karthikeyan Natesan Ramamurthy, Tolga Birdal, Tamal K Dey, Soham Mukherjee, Shreyas N Samaga, et al. Topological deep learning: Going beyond graph data. arXiv preprint arXiv:2206.00606, 2022.

[4] Vıctor Garcia Satorras, Emiel Hoogeboom, and Max Welling. E (n) equivariant graph neural networks. In International conference on machine learning, pages 9323–9332. PMLR, 2021.

**Questions:**

see weaknesses.

---

> ### Author Response · Authors · 2024-11-16
> **Addressing Reviewer's comments**
>
> We would like to thank the Reviewer for both pointing out the paper’s strengths, as well as their raised concerns regarding novelty, expressivity and the comparison with the CCMPNs. Next, we address these points.
>
> ## Weaknesses.
> **[W1] Novelty**
>
> On the one hand, it is true that ETNNs are scalarization-based architectures, as EGNNs [4], on higher-order combinatorial domains, as EMPSNs [2]. As such, they inevitably resound with each other.
>
> However, we kindly disagree with the reviewer’s comment since ETNNs are a formal and more expressive generalization of both EMPSNs and EGNNs. For this reason, ETNNs unlock several features (e.g. arbitrary modeling of heterogeneous hierarchical higher-order interactions, tunable expressive power, and general learnable geometric invariants) that the constrained graph or simplicial structures of EGNNs and EMPSNs cannot accommodate (see **Appendix C** and **F**). As such, our framework can be used to design arbitrary E(n) equivariant TDL models. No other framework has the same power.
>
> Moreover, although we believe that the generality of our framework is fundamental, we also recognized the necessity for applications whose exhaustive modeling (in a combinatorial/topological sense) is possible only via combinatorial complexes, and we introduced MolecularCCs and GeospatialCCs to tackle this problem (no other complex nor graph can model molecular and multi-resolution irregular geospatial data as straightforwardly as CCs). As a consequence, we achieved SotA results or matched the existing SotA results among the Equivariant TDL models with a huge reduction in computational complexity.
> As a byproduct, as reviewer NQcw noticed too, our air pollution downscaling task represents a novel benchmark for the TDL community, which can be proven valuable to its needs, as noticed in the position paper [5].
>
> Finally, we believe that the expressivity analysis is novel. In particular, our approach is, to the best of our knowledge, currently the most exhaustive one for scalarization-based equivariant TDL models, as it could be applied to analyze the (geometric WL) expressivity of any scalarization-based equivariant TDL model without the need for domain-specific coloring procedures.
>
> **[W2] Expressivity and Geometric Invariance**
>
> Our expressivity analysis in **Appendix F** assumes that the employed geometric invariant is the sum of pairwise distances. We believe that this is already the simplest and most computationally efficient geometric invariant. Moreover,  as discussed in **Remark 5**,  the assumptions of Proposition 1 are required for a clean theoretical treatment, but we empirically observed that different aggregations, as well as different message functions and geometric invariants, do not affect the expressiveness. This makes sense, as the assumptions in **Proposition 1** lead to the simplest possible architectural setting for an ETNN. About what geometric invariant functions graph models are unable to compute/learn, we are not sure we understood what the reviewer meant. The geometric invariants take as input all the geometric features of the nodes belonging to a certain cell. As such, a graph cannot compute/learn anything that takes as input more than two node geometric features (i.e., what is induced by the presence of an edge), as it cannot jointly access higher-order information (i.e., geometric features of more than two nodes).
>
> **[W3] ETNN vs EGNN expressivity**
>
> The reviewer is correct that we need to refine the statement to make it clearer. In the revised manuscript, we clearly stated the condition of improved expressivity induced by **Proposition 2** in the new **Corollary 1**. In particular, an ETNN is strictly more powerful than an EGNN in distinguishing $k$-hop distinct graphs if the skeleton-preserving lifting is such that the number of ETNN layers required to have at least one cell in  $X_{G_1}$/$X_{G_2}$ whose receptive field is the whole set of nodes in $G_1$/$G_2$ is smaller than the number of EGNN layers required to have at least one node in  $G_1$/$G_2$ whose receptive field is the whole set of nodes in $G_1$/$G_2$. This statement is exhaustively empirically confirmed in **Table 3**. Regarding the plot, including it in the main body of the paper is challenging due to space constraints. However, we will make an effort to incorporate it in the final camera-ready version of the paper.
>
> *Continuing response in the thread (References are included in the last part of the thread)*

---

> > ### Author Response · Authors · 2024-11-16
> >
> > **[W4] ETNN vs EMPSN expressivity**
> >
> > As a consequence of what we wrote in our reply to **[W1]**, a comparison is not required, because CCs generalize Simplicial Complexes (SCs) and our approach based on the novel notion of *Geometric Augmented Hasse Graph* can thus be directly applied to SCs. As we carefully explained in the paper, an expressivity comparison (in the geometric WL sense) is fair only when the lift is skeleton-preserving, i.e. when the underlying graph connectivity is preserved; in [2], they do not lift the graphs (and they do not provide any expressivity analysis), but rather they replace the connectivity with a Vietori-Rips SC. Although one can indeed lift a graph into a simplicial complex preserving the graph connectivity by taking triangles or cliques as higher-order simplices, the expressivity would be characterized exactly as for **Proposition 2** (and as in **[W3]** to compare with EGNNs).
> >
> > **[W5] Comparison with CCMPNs**
> >
> > We believe this is an interesting point, as there is an ongoing debate on where and when either hard equivariance inductive bias or a massive data augmentation is required. However, implementing the data augmentations and rerunning all the experiments is beyond what can be accomplished within the given 1-week timeframe of the rebuttal. Having said that, we are committed to conducting these experiments and incorporating the results in the final camera-ready version of the paper. Nevertheless, we exhaustively showed the superior performance of ETNNs against CCMPNs with no data augmentation (configurations. 16-17-25-28-37-38-39 of **Tables 10-11**).
> >
> > ## References
> >
> > [1] Cristian Bodnar, Fabrizio Frasca, Nina Otter, Yuguang Wang, Pietro Lio, Guido F Montufar, and Michael Bronstein. "Weisfeiler and Lehman Go Cellular: CW Networks." *Advances in Neural Information Processing Systems*, 34:2625–2640, 2021.
> >
> > [2] Floor Eijkelboom, Rob Hesselink, and Erik J Bekkers. "E(n) Equivariant Message Passing Simplicial Networks." In *International Conference on Machine Learning*, pages 9071–9081. PMLR, 2023.
> >
> > [3] Mustafa Hajij, Ghada Zamzmi, Theodore Papamarkou, Nina Miolane, Aldo Guzmán-Sáenz, Karthikeyan Natesan Ramamurthy, Tolga Birdal, Tamal K Dey, Soham Mukherjee, Shreyas N Samaga, et al. "Topological Deep Learning: Going Beyond Graph Data." *arXiv preprint* arXiv:2206.00606, 2022.
> >
> > [4] Víctor Garcia Satorras, Emiel Hoogeboom, and Max Welling. "E(n) Equivariant Graph Neural Networks." In *International Conference on Machine Learning*, pages 9323–9332. PMLR, 2021.
> >
> > [5] Papamarkou, Theodore, et al. "Position: Topological Deep Learning is the New Frontier for Relational Learning." *Forty-first International Conference on Machine Learning*, 2024.

---

> ### Comment · Reviewer_wwvW · 2024-11-18
> **Response:**
>
> I appreciate the authors' detailed response, which I will now address:
>
> [W1] I recognize that ETNNs can process combinatorial complexes, whereas EMPSNs are limited to simplicial complexes, making ETNNs more versatile. I also agree that the inclusion of two novel combinatorial complex benchmarks is a valuable contribution, highlighting the strengths of ETNNs. However, I believe the main advantage of ETNNs over EMPSNs is primarily technical, as both rely on scalarization combined with neighborhood-dependent updates. To my understanding, there is nothing inherently restrictive in EMPSNs update rule that prevents them from handling combinatorial complexes, even though they have been framed as a simplicial complex-focused architecture. Despite this, I agree that the ETNN update rule is more generic and flexible than EMPSNs.
>
> Regarding the expressivity analysis, I agree that naturally extending the augmented geometric WL test is relevant and provides a non-domain-specific framework for analyzing geometric WL expressivity. However, the reduction of combinatorial complexes to Hasse graphs has been explored on multiple occasions  and, in my view, applying a linear permutation-invariant map to the nodes of the cells appears to be a straightforward extension. Still the additional experiments testing ETNN on the counterexample structures from previous papers are a nice addition and the experimental section of the paper in general is strong.
>
>
>
> [W2] I agree that the sum of pairwise distances is a natural and straightforward invariant to consider, and I appreciate the empirical demonstration that different aggregation methods do not impact expressiveness. However, from a theoretical perspective, I would have liked to see formal proofs for statements such as: “ETNN with permutation-invariant functions of pairwise distances can implement any function that ETNN with Hausdorff distance or convex hull volume can”. This would theoretically validate the empirical claim that  different aggregation methods do not impact expressiveness.
>
> Additionally, I believe that a graph could theoretically learn geometric node features for a node  x, such as "the volume of the convex hull of the  k-neighborhood of x" or "the distance of
> x from the barycenter of the graph." However, it is unclear whether either ETNN with permutation-invariant functions of pairwise distances or EGNN can effectively learn these types of node features.
>
> [W3] I thank the authors for the clarification.
>
> [W4] The authors note that “one can indeed lift a graph into a simplicial complex preserving the graph connectivity by taking triangles or cliques as higher-order simplices”. Given the paper's claim that using combinatorial complexes is more beneficial than simplicial complexes, a compelling theoretical result would be to demonstrate that ETNNs using lifts that result in non-simplicial complexes are more expressive than ETNNs using lifts that result in  simplicial complexes. This might be a generalization of proposition 2, but I think it is an important  one.
>
> [W5] I thank the authors for their commitment to adding the requested experiments.

---

> > ### Author Response · Authors · 2024-11-18
> >
> > **[W1]** We are glad the reviewer appreciated the increased versatility of ETNN and the strength of our experimental section, recognizing the contributions of our work. We also thank them for their prompt response, as usually in these cases, it is vital to have a constant and fruitful discussion.
> >
> > However, we would like to point out that the sentence *“To my understanding, there is nothing inherently restrictive in EMPSNs update rule that prevents them from handling combinatorial complexes, even though they have been framed as a simplicial complex-focused architecture.”* is slightly tendentious: the fact that an architecture can be directly generalized does not imply that the generalization has no value. Especially when the treatment, as in our ETNN case, is clearly broader, more comprehensive, and supported by novel and non-trivial experiments. EMPSN is a simplicial neural network and should be considered so. It could be generalized, and we did it by clearly motivating it both in methodological (i.e., expressivity and geometric invariants), experimental (i.e., a novel benchmark), and computational (i.e., less than half of EMPSN memory usage and almost half of EMPSN runtime with same or better performance) terms. In mere terms of forward pass (that obviously is just a piece of our work),  while ETNN forward can process data on a SC, EMPSN forward cannot be used to process data on a CC. The main reasons regard the two most important components of a scalarization-based equivariant architecture, i.e., (1) Feature updates and (2) Geometric invariants.
> >
> > Regarding (1), SCs induce specific neighborhood functions (Boundary, Coboundary, Up/Down Adjacencies), thus if a CC comes with a non-standard collection of neighborhoods, EMPSN forward cannot handle it, both for non-geometric and geometric features updates.
> >
> > Regarding (2), EMPSN forward uses tailored geometric invariants for SCs. Computing the volume of a non-simplex makes no sense. Similarly, pruning the pairwise distances based on the simplicial structure or using the angles of the planes induced by the dimension of the simplices makes no sense in the CC case.
> >
> > Regarding expressivity, although the reduction of combinatorial complexes to Hasse graphs has been explored in a couple of works, the only benchmark study using it for expressivity purposes is [1]. As such, we still believe that our expressivity analysis, being the first one to (partially) generalize the arguments from [1] to the geometric setting, relating it to the geometric WL, is a significant contribution.
> >
> > **[W2]** We agree with the reviewer that questions like *“can ETNN with permutation-invariant functions of pairwise distances implement any function that ETNN with Hausdorff distance can?”* would be interesting for (mainly) computational aspects. However, formally proving general statements on this topic would require an entire paper (maybe a paper also related to the generalization capabilities of ETNN?). The simplistic answer to questions like the one above is yes. In the specific case of the Hausdorff distance, for example, it would suffice to use a learnable weighted sum of pairwise distances and the network should learn to set to zero all the weights of the sum not corresponding to the pair of points in the two cells that are maximally distant (in the set sense of the Hausdorff distance).
> >
> > Regarding graph models, could the reviewer point us to some work showing what are the geometric invariants that geometric graph models can learn starting from pairwise distances? Intuitively, given our answer above, ETNN should be able to do it more straightforwardly anyway.
> >
> > Overall and most importantly, we would like to stress that, although an interesting problem, there is no need to focus too much on which kind of invariants can be learned from pairwise distances because ETNN is natively ready to be fed directly with precomputed Hausdorff distances, the volume convex hulls or, in general, higher-order geometric invariants. This is actually one of the main features of Equivariant TDL models in general.
> >
> > **[W4]** We believe that the notion itself of a theoretical expressivity result about (geometric) CCs vs SCs is ill-conditioned. The theoretical reason is that the statement would be equivalent to the new **Corollary 1** but replacing EGNN and graphs with EMPSN and simplicial complexes. More importantly, the practical reason is that on both the numerical expressivity tasks that we show, the $k$-chain graphs (**Table 3**) and the counterexamples (**Table 4**), EMPSN with any skeleton preserving lifting would be as expressive as an EGNN (and, thus, way less expressive than an ETNN), as there are no triangles nor cliques.
> >
> > ## References.
> > [1] Jogl, Fabian, Maximilian Thiessen, and Thomas Gärtner. "Expressivity-preserving GNN simulation." Advances in Neural Information Processing Systems 36 (2024).

---

> > > ### Author Response · Authors · 2024-11-20
> > > **Follow-up on Reviewer’s feedback**
> > >
> > > We would like to ask you whether our response addressed your concerns, weaknesses, and questions so far. Also, we’d like to know whether you have any other questions?
> > >
> > > We would greatly appreciate a prompt feedback, as it would allow us to clarify any remaining issues and further improve the quality of our manuscript.

---

> > > > ### Author Response · Authors · 2024-11-22
> > > > **2nd Follow-up: Are there any outstanding points?**
> > > >
> > > > We'd like to thank once again the Reviewer for the constructive review. We want to follow up to ask whether our responses addressed their questions so far. Otherwise, if the Reviewer is satisfied with our response and the clarifications, we would kindly ask them to reconsider the rating of our submission.

---

> > > ### Comment · Reviewer_wwvW · 2024-11-23
> > > **Response**
> > >
> > > I thank the authors for their detailed response and address it below.
> > >
> > > [W1] I want to emphasize that I don’t believe the paper lacks novelty entirely; rather, its novelty is somewhat limited. Though I agree that the generalization from simplical complexes to combinatorial complexes has value, and that including geometric invariants in the update is a good addition to previous methods, I still find the novelty of these contributions to be somewhat modest.
> > >
> > > [W2] Though I agree that  “ETNN is natively ready to be fed directly with precomputed Hausdorff distances, the volume convex hulls or, in general, higher-order geometric invariants” I feel like answering these type of questions would make  for a more comprehensive theoretical section. While I don’t expect the authors to answer all of the  theoretical questions I posed in my review, I gave these as examples to demonstrate that many theoretical aspects of the proposed architectures remain unexplored.
> > >
> > > [W4] I thank the reviewer for this answer and agree with it.
> > >
> > > In general, I agree with other reviewers saying "Despite being a very well written and tested paper, the main issue I have is the extent of contribution and novelty, as I outline in my answers above. I think the paper deserves to be published in any case" and will keep my score as it is.

---

> ### Author Response · Authors · 2024-11-24
>
> We thank the reviewer for their engagement, further feedback, and for stating that our paper deserves to be published. We sincerely and deeply appreciate the time the reviewer dedicated to our work. Here are some further comments.
>
>
> **[W1]** It is undoubtedly true that ETNN is a simple equivariant generalization of CCMPNs, being a scalarization-based equivariant model, similar to  EMPSN  being a simple generalization of MPSNs, and EGNN  being a simple generalization of MPNs. However, beyond the technical differences we already described, overall we showed that ETNNs are **more general and flexible**, **more expressive**, and **similar or better performing while being hugely more scalable** than SotA methods in the same class. These facts, together with the **benchmark**, **software**, and **outreaching** contribution of our work, should motivate its publication, as the reviewer agrees. This said, we already wrote in the future directions section that beyond scalarization-based architectures are of great interest. However, given that **a gap in the literature was still present for scalarization-based models** too, it made sense to us to **first comprehensively fill it** by working in the direction of ETNN.
>
>
>
> **[W2]** We agree with the reviewer that open questions remain despite our extensive analysis and treatment. However, we believe this is almost always the case for any new architecture, especially in the case of a very young field such as Equivariant TDL (indeed, the same questions could be rightfully asked for the other architectures in the field). In this sense, ETNN, in our opinion, already represents the most comprehensive work in its related literature, and we believe it can spark interest to be further studied, by the community and by us as well.
>
> Thanks again!

---

> ### Author Response · Authors · 2024-11-26
> **Ablation study on geometric invariants**
>
> **[W2]** Although, as the reviewer agreed, it is unfeasible but interesting to add further theoretical results about this matter to this already extensive work, we, as we said, agree and believe that the theoretical and empirical impact of geometric invariants matters. For this reason, we conducted a small ablation study to analyze the impact of different geometric invariants on model performance. In this experiment, we used the property $\alpha$ of QM9 as the target variable. For the Combinatorial Complex configuration, we employed the first configuration outlined in Table 11 of Appendix J (which consists of atom, bond, and virtual cell with incidences and max adjacency). Regarding the choice of invariants, we utilized the following:
>
> - Centroids Distance: This metric represents the Euclidean distance between the centroids of the two participating cells. It belongs to the second family of measures outlined in the manuscript, specifically the “Distances of permutation-invariant functions”.
>
> - Hausdorff Distance: AS described in the paper. This measures the largest minimum distance between nodes in the two cells.
>
> We indicate whether an invariant is active or inactive by setting it to 1 or 0, respectively.
>
>
> |   Run |   Centroids Distance |  Hausdorff Distance  |   Test MAE |
> |------:|---------------------:|---------------------:|-----------:|
> |     1 |                    1 |                    0 |  0.0834196 |
> |     2 |                    0 |                    1 |  0.0747732 |
> |     3 |                    1 |                    1 |  0.0733149 |
> |     4 |                    0 |                    0 |  0.416896  |
>
> Based on the results in the table, we make the following preliminary observations:
>
> - Using both geometric invariants yields the best performance (Run 3).
>
> - Excluding both Centroids Distance and Hausdorff Distance (no geometric information, CCMPN-like) significantly harms performance (Run 4).
>
> - Hausdorff Distance appears to be a slightly better choice than Centroids Distance as a geometric invariant (Run 2 vs. Run 1).
>
> These results show an interesting interplay among the invariants, giving hints on the impact of each one. As today is the last day to submit a revision of the paper, we plan and commit to extending this ablation study to include additional configurations and target variables for the camera-ready version of the paper.

---

> > ### Author Response · Authors · 2024-11-28
> > **Follow-up after deadline extension**
> >
> > In light of the extension of the discussion deadline, we would like to thank the reviewer again for the already positive assessment of our work and for their engagement, and ask if there is anything else we can do to further improve the manuscript and its score.
> >
> > We also kindly invite the reviewer to parse the discussions we had with the other reviewers, as we believe that the overall engagement and the individual feedback have clarified or improved several aspects of the paper.
> >
> > Thanks a lot!

---

> ### Author Response · Authors · 2024-12-03
> **Extension of the ablation study on the geometric invariants**
>
> As we mentioned in our previous message about the ablation study over the impact of the geometric invariant choice, we follow up with an extended study over multiple targets, and multiple experiment configurations (as these are reported in **Appendix J**. Below, we present the results of this study. The numbers of the experiment configurations correspond to the Combinatorial complex configuration, as presented in **Table 11**.
>
> **Remark:** We note that due to time constraints we run the next configurations for a total of 350 epochs (instead of 1000 epochs of our original results in Table 11).
>
> Experiment Configuration: 1
> | Invariant Choice   |     $\alpha$|       $\Delta\epsilon$ |      $\epsilon_\text{HOMO}$ |      $\epsilon_\text{LUMO}$ |        $mu$ |
> |:--------------------|----------:|----------:|----------:|----------:|----------:|
> | Centroid       | 0.0878398 | 0.0496196 | 0.0366693 | 0.0247688 | 0.0345231 |
> |  No invariants       | 0.0946521 | 0.105025  | 0.0347686 | 0.129023  | 0.0672308 |
> | Centroid + Haudsorff        | 0.0837419 | 0.0465217 | 0.0355672 | 0.0197688 | 0.0295231 |
> | Hausdorff        | 0.089013  | 0.0593732 | 0.0453292 | 0.0327621 | 0.0403032 |
>
> Experiment Configuration: 2
> | Invariant Choice   |     $\alpha$|       $\Delta\epsilon$ |      $\epsilon_\text{HOMO}$ |      $\epsilon_\text{LUMO}$ |        $mu$ |
> |:--------------------|------------:|------------:|------------:|-----------:|------------:|
> | Centroid            |   0.109712  |   0.061029  |   0.0318968 |   0.02739  |   0.0388485 |
> | No invariants       |   0.0923415 |   0.0892041 |   0.0468100 |   0.037918 |   0.068132  |
> | Centroid + Haudsorff|   0.0939814 |   0.0529162 |   0.0330245 |   0.025901 |   0.0317646 |
> | Hausdorff           |   0.0982691 |   0.0510092 |   0.0331916 |   0.026162 |   0.0399919 |
>
>
> Experiment Configuration: 3
> | Invariant Choice   |     $\alpha$|       $\Delta\epsilon$ |      $\epsilon_\text{HOMO}$ |      $\epsilon_\text{LUMO}$ |        $mu$ |
> |:--------------------|------------:|------------:|-----------:|------------:|------------:|
> | Centroid            |   0.0978212 |   0.0621093 |   0.037318 |   0.0373242 |   0.0314632 |
> | No invariants       |   0.123016  |   0.194019  |   0.041092 |   0.1209326  |  0.0410223 |
> | Centroid + Haudsorff|   0.0865784 |   0.0517418 |   0.034421 |   0.0265419 |   0.0320411 |
> | Hausdorff           |   0.0921039 |   0.0580293 |   0.039023 |   0.0282301 |   0.0390323 |
>
>
> Experiment Configuration: 4
> | Invariant Choice   |     $\alpha$|       $\Delta\epsilon$ |      $\epsilon_\text{HOMO}$ |      $\epsilon_\text{LUMO}$ |        $mu$ |
> |:--------------------|-----------:|------------:|------------:|------------:|-----------:|
> | Centroid            |   0.123991 |   0.0591344 |   0.0320109 |   0.027891  |   0.320119 |
> | No invariants       |   0.180012 |   0.0712032 |   0.0371132 |   0.0314019 |   0.037239 |
> | Centroid + Haudsorff|   0.105327 |   0.0587845 |   0.0355854 |   0.0264562 |   0.031021 |
> | Hausdorff           |   0.112335 |   0.0595528 |   0.0347739 |   0.0292914 |   0.034909 |
>
>
> Following our preliminary results (see previous response), we validate again that:
> 1. Not using Centroids distance nor Hausdorff distance exhibits the **worst performance**.
> 2. In the majority of the runs, Hausdorff distance seems to perform **slightly better** than the Centroids distance.
> 3. Using both invariants usually yields the **best performance**.
>
> We’d like to thank once again the reviewer for the constructive feedback, and we plan to include such analysis in the camera-ready version.

---

### Official Review · Reviewer_h23Z · 2024-11-03

**Soundness:** 2
**Presentation:** 4
**Contribution:** 3
**Rating:** 6
**Confidence:** 3

**Summary:**

The authors propose an equivariant Topological Deep Learning framework that deals with geometric node features. The frameworks can be generalized to many topological domains including simplicial, cell, combinatorial, and path complexes. The authors provide theoretical analysis regarding his design choice as well as expressiveness of the proposed method. Lastly, the authors support his arguments via real-world datasets QM9 and his proposed benchmark - air pollution downscaling benchmark.

**Strengths:**

The authors add an important piece of work for the Topological Deep Learning (TDL) community as there is not much literature on Equivariant TDL. The work is well-formulated with clear motivations. The theoretical contributions are well-written. The paper is also self-contained and easy to follow, given the substantial explanations from related prior literature. The ablation studies are well-conducted via many synthetic graphs and additional information (hyperparameters, data statistics, etc.) are provided. The novel benchmark based on geospatial information is novel.

**Weaknesses:**

Novelty is the key disadvantage of the paper. It seems that the work just extends prior works on graphs to TDL. Even though the theoretical insights are important, yet they are mostly an extension from graphs. Another important weakness is scalability and practicability  of the problem. There are only two real-world datasets evaluated, and in both cases, graph sizes are small. Furthermore, the performance isn’t convincing given there are only minor improvements over the graph counterparts. The performance could be due to extra parameters for higher-order filters. Perhaps an experiment on model comparison with constraints on parameter budgets and an ablation study on higher-order features masked out are needed to prove your arguments. Lastly, even when the framework makes sense, it is unclear how we can obtain geometric features for higher-order cells. Please refer to question 6 for my concern.

**Questions:**

1. If I understand correctly, your argument on “heterogeneous interactions” focuses on different relationships between cells. Meanwhile, this property was actually mentioned in prior literature ([3] to name a few). I don’t think it is fair to claim that ETNNs are set up for this characteristic, but more like TDL in general already possesses this property.
2. The paper mentioned that Combinatorial Complex subsumes Path Complex. I believe there are two distinct lines of work regarding complexes arising from paths. One is path-based Combinatorial Complex [3], and another work is a simplified path complex [2] based on path complex [1]. [2] can’t be derived directly from [3] because [2] reserves the sequential information of paths. From my understanding, your work focuses more on path-based combinatorial complexes.
3. A relevant work [4] is not discussed in the paper.
4. It seems to me that $\mathcal{N}_{A, \text{max}}$ isn’t supported by any experiments. It would be better if the authors provide a set of experiments to support this design choice.
5. A minor subjective feedback on writing. I think it is better to have different notations for “containment” and “is a subset of” operation for clarity (Equation 6 and 7).
6. Regarding section I.1.2, it seems that 2-cell features do not encode any geometric features (velocity for example) but only invariant features. I think it is a missing piece to convince the audience that your framework can work with geometric features. Also, suppose that even when we have geometric features at node levels, it is unclear how to lift these features into higher-order spaces.

[1] Grigor’yan, A.A., Lin, Y., Muranov, Y.V. et al. Path Complexes and their Homologies. J Math Sci 248, 564–599 (2020).

[2] Truong, Q., & Chin, P. (2024). Weisfeiler and Lehman Go Paths: Learning Topological Features via Path Complexes. Proceedings of the AAAI Conference on Artificial Intelligence, 38(14).

[3] Hajij, M. et al. Topological Deep Learning: Going Beyond Graph Data (2023).

[4] Li, L. et al. Path Complex Neural Network for Molecular Property Prediction. ICML 2024 Workshop GRaM (2024).

---

> ### Author Response · Authors · 2024-11-16
> **Addressing Reviewer's comments and questions**
>
> We sincerely thank the reviewer for their valuable feedback and suggestions. We would also like to thank the reviewer for pointing out as strengths the importance of this work for the TDL community, the text clarity, its clarity, completeness and self-containment. Next, we address each point raised as weakness and answer all of the reviewer’s questions.
>
> ## Weaknesses.
> **[W1] Novelty**
>
> On the one hand, it is true that ETNNs are scalarization-based architectures, as EGNNs, on higher-order combinatorial domains, as EMPSNs. As such, they inevitably resound with each other.
>
> However, we kindly disagree with the reviewer’s comment since ETNNs are a formal and more expressive generalization of both EMPSNs and EGNNs. For this reason, ETNNs unlock several features (e.g. arbitrary modeling of heterogeneous hierarchical higher-order interactions, tunable expressive power, and general learnable geometric invariants) that the constrained graph or simplicial structures of EGNNs and EMPSNs cannot accommodate (see **Appendix C** and **F**). As such, our framework can be used to design arbitrary E(n) equivariant TDL models. No other framework has the same power.
>
> Moreover, although we believe that the generality of our framework is fundamental, we also recognized the necessity for applications whose exhaustive modeling (in a combinatorial/topological sense) is possible only via combinatorial complexes, and we introduced MolecularCCs and GeospatialCCs to tackle this problem (no other complex nor graph can model molecular and multi-resolution irregular geospatial data as straightforwardly as CCs). As a consequence, we achieved or matched SotA results among the Equivariant TDL models with a huge reduction in computational complexity.
>
> As a byproduct, as **Reviewer NQcw** noticed too, our air pollution downscaling task represents a novel benchmark for the TDL community, addressing a need highlighted in the recent position paper [6].
> Finally, we believe that the expressivity analysis is novel. In particular, our approach is, to the best of our knowledge, currently the most exhaustive one for scalarization-based equivariant TDL models, as it could be applied to analyze the (geometric WL) expressivity of any scalarization-based equivariant TDL model without the need for domain-specific coloring procedures.
>
> **[W2] Scalability**
>
> We kindly disagree with the reviewer since we believe that scalability is one of ETNN’s strengths (as pointed out by **Reviewer NQcw** too). In particular, as described in **Section 5** and **Appendix G**, we showed that ETNN achieves or matches SotA results among E(n) Equivariant TDL models with considerable gains in memory, time, and, overall, scalability. These facts make ETNN amenable to be used on larger datasets in tailored, future works. The scope of this work is to introduce the framework, analyze it in detail, make it accessible, and show how it can be easily used for problems of vastly different natures. About the ablation, EGNN-graph-W has been introduced exactly to show how a graph model with the same parameter budget would perform, while the use or not of features has been extensively studied in **Appendix J**, **Tables 10-11** (HO Features column). Finally, we do not consider an average improvement of more than 11% incremental.
>
> ## Questions.
>
> **[Q1]  Heterogeneous Interactions**
>
> The reviewer is right. Indeed, we state that CCMPNs can handle heterogeneity before we even introduce ETNN, and we give the right credit to [3]. However, it remains true that ETNN are set up for heterogeneity (in the geometric setting too). To improve the fairness of our presentation, in the revised manuscript we made clear that TDL models have this feature in the abstract.
>
> **[Q2] Path Complexes as CCs**
> We believe that, as long as the considered paths are undirected (and this seems to be the case in [2], [3], [4] but not in [1]), a path complex can be easily cast as a CC in which the cells are the paths and the incidences and adjacencies are defined via the boundary relation of the complex (that, of course, takes into account the sequential nature of paths by definition) and not the set inclusion, as it happens in cell complexes (that indeed generalize path complexes). In the case of directed paths in the complex, then CCs cannot model it as the neighborhood structure would be possibly asymmetric. In this case, a different approach as the one in [5] should be employed. We also added [4] in our related works section.
>
>
> *Continuing response in the thread (References are included in the last part of the thread)*

---

> ### Author Response · Authors · 2024-11-16
>
> **[Q4] Max Adjacency**
>
> We exhaustively study how the max adjacency (and the virtual cell) impacts the performance. In **Tables 10-11**, the “Adjacencies” column tells exactly what adjacencies are used. In general, the max adjacency and the virtual cell increase performance, and this is expected, as the virtual node [7] had a similar effect on graph-based models.
>
> **[Q5] Notation**
>
> The equations only require “containment” notation since $\mathcal{CN}$ contains neighborhood systems, $\mathcal{N}(x)$ contains neighboring cells, $x$ and $y$ are cells that contain atoms $z$.
>
> ***Could the Reviewer please clarify which components of **Equations 6** and **7** are they referring to?***
>
> **[Q6] Higher-order geometric features**
>
> The setting that we tackle, which is the same as EGNN and EMPSN and is clearly stated at the beginning of **Section 3**, is the setting in which nodes (0-cells) are embedded in some Euclidean space, i.e., they come with both non-geometric and geometric features. This makes sense as higher-order cells are not necessarily physical entities. That said, in **Remark 1** we also explain how to modify ETNN to integrate higher-order cells with geometric features, if available. Further, **Appendix D** discusses velocity-type inputs. However, in both cases, it is not clear to us what the reviewer means when they write about lifting (node) geometric features. The geometric features, as it is written in **Equation 7**, are updated only for the cells (say the nodes) that come with geometric features. On the other hand, the non-geometric features are updated for all the cells (**Equation 6**), and geometric invariants take into account the geometric features of all the nodes that are a part of a higher-order cell. Because of the message-passing operations between the different ranks, subsequent layers of message-passing at higher-order ranks will use the updated information from the geometric features at the node level.
>
> ## References
>
> [1] Grigor’yan, A.A., Lin, Y., Muranov, Y.V., et al. "Path Complexes and their Homologies." *Journal of Mathematical Sciences*, 248, 564–599 (2020).
>
> [2] Truong, Q., & Chin, P. "Weisfeiler and Lehman Go Paths: Learning Topological Features via Path Complexes." *Proceedings of the AAAI Conference on Artificial Intelligence*, 38(14) (2024).
>
> [3] Hajij, M., et al. "Topological Deep Learning: Going Beyond Graph Data." (2023).
>
> [4] Li, L., et al. "Path Complex Neural Network for Molecular Property Prediction." *ICML 2024 Workshop GRaM* (2024).
>
> [5] Lecha, Manuel, et al. "Higher-Order Topological Directionality and Directed Simplicial Neural Networks." *arXiv preprint* arXiv:2409.08389 (2024).
>
> [6] Papamarkou, Theodore, et al. "Position: Topological Deep Learning is the New Frontier for Relational Learning." *Forty-first International Conference on Machine Learning* (2024).
>
> [7] Sestak, Florian, et al. "VN-EGNN: E (3)-Equivariant Graph Neural Networks with Virtual Nodes Enhance Protein Binding Site Identification." *arXiv preprint* arXiv:2404.07194 (2024).

---

> > ### Author Response · Authors · 2024-11-20
> > **Follow-Up on Reviewer Feedback**
> >
> > We would like to follow up to ask if our response addresses the reviewer’s concerns, weaknesses, and questions. We would greatly appreciate a prompt feedback, as it would allow us to clarify any remaining issues and further improve the quality of our manuscript. In case the Reviewer is satisfied with our response and the clarifications, we would kindly ask them to reconsider the rating of our submission.

---

> > > ### Author Response · Authors · 2024-11-22
> > > **2nd Follow-Up: Are there any outstanding questions or concerns?**
> > >
> > > We would like to ask the reviewer if there are any outstanding questions or concerns, that were not addressed in our previous responses. We would greatly appreciate prompt feedback to address any points and ensure our manuscript meets expectations.

---

> > ### Comment · Reviewer_h23Z · 2024-11-22
> > **Response to the Authors regarding Weaknesses and Questions**
> >
> > First of all, I apologize for the late response. Below are my detailed responses regarding your rebuttal.
> >
> > **[W1, Q6] Novelty and Higher-order Geometric Features**
> >
> > I kindly disagree with the authors on the novelty of the paper. I acknowledge that the work is critical to the field as there is limited existing work attempts to formally generalize the Topological Deep Learning (TDL) framework to geometric spaces; however, the generalization is trivial as similar to how EGNNs generalize to GNNs. As higher-order relations are just a generalization of adjacency matrices in graphs, it is trivial to insert any geometric invariant features during message passing with higher-order relations to make the TDL framework equivariant.
> >
> > What I am interested more is how the authors construct geometric features for higher-order cells as stated in Q6. As the authors clarify, it is possible that higher-order cells are not physical entities so they may not have higher-order geometric features. Therefore, the geometric information is only inserted for cells that contains this information, and we rely on message passing to pass this geometric information to subsequent higher-order cells. In other words, it seems that there is no geometric features for higher-order cells. While the proposed framework allows TDL to be equivariant for every order, the experiment for QM9 only leverages the geometric invariant features for 0-cells only. It makes the experiments may not support the framework fully.
> >
> > Also, I still have questions regarding the Geospatial CCs. Which geometric information is provided for different cells? I checked the Appendix and I only saw non-geometric features; I may be wrong, so please kindly point out which features are geometric. Or do you mean that distance to nearest is geometric invariant feature in this benchmark?
> >
> > **[Q2] Path Complexes**
> > Path Complex in [2] strictly follows the boundary operation defined in [1], where a k-path contains (k-1)-paths that exist in the graph. With that being said, it is possible that given a triangle, there can be 3 distinct 2-paths, but each 2-path shares the same set of edges on its boundary, even when an edge may not be a subset of the 2-path. I think your ablation studies treat 3-paths as 2-cells and consider edges belong to the 3-paths as their boundaries. This notion is different from what is proposed in [2]. Even though this part may not directly affect your work, I strongly encourage the authors to explain more on "Lift 4b" instead of just stating this is equivalent to the Path complex, as it may create confusion for future readers.
> >
> > **[Q5] Notation**
> > This is just my personal preference to improve readability, so it doesn't affect the score. I meant Eq. 6 and Eq. 7 on page 5, where $\mathcal{N} \in \mathcal{CN}$ and $x \in \mathcal{X}$ both use $\in$ to describe two different notions.
> >
> > **[W2] Scalability**
> > What I meant is that if there are any large graphs with geometric features (maybe millions of nodes for example) to see how the framework performs with respect to GNNs. However, I acknowledge this part may be out of this paper's scope, and as the authors demonstrated the framework superiority with respect to EMPSN, this concern is addressed.
> >
> > [1] Grigor’yan, A.A., Lin, Y., Muranov, Y.V., et al. "Path Complexes and their Homologies." Journal of Mathematical Sciences, 248, 564–599 (2020).
> >
> > [2] Truong, Q., & Chin, P. "Weisfeiler and Lehman Go Paths: Learning Topological Features via Path Complexes." Proceedings of the AAAI Conference on Artificial Intelligence, 38(14) (2024).

---

> ### Author Response · Authors · 2024-11-23
>
> We thank the reviewer for the detailed response and for agreeing that our work is critical to the TDL field and starts filling an important gap in the literature. In the following, we address their follow-up comments. We hope that these further replies along with the overall positive feelings the reviewer showed for our work, can convince them to raise their score.
>
> **[W1, Q6] Novelty and Higher-order Geometric Features**
>
> Elaborating more about what we wrote in the previous reply, we would like to point out that:
>
> (i) The *architectural* novelty is not representative of the novelty of our work as a whole. In this specific case, the novelty is given by:
>
> - (a) **Architectural** novelty: ETNN improves on the two most important components of a scalarization-based equivariant architecture, i.e., feature updates and geometric invariants. Regarding features update, graphs and SCs induce specific neighborhood functions (node adjacency for the former, boundary, coboundary, up/down adjacencies for the latter), thus if a CC comes with a non-standard collection of neighborhoods, neither (ofc) EGNN nor EMPSN can handle it, both for non-geometric and geometric features updates. Regarding geometric invariants, EGNN/EMPSN uses tailored geometric invariants for graphs/SCs. Imagine applying EMPSN on a CC. Then, computing the volume of a non-simplex makes no sense. Similarly, pruning the pairwise distances based on the simplicial structure or using the angles of the planes induced by the dimension of the simplices makes no sense. On the other hand, ETNN can handle arbitrary neighborhoods, and its geometric invariants work on arbitrary CTSs (and formally generalize some of the invariants of EMPSN and EGNN).
>
> - (b) **Experimental** novelty: the introduction of the new geospatial benchmark and our novel approach for molecular modeling represent a significant effort to tackle some of the important open problems of the field [3] and are a significant source of novelty (and an important resource for the community) as well.
>
> - (c) **Theoretical** novelty: ETNN is the only available framework for designing arbitrary E(n) equivariant scalarization-based TDL models and the expressivity proof leveraging the novel notion of geometric augmented Hasse graph has not appeared before in the literature of equivariant TDL.
>
> (ii) We believe that our architecture should be considered **simple** rather than *trivial*. If a model is designed to be general and flexible, and it is shown to be more expressive, scalable, and better performing than the SotA of its class, defining it trivial just because of the simplicity of its architectural definition sounds reductive (and slightly impolite) to us. This said, we explicitly wrote in the future directions section that beyond scalarization-based architectures are of great interest. However, given that the gap in the literature was present for scalarization-based models too, it made sense to us to comprehensively work in the direction of ETNN.
>
> (iii) Higher-order relations are not just a generalization of adjacency matrices in graphs. In the context of ETNN, it is true that it is a widely general class of models on a broadly general space. However, E(n) Equivariant versions of architectures, e.g., [4], leveraging powerful concepts coming from algebraic topology, e.g., homology groups, stratifications, filtrations, etc., arise when ETNNs and CCs are particularized to specific combinatorial topological spaces, e.g., cell complexes. Similarly,  E(n) Equivariant versions of architectures using specific insights on hypergraphs, e.g. [5], arise when ETNNs and CCs are particularized to hypergraphs. ETNN is the only framework having this joint feature, to the best of our knowledge.
>
> *Continuing response in the thread (references are at the end of the thread)*

---

> ### Author Response · Authors · 2024-11-23
>
> **[W1, Q6] Cont.** Regarding the geometric features for higher-order cells, we apologize if our previous reply was not clear enough. It is not true that “the geometric information is only inserted for cells that contain this information, and we rely on message passing to pass this geometric information to subsequent higher-order cells”. This is because geometric information can be leveraged either in the form of geometric features and/or in the form of geometric invariants. It is true that the **geometric features** are attacched only to cells that come with them, but the **geometric invariants** are used to update the subsequent (possibly) higher-order cells without any reliance on message passing, i.e. the geometric information contained in the geometric invariants is directly leveraged from the subsequent cells even if they do not come with geometric features. In this sense, what message-passing enables is the exchange of the (already attached) “geometric information” of all the cells, realized as geometric features and/or geometric invariants. This said, we totally agree that tailoring ETNN for applications in which higher-order cells have a (possibly) physical meaning with attached geometric features is intriguing, but it is an interesting future direction. Here, geometric invariants already represent a source of geometric higher-order information.
>
> Regarding the geospatial task, we use the coordinates of the points (0-cells) as geometric features. We are really sorry we forgot to write it in Appendix I, it was not on purpose. We will add this specification in the next revision of the manuscript we are planning to submit in the next few days.
>
>
> **[Q2] Path complexes**
> The reviewer is right about Lift 4b. It is not equivalent to a path complex, we will delete the sentence in the next revision to avoid adding further complexity. However, the reviewer's previous comment was about the possibility of rewriting a path complex as a CC. Regarding this, our answer remains valid: a CC where the cells (i.e., the elementary paths) are correctly specified and the boundary operation is used to define the neighborhood functions is equivalent to a path complex as in [2].
>
> **[Q5] Notation** We believe they both correctly indicate containment. A cell $x$ is an element of the set of cells $\mathcal{X}$, a neighborhood function $\mathcal{N}$ is an element of the set of the neighborhood functions $\mathcal{CN}$ (i.e. the collection of neighborhoods). However, in the revision, we will make it clearer that $\mathcal{CN}$ is a set when we first introduce it before Equation 3.
>
> ## References.
>
> [1] Grigor’yan, A.A., Lin, Y., Muranov, Y.V., et al. "Path Complexes and their Homologies." Journal of Mathematical Sciences, 248, 564–599 (2020).
>
> [2] Truong, Q., & Chin, P. "Weisfeiler and Lehman Go Paths: Learning Topological Features via Path Complexes." Proceedings of the AAAI Conference on Artificial Intelligence, 38(14) (2024).
>
> [3] Papamarkou, Theodore, et al. "Position: Topological Deep Learning is the New Frontier for Relational Learning." Forty-first International Conference on Machine Learning. 2024.
>
> [4] Battiloro, Claudio, et al. "Generalized simplicial attention neural networks." IEEE Transactions on Signal and Information Processing over Networks (2024).
>
> [5] Bai, Song, Feihu Zhang, and Philip HS Torr. "Hypergraph convolution and hypergraph attention." Pattern Recognition 110 (2021): 107637.

---

> > ### Author Response · Authors · 2024-11-24
> > **Did our response address the Reviewer's questions?**
> >
> > We’d like to ask the Reviewer whether our previous response addresses their latest questions. We’re more than happy to elaborate more on any raised concern from the Reviewer’s side.
> > Once again, we'd like to thank them for the engaging discussion so far!

---

> ### Comment · Reviewer_h23Z · 2024-11-25
> **Response to the Authors**
>
> I would like to thank the authors for the detailed response.
>
> **[W1. Q6]** I apologize if my previous comment on simplicity of the model sounds impolite. I agree with the authors on the scope of the project, whose novelties focus on a flexible and a general pipeline for TDL. However, my main point still remains the same; it seems to me that your experiments cannot fully support the pipeline. However, I think it is worth a future research direction on defining geometric features for non-standard collection of neighborhoods. Given that your paper already brought many interesting insights and a novel benchmark, and my expectation may not be reasonable for an already informatively dense paper, I will consider to raise my score after the following concerns are resolved.
>
> > However, E(n) Equivariant versions of architectures, e.g., [1], leveraging powerful concepts coming from algebraic topology, e.g., homology groups, stratifications, filtrations, etc., arise when ETNNs and CCs are particularized to specific combinatorial topological spaces, e.g., cell complexes.
>
> Could you reference relevant works on E(n) architectures that bridge the gap between Topological Deep Learning (TDL) and Traditional Topological Machine Learning methods that rely on homology groups, stratifications, filtrations, ... I checked [1], and it didn't mention anything about TDL and the traditional topological machine learning methods. Please kindly correct me if I am wrong. I think there is one concurrent work [2] trying to bridge the literature gap, but it seems the goal of your paper and this paper is very different. Because it is a concurrent work, so I just put it here as a reference and it doesn't affect the the paper score.
>
> > Regarding the geospatial task, we use the coordinates of the points (0-cells) as geometric features.
>
> This is my main concern regarding your response. Even though coordinates play as an important geometric features for molecular graphs, I don't think it is the same for the novel benchmark. In many real world applications, if you compute geometric invariant features based on coordinates, it doesn't make a lot of sense. For example, housing price in big cities will be different from that in rural area; however, under the E(n) architecture, you are computing geometric invariant features and eliminate such a useful information. Can you elaborate more how coordinates of the points are appropriate geometric features for the benchmark?
>
> **[Q2]** I completely agree with your argument; CC is the most general form of complex where one can define relations flexibly. I just meant that you should be more explicit on how you define Lift 4b to avoid confusion, as there are many complexes defined on paths.
>
> [1] Battiloro, Claudio, et al. "Generalized simplicial attention neural networks." IEEE Transactions on Signal and Information Processing over Networks (2024).
>
> [2] Topological Blindspots: Understanding and Extending Topological Deep Learning Through the Lens of Expressivity. Under review at ICLR'25.

---

> ### Author Response · Authors · 2024-11-26
>
> **[W1. Q6] TDL and TML**
> We thank the reviewer for bringing the reference to our attention. The goal of our previous reply was not to reference methods that bridge TDL with classical TML though,  as reviewer’s comment **was not** about this. We wanted to stress that higher-order interactions are not just encoded as generalization of graph adjacencies, but can rather be leveraged using algebraic topology tools as well. In particular, the work in [1] we refer to as an example, (i) explicitly and formally uses (weighted) Hodge and Dirac theories, (ii) explicitly uses their corresponding spectral theories [2,3,4] (that can be leveraged in combination with filtrations [5]), and (iii) implicitly uses homology theory because the kernels of the Hodge Laplacian (and, as a consequence, the kernel of the Dirac operator) contain homology information as their dimensions are the Betti numbers of the complex. To link to our previous comment about this, E(n) Equivariant versions of architectures like [1] can be readily obtained from our ETNN framework.
>
> In terms of macro-areas, works like [1] help to bridge TDL with algebraic topology using more of a Topological Signal Processing (TSP) [6] perspective, while we agree that works like [7] help to bridge TDL with algebraic topology using more of a TML  perspective. Furthermore, works like [8] help to bridge TDL with equivariant DL using more of a TML  perspective, while a general work like ETNN helps to bridge TDL with equivariant DL implicitly using more of a TSP (and traditional relational DL) perspective. Overall, the three mentioned disciplines (TSP, TDL, TML) are surely overlapping, fall under the same umbrella, and many links among them are yet to be studied (e.g., further insight on if and how results from [7] applies to  proper convolutional models like [1] that explicitly take some topological invariants into account). However,  this surely is an exciting motivation for being in this field at this time. Finally, we will also cite [7] in our paper as it is surely relevant to our work.
>
> **[W1. Q6] Geospatial task coordinates as geometric features**
> We would like to clarify some potential misunderstandings. Most coordinate systems are, by construction, artifacts built with notions of symmetry and equivariance, as they are often derived from projections. Specifically, we use the Mercator projection, which uses meters as the unit scale. This projection has the key property that Euclidean distance in coordinate space corresponds to geodesic distance. Thus, E(n)-equivariance is implicitly essential to the purpose of the Mercator projection. The exact coordinate values are irrelevant; what is important for our task is the projection's usefulness in expressing geodesic distances. Consequently, it is never desirable for the model to behave erratically with respect to a Euclidean transformation of the Mercator coordinates.
>
> In light of the above remark, it is not true that geometric invariant features eliminate useful information. It is known that even just pairwise distances are sufficient to recover angles in Euclidean space (cf., [9; Appendix E]), and therefore a coordinate system modulo the Euclidean group. The same logic applies to our study: by using geometric invariants in geospatial coordinate systems, we allow the model to use the coordinates modulo Euclidean transformations, which is the original intention of the Mercator projection. This implies that the model can still capture location-specific random effects in node-level tasks using only invariants, which is the reviewer's main concern.
>
> We recognize that this is a confusing point and central to the paper. Therefore, we have added a summarized version of our response to the revised text.
>
> [1] Battiloro, et al., 2024. "Generalized simplicial attention neural networks." IEEE Transactions on Signal and Information Processing over Networks.
>
> [2] Calmon et al. 2023. "Dirac signal processing of higher-order topological signals." New Journal of Physics 25.9.
>
> [3] Hansen & Ghrist, 2019. Toward a spectral theory of cellular sheaves. Journal of Applied and Computational Topology.
>
> [4] Yang, et al. 2022. "Simplicial convolutional filters." IEEE Transactions on Signal Processing.
>
> [5]  Grande & Schaub, 2024. "Disentangling the Spectral Properties of the Hodge Laplacian: not all small Eigenvalues are Equal." ICASSP.
>
> [6] Barbarossa & Sardellitti 2020. "Topological signal processing over simplicial complexes." IEEE Transactions on Signal Processing.
>
> [7] Eitan et al., 2025. "Topological blind spots: Understanding and extending topological deep learning through the lens of expressivity.", under review at ICLR'25.
>
> [8] Verma et al., 2024. "Topological Neural Networks go Persistent, Equivariant, and Continuous." ICML.
>
> [9] Satorras et al., 2021. "E(n) Equivariant Graph Neural Networks." ICML.

---

> > ### Comment · Reviewer_h23Z · 2024-11-26
> >
> > Thank you very much for the clarification. I really appreciate your dedicated efforts. As all concerns are well answered, I am pleased to raise the score. I hope this self-contained and informative piece of work will be an important literature for TDL in general.

---

> > > ### Author Response · Authors · 2024-11-28
> > > **Follow-up after deadline extension**
> > >
> > > In light of the extension of the discussion deadline, we would like to thank the reviewer again for the already positive assessment of our work and for their engagement, and ask if there is anything else we can do to further improve the manuscript and its score.
> > >
> > > We also kindly invite the reviewer to parse the discussions we had with the other reviewers, as we believe that the overall engagement and the individual feedback have clarified or improved several aspects of the paper.
> > >
> > > Thanks a lot!

---

### Official Review · Reviewer_NQcw · 2024-11-03

**Soundness:** 3
**Presentation:** 4
**Contribution:** 2
**Rating:** 6
**Confidence:** 3

**Summary:**

This paper introduces an equivariant model within the framework of topological deep learning. The architecture generalizaes the equivariant graph neural network architecture from Santorras at al. from the setting of graphs to message passing over combinatorial complexes. Notably, this architecture allows for message passing with cells that have heterogeneous node feature over differening ranks.

The authors first introduce the relevant theory of combinatorial complexes, topological deep learning and important noitions of equivariance.  In section 3, they introduce their architecture using the formalism from the previous section. The authors then discuss important theoretical aspects of their model; proving equivariance, comparing expressiveness with traditional equivariant, scalarization-based techniques and discussing computational complexity of their design.

The authors discuss in section 4 two complementary examples of how to attain combinatorial complexes: firstly in molecular data and secondly in geospatial data. In section 5, the authors use said combinatorial complexes to benchmark their method against common architectures on the QM9 dataset. Secondly, the authors introduce a novel benchmark for predicting airpollution from annoted geospatial data.

**Strengths:**

Presentation, clarity and experimental transparency

I found the paper to be written clearly, and the mathematical statements and proofs were precise, well formulated and easy to follow. Further, the authors included a lot of helpful background about topological deep learning, and explained concepts clearly.

One of the biggest strengths of this paper is the much appreciated transparency around testing in the appendix. This allowed a confident and clear understanding of what the authors actually did to obtain their results. I really appreciated the detailled description of cell features in section I.1.2 and I.2.2. The detailled ablation studies on the were also super helpful to understand and evaluate overall performance.

Novel geospatial task

I really like that the authors introduced a novel geometric prediction task into the literature. In reading the literature for this review, it struck me that many of the benchmarks for TDL were somewhat old and outdated. It seemed appropriate that the task featured integration of data over different dimensional regions (points, lines, cells) in a way that showcased the central feature of the paper — reconciling data with features on subspaces of differing dimension i.e. the ‘heterogeneous interactions’ promised in the abstract.

Computational

I think one of the main strengths of the paper is the computational benefits, and I would personally focus on this more in the introduction. The tailored lifting of molecular graphs to the higher order CCs dramatically decreases the number of higher order cell, which is a problem that plagues many architectures in learning on simplicial/cellular complexes. The section on computational complexity also demonstrates this in a robust mathematical setting.

**Weaknesses:**

TDL vs GDL: novelty as a conceptual framework

For me personally I find it hard to understand the framing of this as a part of an entirely new conceptual field of topological deep learning beyond GNNs, and question the genuine novelty of papers like this. This is a concern I have of the field of TDL more generally, but I hope that the authors may be able to help clarify given their excellent communication skills demonstrated in the paper.

Unless I’m mistaken, the basic content of proposition (1) is that an ETNN can be reformulated as an EGNN. This means that the main novelty is the clever choice of ‘lifting' the data into a certain graph and some delination of the learning based on ‘rank’. Indeed, even the proof of theorem (1) is basically a straightforward adaption of the corresponding result for EGNNs. I think the need to reformulate everything as an ETNN then show it’s equivalent to some specific EGNN needs more justification.

I still think that the experiments, results and set-up of the paper are interesting, but I personally get the sense that the ’topology’ part — and hence the novelty of these kinds of architectures — is overplayed a little.  One gets the impression from reading the introduction that there is some topological thing deep down somewhere that is making the difference in performance, whereas my feeling is that the inclusion of domain-specific data — functional groups, rings, etc. — along with the design of the graph is doing most of the work.

On a similar theme, I don’t personally see a strong connection with the ‘lifted’ combinatorial complexes used in this paper and topology in the classical sense. It’s true that cell complexes and simplicial complexes also have the structure of a combinatorial complex (as in Appendix C), but these classical objects have additional connections to topology — i.e. they are stratifications of a genuine topological space, posses a homology theory, etc.

QM7 Benchmarking comments

I find the approach to testing for this benchmark slightly unfair on other methods. Searching through so many iterations of hyperparameters, it seems inevitable that eventually some parameter set will outperform current methods at least once just on the balance of probabilities rather than model strength. ﻿Surely a fairer test would be run the competing methods along a similarly vast set of hyperparameters? At minimum, this experiment should be run multiple times, with the variance on results included, to test the robustness of these results.

Could the authors elaborate on why they chose to focus only on comparing the ETNN only with EGNN in the bottom line of Table 1? Upon first look, I assumed that ETNN outperformed SotA, but then on closer look saw that Equiformer outperformed ETNN in (almost)( every category. I would recommend highlighting the best performing method in bold in each column so that it’s more immediately clear that the while ETNN is improving on EGNN, it is still behind the SotA methods like DimeNet++ and Equiformer. My personal opinion is that the bottom line should be removed — the fact that it outperforms EGNNs is not surprising considering the fact that EGNNs are somehow a sub-architecture given the results of the expressiveness appendix.

**Questions:**

Experiments

Could you provide more details on how functional groups were defined or identified in the QM7 dataset? Were these pre-annotated in the data or determined through some other process?

It also seems like many of the best results are given in hyper parameter configurations without the use of higher order cells. Can the authors comment on this? Does this detract from the main thrust of the paper, which is the inclusion of higher order cells?

Novelty: TDL vs GDL

Turning my comments into the weaknesses section in a specific question: what do we gain from introducing the notion of a combinatorial complex? As above, (1) the procedures for turning graphs into combinatorial complexes in this paper are not objects commonly studied in topology — i.e. do not correspond to a simplicial or cellular complex — so do not have access to additional mathematical theory and (2) we could seemingly equally well-formulate this paper as an EGNN over a specific procedure for turning the data into a graph﻿ as per Prop. 1. Why not just say this is a specific example of a EGNN using the construction in Prop 1?

I am very open to hear the author’s thoughts/be corrected on this!

Virtual cells

The virtual cell seems like to dramatically increase performance. However, my understanding is that this totally change the connectivity of the underlying data, meaning that
everything is now connected (as per the comment on 389). ﻿What does this say about the actual importance of the graph structure?

It seems to suggest the ’topology’ of the underlying molecular graph is basically ignored once the virtual cell is included. Why not just define the architecture as a family of fully connected EGNN over the nodes, edges and potentially the faces, given that the hyperparameters including the virtual cell almost always are superior?

On a related note: "it is clear how ETNN naturally handles heterogeneity, e.g., the same pair of bonds could be connected because part of the same functional group (up adjacency) and the virtual cell (max adjacency), but the two messages will be different across the neighborhood”. Unless I’ve missed something, how are we certain that the two types of propagation add to performance, rather than just overcomplicating the situtation?

Miscellaneous

Line 354: In the defintiion of geospatial CC, the mapping function s : X \to T takes cells to points in T, but the geographic space s(X) is now a subset of T rather than an element. Should s be mapping to the powerset of T, rather than T, if indeed it takes each cell to a subspace of T?

In general, it would better to have a more concrete definition of what is meant by a ‘geospatial combinatorial complex’. I’m also unfamilar with the term polyline.

Line 375: "This work is the first to explore combinatorial topological modeling of multi-resolution irregular geospatial data..” The claim that this is the first work to explore combinatorial topological modeling of multi-resolution irregular geospatial data seems overstated given existing literature in Topological Data Analysis applied to geospatial data. Could you clarify how your approach differs from or advances beyond these existing works?" https://arxiv.org/pdf/2104.00720, https://www.researchgate.net/publication/366891451_Topological_data_analysis_for_geographical_information_science_using_persistent_homology, https://pubmed.ncbi.nlm.nih.gov/26353267/

Table 2: It would be helpful to have the variance included over the multiple runs

Line 242: "geometric invariants should make use of the underlying topological structure.” The pairwise distance is more of a geometric structure than a topological one.

---

> ### Author Response · Authors · 2024-11-16
> **Addressing Reviewer's comments and questions**
>
> We appreciate the thorough reviewer’s constructive comments and the valuable questions. We would also like to thank the reviewer for pointing out as strengths the transparency of our experiments, the text clarity, the novelty of the geospatial task, and the computational efficiency of ETNN. Next, we address each point raised as weakness and answer all of the reviewer’s questions.
>
> ## Weaknesses.
> **[W1] TDL vs GDL: novelty as a conceptual framework**
>
> As long-time practitioners of TDL (and GDL) we the authors completely understand the point raised by the reviewer. However, there are multiple answers to why TDL is a motivated framework, spanning from (i) more conceptual (about the framing of TDL and GDL as fields in the ML landscape) to (ii) more pragmatic (about the usage of TDL models in real-world applications).
>
> About (i), the quick (but not exhaustive) answer is that GDL and TDL stem from two formally and inherently different perspectives on the same objects, “non-Euclidean spaces”.  GDL (in the sense of [1]) is built on group-theoretic arguments along with the frequent usage of Hilbert Spaces (strictly related to manifold learning and, in general, to metric spaces), while TDL is solely built on the modeling assumption of data living on the neighborhoods of a combinatorial and/or topological space and having a relational structure induced by the neighborhoods’ overlap. As such, TDL works tend to put more emphasis on what graph (or manifold) based models struggle to capture, e.g., higher-order combinatorial interactions. Overall, the intersection of GDL and TDL is clearly not empty, but we believe that both fields have a well-framed conceptual motivation. Further insights can be gained from the thesis in [2]-[3].
>
> About (ii), we firmly believe that TDL as a field needs stronger empirical evidence of its effectiveness. For this reason, in this work, we focused on proposing applications in which higher-order interactions (a) matter and (b) can be captured via ETNN (and some other TDL models in general). Indeed, and this is also a partial answer to **[Q2]** below, the aim of the proposed molecular CC and the novel geospatial benchmark is exactly to prove (a)-(b) true and, more, to also show that there are situations in which neither graphs nor higher-order combinatorial topological spaces (SC or CW complexes) can leverage the available information in a jointly exhaustive and computationally efficient way as CCs do. We thus prove that the modeling of higher-order interactions through CCs indeed offers benefits, especially for structured hierarchical data (e.g., molecular and geospatial data).
>
> Finally and most importantly, to the best of our knowledge, ETNN is currently the most exhaustive framework for merging scalarization-based equivariance arguments from GDL with pure TDL arguments.
>
> **[W2] ETNN vs EGNN**
>
> We kindly disagree with this observation since the aim of **Proposition 1** is not to show that ETNN is equivalent to some specific EGNN, but rather the opposite:  to show that some specific ETNN is computationally equivalent to a (non-standard) EGNN over a Geometric Augmented Hasse Graph. As such, the considered subclass  (that is the easiest possible) is not representative at all of the whole class of architectures that can be derived from the ETNN framework. EGNN is formally a particular case of ETNN. The “graph-based” method we introduce is only useful to easily and elegantly prove expressiveness in the geometric WL sense, which is dependent only on the relational structure and the geometric features induced by the CC, not on the specific ETNN architectural choice. This is an intuition already proved successful in works like [4] for non-geometric settings.
>
> **[W3] Classical Topology in TDL**
>
> This is a key comment (for this work, its novelty, and TDL in general). We agree that the inclusion of domain-specific data along with the design of a CC (not the graph) that can directly exploit them in a principled way,  is doing most of the work. However, given our reply to **[W1]**, we do not believe that this should not be considered topological (in a non-classical sense), as it is the higher-order relational structure induced by the underlying space that contributes to the effectiveness of ETNN. That said, the formal concepts coming from classic algebraic topology, e.g., homology groups, stratifications, filtrations, etc., arise when CCs are particularized to specific combinatorial topological spaces. Several works, e.g., [5]-[6], have shown improvements related to classical topological arguments. E(n) Equivariant versions of these architectures can be directly derived from the ETNN framework.
>
> *Continuing response in the thread (References are included in the last part of the thread)*

---

> > ### Author Response · Authors · 2024-11-16
> >
> > **[W4] QM9 Testing Unfairness**
> >
> > We kindly disagree with the reviewer, because we put most of our efforts into the fairness of the experiments.  At the beginning of Appendix J (and in Appendix I, as the reviewer mentioned), we exhaustively explain why we don’t run the model on multiple splits (each one of the baseline models uses different splits) and, in general, our meticulous care for reproducibility and fairness. Practical examples of this are also the fact that, despite its partial irreproducibility, we cared a lot about a fair comparison with EMPSN, or with an EGNN having our same parameter budget (ETNN-graph-W). Moreover, the reported results from the other baselines are indeed their strongest performances, occurring in their own experimentation setup. This means that the reported results of the other models have already undergone a hyperparameter tuning process in the corresponding papers. Overall, we believe that ETNN sets an unprecedented standard in the TDL community regarding exhaustiveness and transparency in the presentation of the results.
> >
> >
> > **[W5] QM9 ETNN vs EGNN**
> >
> > We report the improvement of ETNN over EGNN because our main focus is not showing overall SotA results (that, by the way, we mostly achieve in our class of interest Equiv+TDL) but rather the advantages of using CCs together with E(n) equi/invariance. We report the results of Equiformer exactly to be transparent and show what the current SotA for molecular graphs is (indeed it is already marked as SotA). As a side note, please notice that Equiformer and DimeNet are both tailored for molecules, while we show ETNN can be used in very different domains with very good performance. To make things clearer, we highlighted in bold the best-performing model in **Table 1** per each property. However, we believe that the improvement over EGNN should be kept (as the name of the row is not misleading).
> >
> > ## Questions.
> > **[Q1] Functional Groups**
> >
> > We extracted functional groups through RDKit’s substructure matching library. This can be seen in our shared repo, in the functional group lifting module. After the identification of the node subsets corresponding to functional groups, we filtered each group’s features through the following categorization:
> > | Functional Group | SMILES Pattern          |
> > |------------------|-------------------------|
> > | carboxyl         | C(=O)O                 |
> > | nitro            | [N+](=O)[O-]           |
> > | ketone           | [CX3](=O)[C]           |
> > | ester            | [CX3](=O)[OX2H0][#6]   |
> > | ether            | [OD2]([#6])[#6]        |
> > | amide            | [NX3][CX3](=[OX1])[#6] |
> > | benzene          | c1ccccc1               |
> > | aniline          | Nc1ccccc1              |
> > | phenol           | Oc1ccccc1              |
> > | carbamate        | [NX3][CX3](=[OX1])[OX2H0] |
> >
> > Further details can be found in the functional group lifting module in the shared repo.
> >
> > **[Q2] Contribution of Higher-Order Cells**
> >
> > We kindly disagree with the reviewer. In **Tables 10-11**, on 9 out of 11 properties, the variants of ETNN matching the best results use higher-order features. Moreover, please also notice that even when only bonds are used, we are still in a higher-order scenario, because ETNN leverages multiple adjacencies among edges that could not be leveraged with a GNN.
> >
> > **[Q2] Novelty: TDL vs GDL**
> >
> > Our comments in **[W1]-[W2]-[W3]** should jointly provide an exhaustive answer to this question. Of course, we are more than happy to elaborate more if needed during the discussion phase.
> >
> > **[Q3] Virtual Cell and Heterogeneity**
> >
> > The virtual cell increases performance in general, and this is expected, as the virtual node [7] had a similar effect on graph-based models. Defining a family of fully connected EGNNs over the different orders is computationally similar (but not equivalent, as in the case proposed by the reviewer is not clear how to handle geometric features) to every configuration in **Tables 10-11** that has only “max” in the Adjacencies column and “0” in the incidence column. As the reviewer can notice, only on 2 properties out of 11 these architectures perform better. About the heterogeneity, the reviewer is right, we cannot be certain that the two (or more) types of propagation add to performance, but this is just an additional amenable property of ETNN, and the user has full control over the architecture configuration, thus on the choice of including multiple types of relations or not.
> >
> >
> > *Continuing response in the thread (References are included in the last part of the thread)*

---

> ### Author Response · Authors · 2024-11-16
>
> **[Q4] Definition of Geospatial CC**
>
> Thank you for your careful reading and for bringing this omission to our attention. The corrected notation should indeed be $s: X \to \mathcal{P}(T)$.
>
> Regarding your second comment, we have revised the manuscript to include a more concrete definition of a geospatial CC (GCC) at the beginning of the relevant section. GCCs are now explicitly defined as CCs in which all cells are subsets of a geospatial domain. However, we believe that the mapping $s$ remains necessary for a more formal definition and to explain the construction of neighborhood systems.
>
> We have also clarified the definition of a polyline in the revised manuscript. It is now stated that a polyline is simply a sequence of connected lines, and we have included a citation to an introduction to common data formats in geographic information systems.
> We hope that these improvements to the wording highlighted in the revised manuscript provide greater clarity.
>
> **[Q5]  Combinatorial topological modeling of multi-resolution irregular geospatial data**
>
> As we explain in Appendix H.2, TDA methods have been applied to geospatial data (we added the missing references among the ones indicated by the reviewer). However, TDA is different from TDL (see again Appendix H.2), and in all the referenced works there is no combinatorial characterization induced by, e.g., political partition and no multi-resolution. They are mostly all variants of PH-related arguments applied to geospatial data.
>
> **[Q6] Variances in Table 2.**
>
> We originally did not report variances since our experiments are averaged over 3 seeds, which makes variance calculations unreliable. We are now working on running with additional seeds and will report the results in the next few days when the experiments are complete.
>
> **[Q7] Line 242**
>
> All the geometric invariants are geometric objects. However, their definition is totally dependent on the underlying topological combinatorial space. E.g., which are the pairwise distances to be summed are given by the relational structure of the space and its neighborhoods.
>
> ## References
>
> [1] Bronstein, Michael M., et al. "Geometric deep learning: Grids, groups, graphs, geodesics, and gauges." *arXiv preprint* arXiv:2104.13478 (2021).
>
> [2] Bodnar, Christian, “Topological Deep Learning: Graphs, Complexes, Sheaves”, *PhD Thesis*, [https://www.repository.cam.ac.uk/items/06b0b8e5-57d1-4120-8fad-643ce4d40eda](https://www.repository.cam.ac.uk/items/06b0b8e5-57d1-4120-8fad-643ce4d40eda) (2022).
>
> [3] Battiloro, Claudio, “Signal Processing and Learning over Topological Spaces”, *PhD Thesis*, [https://theses.eurasip.org/theses/974/signal-processing-and-learning-over-topological/](https://theses.eurasip.org/theses/974/signal-processing-and-learning-over-topological/) (2024).
>
> [4] Jogl, Fabian, Maximilian Thiessen, and Thomas Gärtner. "Expressivity-preserving GNN simulation." *Advances in Neural Information Processing Systems* 36 (2024).
>
> [5] Yang, Maosheng, Elvin Isufi, and Geert Leus. "Simplicial convolutional neural networks." *ICASSP 2022-2022 IEEE International Conference on Acoustics, Speech and Signal Processing (ICASSP)*. IEEE, 2022.
>
> [6] Battiloro, Claudio, et al. "Generalized simplicial attention neural networks." *IEEE Transactions on Signal and Information Processing over Networks* (2024).
>
> [7] Sestak, Florian, et al. "VN-EGNN: E (3)-Equivariant Graph Neural Networks with Virtual Nodes Enhance Protein Binding Site Identification." *arXiv preprint* arXiv:2404.07194 (2024).

---

> ### Author Response · Authors · 2024-11-21
>
> ​​**Standard errors for the geospatial task and additional seeds**
>
> Dear reviewer, we have conducted multiple additional runs for the geospatial task and computed standard errors based on your suggestion. We ran 10x more seeds (30 in total). Despite the high variance, the results are very similar to those reported in the paper. Standard errors are computed with the formula $\sigma/\sqrt{n}$, which is recommended when having at least sample size 30.
>
> | Baseline | Av. Explained Variance ($R^2$) | Std. Err ($R^2$) | **MSE** |
> | -------- | ------------------------------ | ---------------- | ------- |
> | ETNN     | 9.34%                           | 2.05%             | 0.935    |
> | GNN      | 2.44%                           | 0.99%             | 0.987    |
> | MLP      | 2.35%                           | 1.61%             | 1.022    |
> | EGNN     | 1.43%                           | 1.2%             | 1.041    |
> | Linear   | 0.51%                           | 0.95%             | 1.106   |
> **Updated Table 2a of the paper**. Baseline model comparison
>
> | Baseline                      | Diff. Explained Variance ($R^2$) | Std. Err  ($R^2$) | **MSE** |
> | ----------------------------- | -------------------------------- | ---------------- | ------- |
> | no geometric features (CCMPN) | -1.08%                            | 2.05%             | 0.946    |
> | no position update (invariant ETNN)     | -1.37%                            | 2.52%             | 0.956    |
> | no virtual node               | -1.80%                            | 2.32%             | 0.957    |
> |                               |                                  |                  |         |
> **Updated Table 2b of the paper**. Ablation study

---

> > ### Comment · Reviewer_NQcw · 2024-11-21
> >
> > I thank the authors for their detailled answers!
> >
> > [W1]
> >
> > I must admit I’m a little confused about this point. Is the Hilbert space structure due to the feature vectors above each node on the graph? If that’s the case, even in TDL we still have (Euclidean) feature vectors sitting above each node in the augmented hasse diagram etc. My understanding of the group-theoretic arguments was that they relate to permutation equivariance of basis vectors in the feature space, which equally applies to TDL given that neighbourhoods need Euclidean ﻿feature vectors. I take your point about the emphasis on the higher order interactions, and accept that this is indeed a different emphasis.
> > (ii) I totally agree that TDL needs more empirical justification, and that this paper is a good foundation for it. I do worry a little (as one of the other reveiwers) that something like the geospatial benchmark ﻿is perhaps a little over-designed specifically TDL methods to be of interest outside of the TDL community --- apart from that I’m on board.
> >
> > [W2]
> > I take your point. I think our only disgreement is the extent to which a “non-standard EGNN over a Geometric Augmented Hasse Graph” is a significantly novel deviation from a standard EGNN. This still feels (to me at least) well within the vicinity of message passing over a graph — albeit with non-uniform graded feature spaces. Again though, I understand your point that the novelty is that the actual Hasse graph emphasises higher-order combinatorial structure.
> >
> > [W3]
> > Fair enough.
> >
> > [W4]
> > OK good explanation — thanks for clarifying, in particular about the hyperparameters.
> >
> > [W5]
> > I appreciate the highlighting of the benchmarks!
> > ﻿
> > ﻿Relating to my previous point, I think that beating EGNNs will mainly be of interest to researchers within the TDL community rather than researchers more broadly. With that ﻿in mind, the bottom line still looks strange to me. It’s not a massive deal so I’m happy for it to be left in if that’s what the authors want.
> >
> >
> > [Q1]
> > Thanks for pointing out where to find this.
> >
> > [Q2]
> > Good explanation!
> >
> > [Q3]
> > Ah I see, I think that actually clarifies a misunderstanding I had. So the bonds still inherit the adacencies of neighbouring higher order cells even when we are not using the feature vectors of the higher order cells in the learning? Have I understood that correctly?
> >
> > [Q4]
> > Updated definition looks good to me!
> >
> > [Q5]
> > Thanks for adding references.
> >
> > [Q6]
> > Thanks for rerunning the experiments! Would it be possible to include the variances in the paper? I think it’s useful for the reader.
> >
> >
> > Based on the responses I'll be upgrading my score to a 6. Despite being a very well written and tested paper, the main issue I have is the extent of contribution and novelty, as I outline in my answers above. I think the paper deserves to be published in any case.

---

> > > ### Author Response · Authors · 2024-11-21
> > >
> > > We would like to thank the reviewer for the prompt feedback, their reconsideration of the score, and their opinion about the fact that our work deserves to be published. Overall, their comments have certainly improved the quality of our manuscripts. Here, we address their latest questions and comments. We are happy to work further to improve the already positive opinion the reviewer has of our work.
> > >
> > > **[W1]** We are happy to discuss this further. About GDL and TDL, the reviewer is totally right on everything, and the immediate main difference remains *the focus on higher-order interactions*.
> > >
> > > Our point was more subtle and related to the approach that the two disciplines *historically* followed during their development, keeping again in mind that the intersection is not empty and the object of study (non-Euclidean spaces) is loosely the same.
> > >
> > > On a more technical side, GDL uses mainly geometric arguments (norms, groups, metrics, and deformations are at the core of the framework) while TDL uses mainly topological arguments (again, our ETNN framework kind of mixes them).
> > >
> > > Given the scope and goals of this work and for the sake of clarity, we did not overcomplicate the definition of the signals. However, in general, in TDL (and Topological Signal Processing), a more exhaustive definition of signals is the vector representation (due to natural isomorphism) of the cochain spaces with coefficients in $\mathbb{R}$ associated with the simplicial/cell complex [1][2] (a more general but less theoretically rich notion of cochain space has been given for combinatorial complexes too [3], however, we believe the reviewer will resonate more with the standard notion). In this case, the cochain spaces are Hilbert spaces as well, as the reviewer correctly pointed out. However, to retain the whole machinery and results of the algebraic topology machinery (and, more in general, to retain the relational structure induced by the space), a cochain space is solely required to have an abelian group structure. This enables the usage of more exotic data structures, e.g. lattices [4]. Unfortunately, it is impossible to exhaustively treat this topic in this rebuttal, and it is more than sufficient to highlight the crucial importance of higher-order interactions. But, to wrap up, an intuitive but *absolutely not exhaustive* way of looking at this comparison, is mathematically looking at GDL as more of a top-down approach while TDL is a bottom-up. This *loose* difference is the main drive for modeling higher-order interactions (that in GDL-related spaces, e.g. manifolds,  grids, or graphs, are harder to derive).
> > >
> > > About the geospatial benchmark, we only partially agree on the specificity for the TDL community. We shall note that this is the first, to our knowledge, TDL work that utilizes a geospatial task as a benchmark, going beyond the more standard molecular tasks, which are widespread in this line of research, and, also, moving away from how classic biostatistics methods treat the downscaling problem. This is exactly what we find exciting in this application domain, and we built this task to show it. So, on one hand, it is true that the specific task is tailored to be a benchmark for TDL models (and this is good for the community as a whole) but on the other hand, we believe it will be useful for *general-purpose ML applications*. To provide the reviewer with a sense of it, we refer to the geospatial foundation model in [4], which appeared online only some days ago, that uses similar (but muchsimpler) arguments to ours for irregular multi-resolution geospatial data employing a heterogeneous graph jargon rather than a TDL one.
> > >
> > > **[W2]** We believe that a way to look at it is the following: ETNNs are very general architectures; to prove WL expressivity what really matters is the relational structure exploited by the network rather than its specific architectural instance; it then makes sense to use the simplest instance of ETNN (the one respecting the assumptions 1-3 in Proposition 1); this specific architecture results in a non-standard EGNN on the geometric augmented Hasse graph and it is sufficient to elegantly prove the WL expressiveness of ETNN.
> > >
> > > Therefore, the non-standard EGNN running on the geometric augmented Hasse graph is just a specific instance of ETNN. Indeed, this ETNN resulting from Assumptions 1-3, does not take into account either the different ranks (because of Assumptions 1-2), or the different geometric invariants (because of Assumption 3), or the different neighborhoods (because, from (27), the graph is built from the union over the neighborhoods of the complex). This collapses many relations induced by the topological domain and applies the same set of weights to the connections that survive the collapse.
> > >
> > > *Continuing the response in the next message. References are included, also, in the next message.*

---

> > > > ### Author Response · Authors · 2024-11-21
> > > >
> > > > *Continuing response*
> > > >
> > > > Take again the MolCC as an example: bonds are modeled as edges (1-cells) and rings such as carbon rings are modeled as faces (2-cells). Two bonds can simultaneously share multiple neighborhoods. For instance, they could be lower adjacent because they have a common atom (0-cell) and, at the same time, also be upper adjacent because they are part of the same molecular ring (2-cell). Despite their different chemical meaning, the whole geometric augmented Hasse graph would collapse these two relations (upper adjacent, lower adjacent) into one. Moreover, the resulting non-standard EGNN would not be able to distinguish anymore which node of the geometric augmented Hasse graph was an atom, a bond, or a ring in the original molecule, and would process all the connections with the same set of weights. However, and this motivates our approach and validates its elegance, the non-standard EGNN resulting from Assumptions 1-3 is as expressive as any general ETNN, as it is the most specific and constrained instance of it.
> > > >
> > > > [Q3] Exactly, as long as the adjacencies are in the collection of neighbrhoods, even if the higher-order cell features are not used.
> > > >
> > > > [Q6] Sure, we are collecting some more suggestions and we will update a further revision in the next few days.
> > > >
> > > > ## References.
> > > > [1] Battiloro, Claudio, et al. "Topological signal processing over weighted simplicial complexes." ICASSP 2023-2023 IEEE International Conference on Acoustics, Speech and Signal Processing (ICASSP). IEEE, 2023.
> > > >
> > > > [2] Roddenberry, T. Mitchell, Michael T. Schaub, and Mustafa Hajij. "Signal processing on cell complexes." ICASSP 2022-2022 IEEE International Conference on Acoustics, Speech and Signal Processing (ICASSP). IEEE, 2022.
> > > >
> > > > [3] Hajij, Mustafa, et al. "Topological deep learning: Going beyond graph data." arXiv preprint arXiv:2206.00606 (2022).
> > > >
> > > > [4] Ghrist, Robert, and Hans Riess. "Cellular sheaves of lattices and the tarski laplacian." arXiv preprint arXiv:2007.04099 (2020).

---

> ### Author Response · Authors · 2024-11-28
> **Follow-up after deadline extension**
>
> In light of the extension of the discussion deadline, we would like to thank the reviewer again for the already positive assessment of our work and for their engagement, and ask if there is anything else we can do to further improve the manuscript and its score.
>
> We also kindly invite the reviewer to parse the discussions we had with the other reviewers, as we believe that the overall engagement and the individual feedback have clarified or improved several aspects of the paper.
>
> Thanks a lot!

---

### Author Response · Authors · 2024-11-16
**Global Response**

We would first like to thank the reviewers for their detailed and constructive feedback. The paper is noted for its clarity (**Reviewers NQcw, h23Z**), rigor and elegance (**Reviewers NQcw, J6CW**), self-containment (**Reviewer h23Z**), computational efficiency (**Reviewer NQcw**), and transparency (**Reviewer NQcw, h23Z**). The expressivity analysis is overall found thorough. Moreover, we are really glad all the reviewers agreed on the importance to the TDL community of the two real-world applications and the novel geospatial benchmark. Overall, the paper is seen as addressing important gaps in the TDL literature with theoretical and practical contributions.

A point-by-point reply to all the comments is individually given in the sequel. We reply to comments in the same order they were presented by the reviewers, and we also group them by "Weaknesses" and "Questions". All the changes in the **revised manuscript** appear in blue color, to facilitate checking. We also summarize here our discussion about the novelty of our work.

**Novelty.**

The reviewers indicated novelty concerns about our work. However, we believe that this work is conceptually,  technically, and practically novel.

ETNN (and TDL in general) is **conceptually** novel and motivated because, Geometric Deep Learning (GDL) and TDL stem from two formally and inherently different perspectives on the same objects, “non-Euclidean spaces”.  GDL (in the sense of [1]) is built on group-theoretic arguments along with the frequent usage of Hilbert Spaces (strictly related to manifold learning and, in general, to metric spaces), while TDL is solely built on the modeling assumption of data living on the neighborhoods of a combinatorial and/or topological space and having a relational structure induced by the neighborhoods’ overlap. As such, TDL works tend to put more emphasis on what graph (or manifold) based models struggle to capture, e.g., higher-order combinatorial interactions. Overall, the intersection of GDL and TDL is clearly not empty, but we believe that both fields have a well-framed conceptual motivation. Further insights can be gained from the thesis in [2]-[3]. Overall, to the best of our knowledge, ETNN is currently the most exhaustive framework for merging scalarization-based equivariance arguments from GDL with pure TDL arguments.

On the one hand, it is true that ETNNs are scalarization-based architectures, as EGNNs, on higher-order combinatorial domains, as EMPSNs. As such, they inevitably resound with each other.

However, ETNN is **technically** novel as it is a formal and more expressive generalization of both EMPSNs and EGNNs. For this reason, ETNNs unlock several features (e.g. arbitrary modeling of heterogeneous hierarchical higher-order interactions, tunable expressive power, and general learnable geometric invariants) that the constrained graph or simplicial structures of EGNNs and EMPSNs cannot accommodate (see Appendix C and F). As such, our framework can be used to design arbitrary E(n) equivariant TDL models. No other framework has the same power. The expressivity analysis is novel as well. In particular, our approach is, to the best of our knowledge, currently the most exhaustive one for scalarization-based equivariant TDL models, as it could be applied to analyze the (geometric WL) expressivity of any scalarization-based equivariant TDL model without the need for domain-specific coloring procedures.

ETNN is **practically** novel because it tackles the fundamental problem of TDL as a field needing stronger empirical evidence of its effectiveness, as described in [4]. In this work, we focused on proposing applications of vastly different scales in which higher-order interactions (a) matter and (b) can be captured via ETNN (and some other TDL models in general). Indeed, the aim of the proposed molecular CC and the novel geospatial benchmark is exactly to prove (a)-(b) true and, more, to also show that there are situations in which neither graphs nor higher-order combinatorial topological spaces (SC or CW complexes) can leverage the available information in a jointly exhaustive and computationally efficient way as CCs do. Moreover, the experiments also show the versatility of ETNN in tackling very different problems. We thus prove that the modeling of higher-order interactions through CCs indeed offers benefits, especially for structured hierarchical data (e.g., molecular and geospatial data). As a consequence, we achieved or matched SotA results among the Equivariant TDL models with a huge reduction in computational complexity. As a byproduct, as **Reviewer NQcw** noticed too, our air pollution downscaling task represents a novel benchmark for the TDL community.

Finally, as a side minor note, we'd like to highlight the outreaching scope of our work. In its current form, it is sufficiently self-contained to be accessible to any ML practitioner, expert, or non-expert.

*The references are included in the next thread.*

---

> ### Author Response · Authors · 2024-11-16
>
> ## References.
> [1] Bronstein, Michael M., et al. "Geometric deep learning: Grids, groups, graphs, geodesics, and gauges." arXiv preprint arXiv:2104.13478 (2021).
>
> [2] Bodnar, Christian, “Topological Deep Learning: Graphs, Complexes, Sheaves”, PhD Thesis, https://www.repository.cam.ac.uk/items/06b0b8e5-57d1-4120-8fad-643ce4d40eda (2022).
>
> [3] Battiloro, Claudio, “Signal Processing and Learning over Topological Spaces”, PhD Thesis, https://theses.eurasip.org/theses/974/signal-processing-and-learning-over-topological/ (2024).
>
> [4]  Papamarkou, Theodore, et al. "Position: Topological Deep Learning is the New Frontier for Relational Learning." Forty-first International Conference on Machine Learning. 2024.

---

### Author Response · Authors · 2024-12-03
**Final Global Response**

We thank the reviewers for their valuable feedback and for their constant engagement in the rebuttal period. **The quality of the reviews was particularly high and our manuscript surely benefited from this and from the exhaustive and constructive communication we had the possibility to constantly have with the reviewers during the discussion period**. Thank you.

Overall, **all** the reviewers agreed that this work is a significant contribution, **deserves to be published** (NCQw, wwvW, J6CW) and it is a **critical** work for the TDL field (h23Z).

Although novelty seems to *partly* remain somehow a concern, we would like to stress that *architectural novelty* is not representative of the novelty of our work as a whole. **Most importantly**, it is undoubtedly true that ETNN is a simple equivariant generalization of CCMPNs [1], being a scalarization-based equivariant model, similar to  EMPSN [2]  being a simple generalization of MPSNs [3], and EGNN [4]  being a simple generalization of MPNs [5]. However, beyond the technical differences we already described, overall we showed that ETNNs are **more general and flexible**, **more expressive**, and **similar or better performing while being hugely more scalable** than SotA methods in the same class. These facts, together with the **benchmark**, **software**, and **outreaching** contribution of our work, should motivate its publication. This said, we already wrote in the future directions section that beyond scalarization-based architectures are of great interest. However, given that **a gap in the literature was still present for scalarization-based models** too, it made sense to us to **first comprehensively fill it** by working in the direction of ETNN.


Finally, elaborating more to strengthen our previous global response, ETNN has:

(a) **Architectural** novelty: ETNN improves on the two most important components of a scalarization-based equivariant architecture, i.e., feature updates and geometric invariants. Regarding features update, graphs and SCs induce specific neighborhood functions (node adjacency for the former, boundary, coboundary, up/down adjacencies for the latter), thus if a CC comes with a non-standard collection of neighborhoods, neither EGNN nor EMPSN can handle it, both for non-geometric and geometric features updates. Regarding geometric invariants, EGNN/EMPSN uses tailored geometric invariants for graphs/SCs. Imagine applying EMPSN on a CC. Then, computing the volume of a non-simplex makes no sense. Similarly, pruning the pairwise distances based on the simplicial structure or using the angles of the planes induced by the dimension of the simplices makes no sense. On the other hand, ETNN can handle arbitrary neighborhoods, and its geometric invariants work on arbitrary CTSs (and formally generalize some of the invariants of EMPSN and EGNN). We also enriched the characterization of the geometric invariants with our ablation presented in the reply to reviewer J6CW, that will be extended and integrated in the camera-ready version of the paper.

(b) **Experimental** novelty: the introduction of the new real-world geospatial benchmark and our novel approach for molecular modeling represent a significant effort to tackle some of the **most relevant open problems of the TDL field** (as described in [1] and recognized by multiple reviewers) and are a significant source of novelty (and an important resource for the community) as well.

(c ) **Theoretical** novelty: ETNN is the only available framework for designing *arbitrary* E(n) equivariant scalarization-based TDL models, and the expressivity proof leveraging the novel notion of geometric augmented Hasse graph has not appeared before in the literature of equivariant TDL.

**References**

- [1] Hajij et al. (2022) *Topological deep learning: Going beyond graph data*. arXiv:2206.00606.
- [2] Eijkelboom et al. (2023). *E(n) Equivariant Message Passing Simplicial Networks*. ICML.
- [3] Bodnar et al. (2021). *Weisfeiler and lehman go topological: Message passing simplicial networks*. ICML.
- [4] Satorras et al. (2021). *E(n) equivariant graph neural networks*. ICML.
- [5] Gilmer et al. (2017). *Neural message passing for Quantum chemistry*. ICML.

---

### Meta-Review · Area_Chair_z9ho · 2024-12-22

**Metareview:**

This paper extends TNN to ETNN, similar to the extension of GNN to EGNN. There have been very intensive discussions among the authors and reviewers, and the authors put a lot of efforts during the process. Eventually, I believe all the major issues have been resolved. I agree with some of the reviewers that the work is not entirely novel given the similarity with extension of GNN to EGNN. In this sense, I tend to think this is a borderline paper. On the other hand, given the many empirical comparisons and insights of add equivariance to existing methods, I tend to recommend an accept.

**Additional Comments On Reviewer Discussion:**

There have been extensive discussions and all the major issues have been resolved.

---

### Decision · Program_Chairs · 2025-01-22

Accept (Poster)